# Dimension-Free Multimodal Sampling
# via Preconditioned Annealed Langevin Dynamics

**Lorenzo Baldassari** [1]   **Josselin Garnier** [2]   **Knut Sølna** [3]   **Maarten V. de Hoop** [4]

## Abstract

Designing sampling algorithms for multimodal targets that remain stable under refinement of the finite-dimensional approximation of an underlying function-space problem is a central challenge. Annealed Langevin dynamics (ALD) is a natural alternative to classical Langevin in this context, since it is often observed to improve exploration across modes. Yet a gap remains between its empirical success and existing theory: under which conditions can ALD be guaranteed to remain stable across dimensions? In this paper, we bridge this gap by providing a uniform-in-dimension analysis of continuous-time ALD for Gaussian-mixture targets. Along an explicit annealing path obtained by gradually removing Gaussian smoothing from the target, we identify spectral conditions linking the smoothing covariance to the component covariances under which ALD achieves a prescribed accuracy in Kullback–Leibler divergence within a dimension-uniform time horizon. We then establish stability in a perturbative regime with imperfect initialization and approximate scores. Under a misspecified-mixture score model, we show that preconditioning ALD with an operator whose spectrum decays sufficiently fast prevents error terms from accumulating across coordinates and thereby preserves dimension-uniform control.

## 1. Introduction

Langevin dynamics is a standard tool for sampling from a given probability distribution (Parisi, 1981; Roberts & Tweedie, 1996; Durmus & Moulines, 2017; Dalalyan, 2017; Durmus & Moulines, 2019). Its appeal lies in the fact that it

defines a dynamics whose invariant law is the target distribution. Its limitations, however, are also well known. While it can mix rapidly when the target is strongly log-concave or satisfies suitable isoperimetric inequalities (Durmus et al., 2019; Vempala & Wibisono, 2019; Chewi et al., 2025), its performance can deteriorate dramatically outside these regimes, especially for multimodal distributions: transitions between modes may occur only on very long time scales, leading to poor finite-time sampling accuracy, and this difficulty typically becomes more severe as the dimension increases (Ma et al., 2019; Schlichting, 2019; Dong & Tong, 2022). Rigorous results document such failures in a variety of settings, including Bayesian posteriors arising from nonlinear inverse problems (Bohr & Nickl, 2024; Bandeira et al., 2023; Nickl, 2023).

These challenges motivate annealed Langevin dynamics (ALD) as a sampling procedure. Building on the empirical success of annealing-based methods (Kirkpatrick et al., 1983; Gelfand & Mitter, 1990; Neal, 2001), ALD applies Langevin dynamics along a path of intermediate distributions: it starts from a heavily smoothed, and hence easier-to-sample, law, and then gradually removes the smoothing until the path approaches the desired target (Song & Ermon, 2019; Block et al., 2020). Empirically, this annealing strategy is often reported to improve exploration across modes and to yield substantially better finite-time sampling accuracy (Song & Ermon, 2020; Zilberstein et al., 2022; Sun et al., 2024). Recent work has begun to clarify why ALD can outperform classical Langevin under minimal assumptions, showing that annealing can improve worst-case complexity bounds for reaching a prescribed Kullback–Leibler accuracy from exponential to polynomial dependence on dimension (Guo et al., 2025). Such bounds, however, may still deteriorate as the dimension increases. This leaves open the stronger question of whether ALD can provide dimension-uniform sampling guarantees, as required when finite-dimensional approximations of an infinite-dimensional multimodal target are progressively refined. This question has been studied extensively for Langevin diffusion (Hairer et al.; 2007), but has not yet been rigorously addressed for ALD. Dimension-robustness, however, is most critical in multimodal sampling problems, where classical Langevin can fail.

[1]University of Basel [2]École Polytechnique, IP Paris [3]University of California Irvine [4]Rice University. Correspondence to: Lorenzo Baldassari <lorenzo.baldassari@unibas.ch>.

*Proceedings of the 43rd International Conference on Machine Learning*, Seoul, South Korea. PMLR 306, 2026. Copyright 2026 by the author(s).

A simple illustration suggests that this goal may not be out of reach. Figure 1 considers, for each dimension $d$, sampling from a bimodal Gaussian mixture on $\mathbb{R}^d$ with weights $(w_1, w_2) = (0.75, 0.25)$, means $(0, 10e_1)$, and covariances $\tau_1 \Sigma^d$ and $\tau_2 \Sigma^d$, where $\tau_1 = 1.2$, $\tau_2 = 2$, and $\Sigma^d$ has a decaying spectrum $(j^{-2})_{j=1}^d$. Fixing $\epsilon = 0.3$, we ask whether the number of ALD steps needed to reach this accuracy grows with $d$. The target is clearly multimodal, and under a default flat-spectrum choice of smoothing and preconditioning, ALD already exhibits a pronounced dependence on $d$, in line with existing pessimistic bounds (Guo et al., 2025); for $d > 10$ the required number of steps exceeds our 20000-step cap. However, with suitable design choices, in particular replacing a flat smoothing spectrum with a sufficiently decaying one, ALD reaches the prescribed accuracy with no visible deterioration in the required number of steps as the dimension increases.

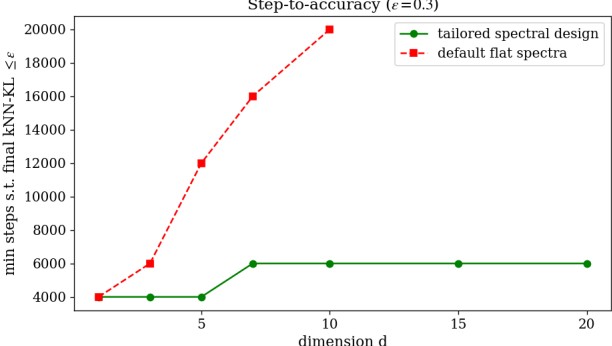

*Figure 1.* ALD step count vs. dimension. Number of ALD steps required for the empirical $\mathrm{KL}(\rho_\star^d \| \rho_T^{\mathrm{ALD},d})$ to fall below the prescribed accuracy $\epsilon = 0.3$, plotted against the truncation dimension $d$. The red curve corresponds to the default flat-spectrum choice, with preconditioner $\Gamma^d = I$ and smoothing $C^d = 40I$, while the green curve corresponds to a tailored spectral design: $\Gamma = \mathrm{Diag}(j^{-1.5})_{j=1}^d$ and $C = \mathrm{Diag}(40 \cdot j^{-2.7})_{j=1}^d$. ALD is described in Section 3.

Although only illustrative, this toy example highlights a gap between existing theory and the empirical behavior observed here. This motivates the central question of this paper: *Can ALD remain stable as the dimension increases, and can this be proved rigorously?* Our main message is that it can. Specifically, we identify an ALD design that yields dimension-uniform sampling guarantees for multimodal distributions in continuous time, and we derive conditions under which these guarantees remain stable under score and initialization misspecification. Crucially, this stability is not automatic: it relies on suitable preconditioning of the ALD diffusion.

Our analysis is carried out for Gaussian mixture models (GMMs), a classical and widely used family for approximating multimodal distributions (McLachlan & Peel, 2000).

For each dimension $d$, we consider targets of the form

$$\rho_\star^d = \sum_{i \in I} w_i \mathcal{N}(m_i^d, \Sigma_i^d),$$

where $I$ may be finite or countably infinite, the weights satisfy $w_i > 0$ and $\sum_{i \in I} w_i = 1$, and $(m_i^d, \Sigma_i^d)$ denote the component means and covariances in $\mathbb{R}^d$. We view the family $(\rho_\star^d)_{d \geq 1}$ as a sequence of increasingly accurate finite-dimensional approximations of an infinite-dimensional target, obtained for example by projecting the target onto its first $d$ coordinates; see Section 2. This setting is general yet tractable: it lets us make explicit how the annealing schedule, the preconditioner, and the mixture geometry (means, covariances, and weights) jointly determine when the continuous-time ALD diffusion achieves a prescribed sampling accuracy within a dimension-uniform time horizon; see Section 3. It also allows us to analyze the effect of score and initialization mismatch; see Section 4. These results underline the importance of preconditioning for stability as the dimension increases; this is illustrated numerically in Section 5. Combined with the discretization analysis in (Baldassari et al., 2026a), our results provide a unified picture of ALD in infinite dimensions.

**Related Work**

Annealing-based strategies have a long history in sampling (Kirkpatrick et al., 1983; Gelfand & Mitter, 1990; Neal, 2001). They replace direct sampling from the target by sampling along a path of intermediate distributions, starting from an easier law and gradually transforming it into the target. Implementing this principle through a Langevin diffusion driven by a time-dependent score schedule leads to annealed Langevin dynamics (ALD).

Recent work has begun to develop theoretical guarantees for ALD and related annealing-based samplers, both from the sampling and generative-modeling perspectives, often under minimal assumptions on the target distribution (Lee et al., 2018; Guo et al., 2025; Vacher et al., 2026; Cattiaux et al., 2025; Cordero-Encinar et al., 2025). Part of this renewed interest is connected to the popularity of score-based generative models (Song & Ermon, 2019; 2020; Block et al., 2020; Song et al., 2021; Albergo et al., 2025), which rely on time-dependent drifts. A related example in conditional generation, discussed in Appendix C, is classifier-free guidance (Pavasovic et al., 2025; Yehezkel et al., 2025; Wang et al., 2024), where conditional and unconditional scores are combined with a guidance weight that is often time-dependent, stronger at early times and tapered later.

Among theoretical analyses of ALD, the closest result to ours is that of Guo et al. (2025): as in that work, we formulate guarantees in terms of Kullback–Leibler (KL) divergence, which implies convergence in total-variation dis-

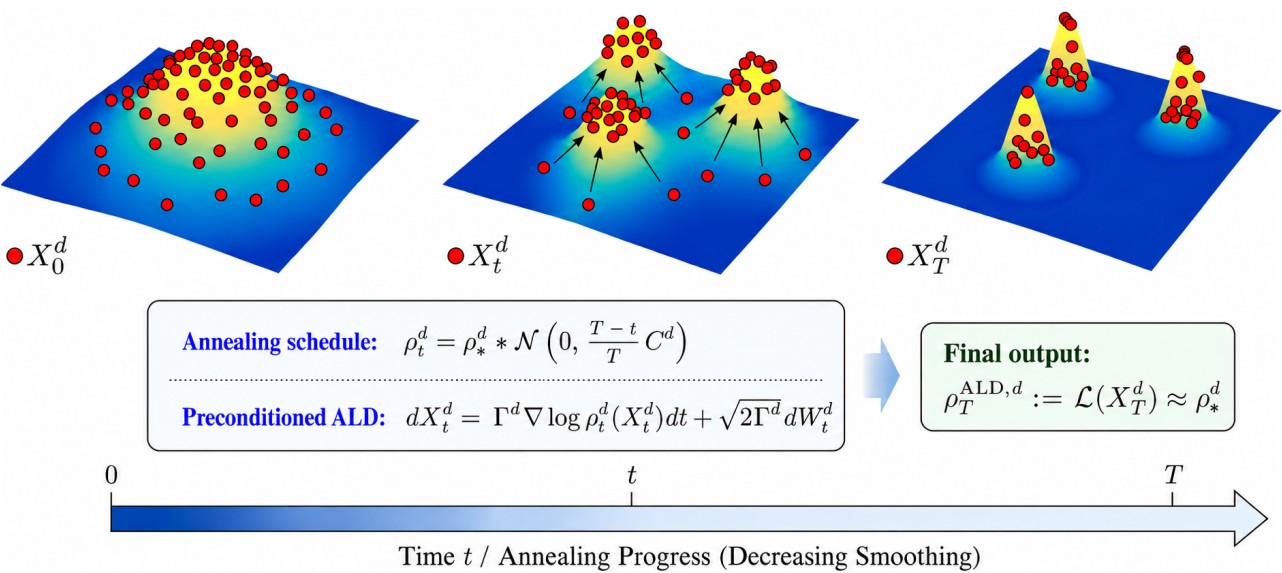

*Figure 2.* Annealing from smoothing to multimodality. The annealed Langevin dynamics scheme considered here starts from a heavily smoothed, and hence tractable, version of the target and simulates a Langevin diffusion whose drift is adapted to the current smoothing level. As time progresses, the smoothing is gradually removed, so the drift guides the dynamics from an almost unimodal distribution towards the original multimodal target through a controlled "complexification" of the landscape.

tance, and bound the global error along the entire curve of probability measures rather than focusing only on the local error at final time, following an approach inspired by (Dalalyan & Tsybakov, 2012; Chen et al., 2022). Compared to Guo et al. (2025), however, our goal is not only to quantify the benefit of annealing for a fixed finite-dimensional target, but to prove dimension-robustness for a family of successive finite-dimensional approximations of an infinite-dimensional multimodal target. We also study a perturbative regime in which the score and the initialization are misspecified, and ask whether ALD remains stable under these errors. This is a standard question in sampling (Huggins & Zou, 2017; Dalalyan, 2017; Dalalyan & Karagulyan, 2019), but it requires special care here because errors can accumulate across high-frequency coordinates as the dimension increases.

One main contribution of our analysis is to make explicit how dimension-uniform sampling guarantees for ALD can be enforced through the spectral properties of both the annealing geometry and the preconditioner. This connects naturally to the classical literature on function-space sampling, where the target is a probability measure on a function space and computation relies on finite-dimensional approximations (Stuart, 2010; Cotter et al., 2013; Hairer et al., 2014). These works have long emphasized that formulating the sampling problem directly in infinite dimension makes it possible to design principled preconditioning schemes

whose behavior remains stable as the approximation is refined (Roberts & Rosenthal, 1998; Hairer et al.; 2007; Cotter et al., 2013; Hairer et al., 2014; Cui et al., 2016; Beskos et al., 2017); this is a principle we follow here.

Similar ideas have recently received renewed attention in the context of infinite-dimensional diffusion and score-based generative models (Kerrigan et al., 2023; Franzese et al., 2023; Baldassari et al., 2023; Pidstrigach et al., 2024; Bond-Taylor & Willcocks, 2024; Baldassari et al., 2024; Lim et al., 2025; Hagemann et al., 2025; Baldassari et al., 2026b; Franzese & Michiardi, 2025). While our work assumes access to the annealed scores and takes a sampling perspective, it can be viewed as parallel to recent analyses of preconditioned score-based generative models (Pidstrigach et al., 2024; Baldassari et al., 2026b), with the crucial difference that the preconditioning conditions derived here are tailored to multimodal target distributions.

## 2. Problem Setting

Our goal is to understand when continuous-time annealed Langevin dynamics can sample robustly across the successive refinements $(\rho_\star^d)_{d \geq 1}$ of an infinite-dimensional multimodal target distribution $\rho_\star^\infty$. As the dimension $d$ grows, resolving progressively finer features of the same target, we ask (i) whether the sampling error can be controlled uniformly in $d$ within a single time horizon when the annealed

scores are available exactly; and (ii) under which conditions this control remains stable under approximation errors in the annealed score and imperfect initialization.

We formulate this question in a Hilbert space setting where refinement has a simple spectral meaning. Let $H$ be a separable Hilbert space with orthonormal basis $(e_j)_{j\geq 1}$. We consider a Gaussian mixture on $H$ of the form

$$\rho_\star^\infty = \sum_{i\in I} w_i \mathcal{N}(m_i, \Sigma_i),$$

where $I$ is finite or countably infinite, $w_i > 0$ for all $i \in I$, and $\sum_{i\in I} w_i = 1$. The truncations below will be taken in this basis, and we assume that the component covariances are diagonal with respect to it:

$$m_i = \sum_{j\geq 1} m_{ij} e_j, \qquad \Sigma_i e_j = \sigma_{ij} e_j, \qquad \sigma_{ij} > 0.$$

At first sight, this diagonal assumption may appear restrictive. However, after truncation to any fixed dimension $d$, it still gives a rich finite-dimensional multimodal model: diagonal Gaussian mixtures, with sufficiently many components, can approximate general mixtures in KL distance.

To ensure that $\rho_\star^\infty$ is a well-defined probability measure on $H$, we impose the following summability condition.

**Assumption 2.1** (Finite-energy). *For each component $i \in I$ of $\rho_\star^\infty = \sum_{i\in I} w_i \mathcal{N}(m_i, \Sigma_i)$,*

$$\sum_{j\geq 1} m_{ij}^2 < \infty, \qquad \sum_{j\geq 1} \sigma_{ij} < \infty.$$

Assumption 2.1 is the Hilbert-space analogue of a finite-energy condition. It ensures that $m_i \in H$ and that the covariance operators $\Sigma_i$ are trace-class, so that each $\mathcal{N}(m_i, \Sigma_i)$ defines a Gaussian measure on $H$ (Hairer, 2009). Hence $\rho_\star^\infty$ is a well-defined mixture measure on $H$, and its finite-dimensional refinements can be obtained by projection.

More precisely, the $d$-dimensional targets are the marginals of this fixed measure on the first $d$ basis coordinates. Let $P_d$ be the orthogonal projection onto $\mathrm{span}\{e_1, \ldots, e_d\}$. We set

$$\rho_\star^d := (P_d)_\# \rho_\star^\infty = \sum_{i\in I} w_i \mathcal{N}(m_i^d, \Sigma_i^d), \qquad (1)$$

where

$$m_i^d = (m_{i1}, \ldots, m_{id}), \qquad \Sigma_i^d = \mathrm{Diag}(\sigma_{i1}, \ldots, \sigma_{id}).$$

Increasing $d$ therefore adds new coordinates in the same basis while leaving the previously retained coordinates unchanged. The sequence $(\rho_\star^d)_{d\geq 1}$ represents successive finite-dimensional approximations of a single target $\rho_\star^\infty$, not a collection of unrelated high-dimensional problems. This is the regime in which dimension-uniform control is meaningful.

## 3. Annealed Langevin Dynamics

The targets $\rho_\star^d$ are multimodal, and direct Langevin sampling is well known to struggle in this setting, even at moderate dimension. In this section, we introduce annealed Langevin dynamics (ALD), which replaces direct sampling from $\rho_\star^d$ by Langevin diffusion along a prescribed sequence of progressively less smoothed targets. We focus on the Gaussian-smoothing variant of ALD, sketched in Figure 2 and formalized below in two steps.

**Step 1 (annealing schedule).** Let $C^d$ be a positive-definite matrix, diagonal in the basis $(e_j)_{j=1}^d$ with eigenvalues $(\lambda_j)_{j=1}^d$. We assume that these eigenvalues are the first $d$ terms of a summable sequence $(\lambda_j)_{j\geq 1}$, namely

$$\sum_{j\geq 1} \lambda_j < \infty.$$

Thus the limiting smoothing covariance is trace-class, so that the corresponding Gaussian perturbation is $H$-valued.

For a time horizon $T > 0$, we use the linear smoothing schedule

$$\kappa_t := \frac{T-t}{T}, \qquad t \in [0, T],$$

and define the Gaussian-smoothing annealing path

$$\rho_t^d = \rho_\star^d * \mathcal{N}\left(0, \kappa_t C^d\right), \qquad t \in [0, T]. \qquad (2)$$

Here $\kappa_t C^d$ is the added covariance, which decreases linearly from $C^d$ to 0. Thus the path interpolates from the smoothed law $\rho_0^d = \rho_\star^d * \mathcal{N}(0, C^d)$ to the target $\rho_T^d = \rho_\star^d$: initially the target is smoothed and easier to explore, while as $t \to T$ the smoothing vanishes.

Finally, the linear choice of $\kappa_t$ is made only for clarity: all arguments below extend to regular decreasing schedules with $\kappa_0 = 1$ and $\kappa_T = 0$.

**Step 2 (time-inhomogeneous dynamics).** We next define a Langevin diffusion driven by the time-dependent drift $\nabla \log \rho_t^d$: starting from the initial smoothed law, the dynamics uses progressively less smoothed scores as $t$ approaches $T$. Specifically, we initialize $X_0^d \sim \rho_0^d$ and evolve

$$dX_t^d = \Gamma^d \nabla \log \rho_t^d(X_t^d)\, dt + \sqrt{2\Gamma^d}\, dW_t^d, \qquad t \in [0, T], \qquad (3)$$

where $(W_t^d)_{t\geq 0}$ is a standard Brownian motion in $\mathbb{R}^d$, and $\Gamma^d$ is a positive-definite matrix, diagonal in the basis $(e_j)_{j=1}^d$ with eigenvalues $(\gamma_j)_{j=1}^d$. The matrix $\Gamma^d$ acts as a preconditioner, and its role will be discussed later.

The ALD output distribution, which we use as an approximation of $\rho_\star^d$, is the law of the process at the final time:

$$\rho_T^{\mathrm{ALD}, d} := \mathrm{Law}(X_T^d).$$

There is an important distinction to keep in mind, which helps clarify the analysis that follows. Standard Langevin dynamics targeting $\rho_\star^d$ uses the fixed drift $\nabla \log \rho_\star^d$. In contrast, ALD uses the time-dependent score $\nabla \log \rho_t^d$, so the dynamics (3) is time-inhomogeneous. Consequently, one should not expect the law $\rho_t^{\mathrm{ALD},d} := \mathrm{Law}(X_t^d)$ to coincide with the instantaneous target $\rho_t^d$; in particular, $\rho_T^{\mathrm{ALD},d}$ may differ from $\rho_\star^d$. We quantify this annealing-induced bias at the final time $T$ through the Kullback–Leibler divergence

$$\mathcal{B}_{\mathrm{ann}}^d(T) \; := \; \mathrm{KL}\big(\rho_\star^d \,\|\, \rho_T^{\mathrm{ALD},d}\big).$$

In what follows, we show that the time $T$ required to make this bias small is dictated by the schedule, by the smoothing and preconditioning geometries through $C^d$ and $\Gamma^d$, and by the mixture covariances $(\Sigma_i^d)$. Under suitable spectral conditions, this time can be chosen uniformly in dimension.

**Annealing-Induced Bias**

To isolate the source of this bias, it is convenient to introduce a reference time-inhomogeneous diffusion that *does* follow the annealing path exactly. Specifically, we consider a time-dependent vector field $v^d = (v_t^d)_{t\in[0,T]}$ and the SDE

$$dY_t^d = \Gamma^d\big(\nabla \log \rho_t^d + v_t^d\big)(Y_t^d)\, dt + \sqrt{2\Gamma^d}\, dW_t^d, \quad (4)$$

where $v^d$ is chosen so that $\mathrm{Law}(Y_t^d) = \rho_t^d$ for all $t \in [0,T]$. Any such correction field induces the path-matching energy,

$$\mathcal{J}_{\mathrm{ann}}^{v^d}(T) \; := \; \frac{1}{4} \int_0^T \int_{\mathbb{R}^d} \|(\Gamma^d)^{1/2} v_t^d(x)\|^2 \, d\rho_t^d(x)\, dt, \quad (5)$$

where, with a slight abuse of notation, we use $\rho_t^d$ both for the law and for its density. Crucially, this energy upper bounds the annealing-induced bias $\mathcal{B}_{\mathrm{ann}}^d(T)$.

Thus, controlling the bias reduces to controlling the energy cost of the correction $v_t^d$ needed to match the annealing path, relative to the ALD drift $\nabla \log \rho_t^d$ used in (3). This is not merely a technical convenience: in general, $v_t^d$ is inaccessible, since identifying it amounts to solving a high-dimensional Fokker–Planck equation. Although for Gaussian mixtures such corrections can be characterized explicitly, we adopt this setting as a tractable proxy in which all quantities are explicit and the dependence on $(C^d, \Gamma^d)$ and on the component covariances $(\Sigma_i^d)$ can be tracked.

This leads to the paper's first question: how large must the annealing horizon be so that this bias stays within a prescribed accuracy $\epsilon$? In particular, how does the required time horizon scale with the dimension $d$? Theorem 3.1 answers this question by giving a time horizon $T$, depending explicitly on $d$, such that

$$\mathrm{KL}\big(\rho_\star^d \,\|\, \rho_T^{\mathrm{ALD},d}\big) \leq \epsilon.$$

We then use this expression to identify regimes in which the same accuracy $\epsilon$ can be achieved with a time horizon chosen uniformly in $d$. The result is proved in Appendix A under exact-score and perfect-initialization assumptions; both are relaxed in the next section.

**Theorem 3.1.** *Fix $d \geq 1$ and $\epsilon > 0$. Define*

$$\mathcal{K}_d := \frac{1}{16} \sum_{i\in I} w_i \sum_{j=1}^d \frac{\lambda_j}{\gamma_j} \log\Big(1 + \frac{\lambda_j}{\sigma_{ij}}\Big).$$

*Consider the ALD dynamics up to time*

$$T := \epsilon^{-1} \mathcal{K}_d.$$

*Then $\mathcal{B}_{\mathrm{ann}}^d(T) \leq \epsilon$.*

Theorem 3.1 shows that this bound yields a dimension-uniform horizon whenever

$$\sup_{d\geq 1} \mathcal{K}_d < \infty,$$

or equivalently whenever

$$\sum_{i\in I} w_i \sum_{j\geq 1} \frac{\lambda_j}{\gamma_j} \log\Big(1 + \frac{\lambda_j}{\sigma_{ij}}\Big) < \infty.$$

In such regimes, the time horizon required by the bound to reach accuracy $\epsilon$ does not deteriorate as $d$ increases, formalizing the dimension-stability suggested by Figure 1. Crucially, this condition can be enforced through the design of the annealing geometry and the preconditioner. Indeed, by choosing the Gaussian-smoothing spectrum $(\lambda_j)$ and the preconditioner spectrum $(\gamma_j)$ so that they are compatible with the component covariance spectra $(\sigma_{ij})$, and using $\log(1 + u) \leq u$ for $u \geq 0$, it is enough to require

$$\sum_{i\in I} w_i \sum_{j\geq 1} \frac{\lambda_j^2}{\gamma_j\, \sigma_{ij}} < \infty. \quad (6)$$

Finally, three remarks are worth noting:

- It may seem surprising that the component means $m_{ij}$ do not appear in (6). This is because Theorem 3.1 assumes an idealized setting; the means $m_{ij}$ play a significant role in the perturbative analysis of the next section, where the score is imperfect or the initialization is misspecified.

- Theorem 3.1 assumes exact initialization $X_0^d \sim \rho_0^d$; it does not quantify the cost of sampling from $\rho_0^d$. This cost is affected by the smoothing $C^d$: stronger smoothing makes $\rho_0^d = \rho_\star^d * \mathcal{N}(0, C^d)$ closer to a unimodal proxy and typically easier to sample from, but it also increases $\mathcal{K}_d$ since the annealing path starts farther from $\rho_\star^d$. Conversely, weaker smoothing shortens the path but leaves $\rho_0^d$ more multimodal, and therefore harder to sample from.

- One should not interpret the dependence of $\mathcal{K}_d$ on $(\gamma_j)$ as saying that preconditioning is a cost-free accelerator. Multiplying $\Gamma^d$ by a large scalar reduces the ratios $\lambda_j/\gamma_j$ appearing in the annealing-bias bound, but this mainly rescales the time scale of the diffusion. Under an Euler–Maruyama time discretization, this requires a correspondingly smaller stepsize to maintain stability; see (Baldassari et al., 2026a). Even in the continuous-time setting, however, there is a trade-off between making the annealing-bias condition smaller by increasing the coefficients $\gamma_j$ and choosing a preconditioner compatible with dimension-uniform stability. When score and initialization errors are introduced, a preconditioner whose spectrum decays sufficiently fast helps prevent coordinatewise errors from accumulating, as shown in the perturbative regime analyzed in the next section.

# 4. Stability under Score and Initialization Perturbations

We now move past the idealized setting of Section 3 and ask whether the dimension-uniform guarantees obtained there persist under perturbations. To separate these effects from the annealing-induced bias analyzed in the previous section, we use the ideal path-matching diffusion (4) as a reference and compare it with an approximate ALD dynamics driven by a misspecified score $\widetilde{s}_t^d$ and initialized from a possibly incorrect law $\widetilde{\rho}_0^d$:

$$d\widetilde{X}_t^d = \Gamma^d\,\widetilde{s}_t^d(\widetilde{X}_t^d)\,dt + \sqrt{2\Gamma^d}\,d\widetilde{W}_t^d, \qquad \widetilde{X}_0^d \sim \widetilde{\rho}_0^d.$$

We denote its law by

$$\widetilde{\rho}_t^{\mathrm{ALD},d} := \mathrm{Law}(\widetilde{X}_t^d),$$

and measure score mismatch through the pointwise error

$$\epsilon_t^d(x) := \nabla\log\rho_t^d(x) - \widetilde{s}_t^d(x).$$

The starting point of our analysis is the following upper bound, proved in Appendix B.1.

**Proposition 4.1.** *We have the upper bound*

$$\mathrm{KL}\big(\rho_\star^d \,\|\, \widetilde{\rho}_T^{\mathrm{ALD},d}\big) \;\le\; \mathcal{E}_{\mathrm{init}}^d + \mathcal{E}_{\mathrm{score},T}^d + \mathcal{E}_{\mathrm{bias},T}^d,$$

*where*

$$\mathcal{E}_{\mathrm{init}}^d := \mathrm{KL}(\rho_0^d \,\|\, \widetilde{\rho}_0^d),$$

$$\mathcal{E}_{\mathrm{score},T}^d := \frac{1}{2}\int_0^T\!\!\int \big\|(\Gamma^d)^{1/2}\epsilon_t^d(x)\big\|^2\,d\rho_t^d(x)\,dt,$$

*and*

$$\mathcal{E}_{\mathrm{bias},T}^d := 2\inf_{\substack{v^d=(v_t^d)_{t\in[0,T]}:\\ \partial_t\rho_t^d=-\nabla\cdot(\rho_t^d\Gamma^d v_t^d)}} \mathcal{J}_{\mathrm{ann}}^{v^d}(T),$$

*with $\mathcal{J}_{\mathrm{ann}}^{v^d}(T)$ defined in (5).*

The term $\mathcal{E}_{\mathrm{bias},T}^d$ is already controlled by the path-matching argument of Section 3. Indeed, by definition,

$$\mathcal{E}_{\mathrm{bias},T}^d = 2\inf_{v^d}\mathcal{J}_{\mathrm{ann}}^{v^d}(T),$$

and the proof of Theorem 3.1 shows that the same path-matching energy is bounded by $\mathcal{K}_d/T$. Hence

$$\mathcal{E}_{\mathrm{bias},T}^d \le \frac{2\mathcal{K}_d}{T}.$$

Thus, the spectral regimes of Section 3, in particular (6), also control of $\mathcal{E}_{\mathrm{bias},T}^d$ uniformly in $d$.

It remains to bound the initialization and score terms. We work in a perturbative regime in which the mixture parameters (weights, means, and covariances) are allowed to vary. This captures mode reweighting, mean shifts, and covariance distortions while preserving multimodality.

**Assumption 4.2.** *The approximate score is the score of a misspecified mixture along the same annealing path:*

$$\widetilde{s}_t^d(x) = \nabla\log\widetilde{\rho}_t^d(x),$$

*where $\widetilde{\rho}_t^d$ is obtained by annealing a perturbed mixture with the same smoothing covariance as in (2), namely*

$$\widetilde{\rho}_t^d = \widetilde{\rho}_\star^d * \mathcal{N}\Big(0,\ \tfrac{T-t}{T}\,C^d\Big), \qquad \widetilde{\rho}_\star^d := \sum_{i\in I}\widetilde{w}_i\,\mathcal{N}(\widetilde{m}_i^d, \widetilde{\Sigma}_i^d).$$

*The perturbed parameters are*

$$\widetilde{m}_i^d = m_i^d + \Delta m_i^d, \qquad \widetilde{\Sigma}_i^d = \Sigma_i^d + \Delta\Sigma_i^d, \qquad \widetilde{w}_i = w_i + \Delta w_i,$$

*with $(\widetilde{w}_i)_{i\in I}$ such that $\widetilde{w}_i > 0$ and $\sum_{i\in I}\widetilde{w}_i = 1$.*

For the calculations that follow, we continue to work in the diagonal setting introduced in Section 2. In particular, for each $d$ and each $i \in I$, we assume

$$\Delta\Sigma_i^d = \mathrm{Diag}(\Delta\sigma_{i1}, \ldots, \Delta\sigma_{id}), \tag{7}$$

so that

$$\widetilde{\Sigma}_i^d = \mathrm{Diag}(\sigma_{i1} + \Delta\sigma_{i1}, \ldots, \sigma_{id} + \Delta\sigma_{id}),$$

with $\sigma_{ij} + \Delta\sigma_{ij} > 0$.

## 4.1. Initialization Mismatch $\mathcal{E}_{\mathrm{init}}^d$

We begin with the initialization contribution. At $t = 0$, the exact and perturbed annealed laws have densities

$$\rho_0^d(x) = \sum_{i\in I}w_i\varphi_{i,0}^d(x), \qquad \widetilde{\rho}_0^d(x) = \sum_{i\in I}\widetilde{w}_i\widetilde{\varphi}_{i,0}^d(x),$$

where $\varphi_{i,0}^d$ and $\widetilde{\varphi}_{i,0}^d$ denote the Gaussian densities associated with the following laws, evaluated at $t = 0$:

$$\mathcal{N}(m_i^d, \Sigma_i^d + \kappa_t C^d), \qquad \mathcal{N}(\widetilde{m}_i^d, \widetilde{\Sigma}_i^d + \kappa_t C^d).$$

Thus $\mathcal{E}_{\mathrm{init}}^d$ measures the discrepancy between two Gaussian mixtures with perturbed weights, means, and covariances. We control it through the following bound.

**Proposition 4.3.** *We have the upper bound*

$$\mathcal{E}_{\text{init}}^d \;\leq\; \mathrm{KL}(w \,\|\, \widetilde{w}) + \sum_{i \in I} w_i \, \mathrm{KL}\big(\varphi_{i,0}^d \,\|\, \widetilde{\varphi}_{i,0}^d\big), \quad (8)$$

*where*

$$\mathrm{KL}(w \,\|\, \widetilde{w}) = \sum_{i \in I} w_i \log\left(\frac{w_i}{\widetilde{w}_i}\right). \quad (9)$$

*Moreover, combining Assumption 4.2 with* (7) *yields the explicit expression*

$$\mathrm{KL}\big(\varphi_{i,0}^d \,\|\, \widetilde{\varphi}_{i,0}^d\big) = \frac{1}{2} \sum_{j=1}^d \left[ \frac{(\Delta m_{ij})^2}{\sigma_{ij} + \Delta\sigma_{ij} + \lambda_j} \right. \\ \left. + \log\left(1 + \frac{\Delta\sigma_{ij}}{\sigma_{ij} + \lambda_j}\right) - \frac{\Delta\sigma_{ij}}{\sigma_{ij} + \Delta\sigma_{ij} + \lambda_j} \right]. \quad (10)$$

The bound (8) separates the effect of weight perturbations from the within-component Gaussian mismatch:

- The weight term (9) can be controlled, for example, by assuming $\widetilde{w}_i = w_i(1 + \eta_i)$, with $\eta_i \in (-\eta, \eta)$ for some $\eta \in (0, 1)$.

- Controlling (10) uniformly in $d$ instead amounts to requiring that the coordinatewise contributions be summable as $d \to \infty$. Using $\log(1 + r) \leq r$ for $r > -1$, it is enough to impose

$$\sup_{i \in I} \sum_{j \geq 1} \frac{(\Delta m_{ij})^2}{\sigma_{ij} + \Delta\sigma_{ij} + \lambda_j} < \infty,$$

$$\sup_{i \in I} \sum_{j \geq 1} \frac{(\Delta\sigma_{ij})^2}{(\sigma_{ij} + \lambda_j)(\sigma_{ij} + \Delta\sigma_{ij} + \lambda_j)} < \infty.$$

### 4.2. Score Mismatch $\mathcal{E}_{\text{score},T}^d$

We now turn to the score contribution, given by the $\Gamma^d$-weighted energy

$$\mathcal{E}_{\text{score},T}^d := \frac{1}{2} \int_0^T \mathbb{E}_{\rho_t^d} \big\| (\Gamma^d)^{1/2} \epsilon_t^d(X) \big\|^2 \, dt.$$

In the Gaussian-mixture setting, the score mismatch admits the decomposition

$$\epsilon_t^d(x) = \sum_{i \in I} p_{i,t}^d(x) \big( S_{i,t}^d(x) - \widetilde{S}_{i,t}^d(x) \big) \\ + \sum_{i \in I} \big( p_{i,t}^d(x) - \widetilde{p}_{i,t}^d(x) \big) \widetilde{S}_{i,t}^d(x),$$

where

$$p_{i,t}^d(x) = \frac{w_i \varphi_{i,t}^d(x)}{\rho_t^d(x)}, \qquad \widetilde{p}_{i,t}^d(x) = \frac{\widetilde{w}_i \widetilde{\varphi}_{i,t}^d(x)}{\widetilde{\rho}_t^d(x)}$$

are the exact and perturbed responsibilities, and

$$S_{i,t}^d(x) := -\big(\Sigma_i^d + \kappa_t C^d\big)^{-1}(x - m_i^d),$$

$$\widetilde{S}_{i,t}^d(x) := -\big(\widetilde{\Sigma}_i^d + \kappa_t C^d\big)^{-1}(x - \widetilde{m}_i^d).$$

This decomposition separates component-score errors from errors in the mixture responsibilities. The following proposition bounds both contributions.

**Proposition 4.4.** *For each $t \in [0, T]$, we have*

$$\mathbb{E}_{\rho_t^d} \Big\| \sum_{i \in I} p_{i,t}^d(X) \, (\Gamma^d)^{1/2} \big(S_{i,t}^d - \widetilde{S}_{i,t}^d\big)(X) \Big\|^2 \;\leq\; \mathsf{B}_{\text{comp}}^d(t), \quad (11)$$

*with*

$$\mathsf{B}_{\text{comp}}^d(t) := 2 \sum_{i \in I} w_i \sum_{j=1}^d \gamma_j \frac{(\Delta m_{ij})^2 + \frac{(\Delta\sigma_{ij})^2}{\sigma_{ij} + \kappa_t \lambda_j}}{\big(\sigma_{ij} + \Delta\sigma_{ij} + \kappa_t \lambda_j\big)^2}.$$

*Moreover, with $\mathsf{B}_{\text{resp}}^d(t)$ defined in* (64)*, we have*

$$\mathbb{E}_{\rho_t^d} \Big\| \sum_{i \in I} \big( p_{i,t}^d - \widetilde{p}_{i,t}^d \big)(X) \, (\Gamma^d)^{1/2} \widetilde{S}_{i,t}^d(X) \Big\|^2 \;\leq\; \mathsf{B}_{\text{resp}}^d(t). \quad (12)$$

*Consequently,*

$$\mathcal{E}_{\text{score},T}^d \;\leq\; \int_0^T \Big( \mathsf{B}_{\text{comp}}^d(t) + \mathsf{B}_{\text{resp}}^d(t) \Big) \, dt.$$

The proof is given in Appendix B.3: the component term is derived in (17), while the responsibility term is bounded in Proposition B.1. For the former, uniform-in-$d$ sufficient conditions are read directly from the explicit expression of $\mathsf{B}_{\text{comp}}^d(t)$; for the latter, the argument is more involved, and explicit sufficient conditions are collected in Proposition B.7.

- The term $\mathsf{B}_{\text{comp}}^d(t)$ depends on $(\Delta m_{ij}, \Delta\sigma_{ij})$ only through $\gamma_j$-weighted coordinatewise contributions. A sufficient condition for dimension-uniform control is

$$\sup_{i \in I} \sum_{j \geq 1} \gamma_j \frac{(\Delta m_{ij})^2 + \frac{(\Delta\sigma_{ij})^2}{\sigma_{ij}}}{(\sigma_{ij} + \Delta\sigma_{ij})^2} < \infty.$$

- The term $\mathsf{B}_{\text{resp}}^d(t)$ is more delicate because it couples component perturbations through the Gaussian-mixture responsibilities. The bound has two parts. The first controls how much the responsibilities change, through $\mathcal{D}(t, d)$ and $\mathcal{R}(t, d)$ in (24)–(25); these quantities are controlled by Proposition B.5. Apart from the low-mode and positivity assumptions stated there, it is enough to assume that, for some $J$ and $\rho \in (0, 1/9)$,

$$\frac{|\Delta\sigma_{ij}|}{\sigma_{ij}} \leq \rho, \qquad \frac{|\Delta m_{ij}|}{\sqrt{\sigma_{ij}}} \leq \rho,$$

for all $i \in I$ and all $j \geq J$, together with

$$\sup_{i \in I} \sum_{j \geq J} \left( \frac{(\Delta \sigma_{ij})^2}{\sigma_{ij}^2} + \frac{(\Delta m_{ij})^2}{\sigma_{ij}} \right) < \infty.$$

The second part controls the pairwise variation of the perturbed component scores,

$$\widetilde{S}_{i,t}^d - \widetilde{S}_{\ell,t}^d,$$

weighted coordinate by coordinate by $(\gamma_j)$. This is the role of $\mathcal{S}_{\text{pair}}(t,d)$ in (36), controlled by Proposition B.6. A simple set of sufficient conditions is

$$\sup_{i,\ell \in I} \sum_{j \geq 1} \gamma_j \frac{\left( (\sigma_{\ell j} - \sigma_{ij}) + (\Delta \sigma_{\ell j} - \Delta \sigma_{ij}) \right)^2}{(\sigma_{ij} + \Delta \sigma_{ij})^2}$$
$$\times \frac{\sigma_{ij} + \lambda_j}{(\sigma_{\ell j} + \Delta \sigma_{\ell j})^2} < \infty,$$

together with

$$\sup_{i,\ell \in I} \sum_{j \geq 1} \gamma_j \left( \frac{(\Delta m_{ij})^2}{(\sigma_{ij} + \Delta \sigma_{ij})^2} \right.$$
$$\left. + \frac{(m_{\ell j} - m_{ij} + \Delta m_{\ell j})^2}{(\sigma_{\ell j} + \Delta \sigma_{\ell j})^2} \right) < \infty.$$

These conditions highlight the importance of preconditioning ALD in the perturbative regime. They make concrete the trade-off anticipated at the end of Section 3: in Proposition 4.1, larger preconditioner coefficients reduce the annealing-induced bias through $\mathcal{E}_{\text{bias},T}^d$, whereas sufficient spectral decay is needed to control the accumulation of tail-coordinate perturbations in $\mathcal{E}_{\text{score},T}^d$. In simple regimes, the same trade-off also suggests concrete choices of $(\gamma_j)$. For instance, if the tail has common covariances $\sigma_{ij} = \sigma_j$, the perturbations are covariance-only with $\Delta m_{ij} = 0$ and $\Delta \sigma_{ij} = \delta_j$, the mean separation is confined to finitely many low modes, and $|\delta_j| \ll \sigma_j$, then the responsibility contribution is controlled, while the annealing-bias and component-score contributions scale as $\lambda_j^2/(\gamma_j \sigma_j)$ and $\gamma_j \delta_j^2/\sigma_j^3$, respectively. Balancing them gives

$$\gamma_j \asymp \lambda_j \sigma_j / |\delta_j|,$$

with both contributions of order $\lambda_j |\delta_j|/\sigma_j^2$; hence the sufficient conditions reduce to $\sum_{j \geq 1} \lambda_j |\delta_j|/\sigma_j^2 < \infty$.

## 5. Numerical Experiments

We now illustrate the dimension-uniform stability of ALD predicted by the theory. We consider successive finite-dimensional truncations of an infinite-dimensional Gaussian-mixture target and compare spectral choices of the smoothing covariance and preconditioner that either satisfy or violate the conditions above. The infinite-dimensional target

and spectral sequences are fixed, while the ALD runs are performed on their finite-dimensional truncations. Specifically, we estimate the empirical KL divergence between the target $\rho_\star^d$ and the ALD output law $\rho_T^{\text{ALD},d}$ as the dimension increases. The experiments have two purposes: to highlight the impact of spectral design choices intrinsic to the infinite-dimensional setting, and to illustrate the role of the preconditioner in ensuring stability under score mismatch. Implementation details are provided in Appendix D.

### 5.1. Annealing-Induced Bias vs. Dimension

We consider a two-component infinite-dimensional Gaussian-mixture target with weights $(0.75, 0.25)$, separated means $m_2 - m_1 = 10\, e_1$, and covariances $\Sigma_1 = 1.2 \cdot \Sigma$ and $\Sigma_2 = 2 \cdot \Sigma$, where $\Sigma = \text{Diag}(j^{-1.25})_{j \geq 1}$. We study its truncations $\rho_\star^d$ for $d \in \{1, 5, \ldots, 65\}$. We work in the idealized setting of Section 3, using the exact score along the annealing path and initializing from the smoothed law $\rho_0^d = \rho_\star^d * \mathcal{N}(0, C^d)$. We fix a common time horizon $T$, run ALD for each truncation at dimension $d$, and report the empirical estimate of the annealing-induced bias, $\text{KL}(\rho_\star^d \| \rho_T^{\text{ALD},d})$, the main source of error in this exact-score, exact-initialization experiment. We compare two regimes. In the first, we follow (6) and choose the infinite-dimensional spectra $\gamma_j = j^{-1.5}$ and $\lambda_j = 40 \cdot j^{-2.7}$, $j \geq 1$; these give the truncated preconditioner $\Gamma^d = \text{Diag}(j^{-1.5})_{j=1}^d$ and smoothing covariance $C^d = \text{Diag}(40 \cdot j^{-2.7})_{j=1}^d$. The factor 40 sets the smoothing scale. In the second, we keep the same target $\rho_\star^d$ but use the flat-spectrum choice, namely $\Gamma^d = I$ and $C^d = 40I$. Figure 3 contrasts the two behaviors on a logarithmic scale: under the prescribed spectral design, the $\text{KL}(\rho_\star^d \| \rho_T^{\text{ALD},d})$ remains uniformly small as $d$ increases, whereas under the flat-spectrum choice it grows with $d$. This illustrates that (6) reflects a genuine high-dimensional stability constraint rather than a proof artifact.

### 5.2. The Importance of Preconditioning

We now introduce controlled perturbations in both the initialization and the score. To isolate the role that our theory attributes to the preconditioner in achieving dimension-uniformity, we use the same initialization for both preconditioners: a mixture with incorrect weights $(0.1, 0.9)$ but with the same component means and covariances as the target. Thus, the resulting initialization mismatch is uniformly controlled in $d$; relative to the target mixture it is only a weight mismatch, while relative to the smoothed initial law the additional covariance mismatch is summable. The ALD drift, in contrast, is computed under a misspecified mixture in which the component covariances are perturbed.

Specifically, in Figure 4 we consider an infinite-dimensional bimodal Gaussian mixture target with weights $(0.75, 0.25)$,

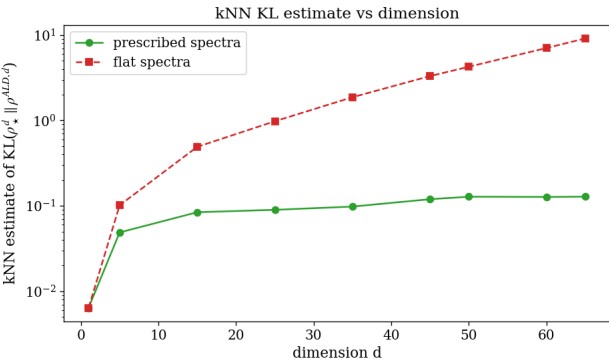

*Figure 3.* Annealing-induced bias across dimensions. Empirical $\mathrm{KL}(\rho_\star^d \| \rho_T^{\mathrm{ALD},d})$ as a function of the truncation dimension $d$, shown on a logarithmic $y$-axis. The green curve corresponds to the prescribed spectral design $\Gamma^d = \mathrm{Diag}(j^{-1.5})_{j=1}^d$ and $C^d = \mathrm{Diag}(40 \cdot j^{-2.7})_{j=1}^d$, while the red curve corresponds to the flat-spectrum choice $\Gamma^d = I$ and $C^d = 40I$. The KL is estimated via $k$NN with $k = 20$; robustness to $k$ is reported in Appendix D.

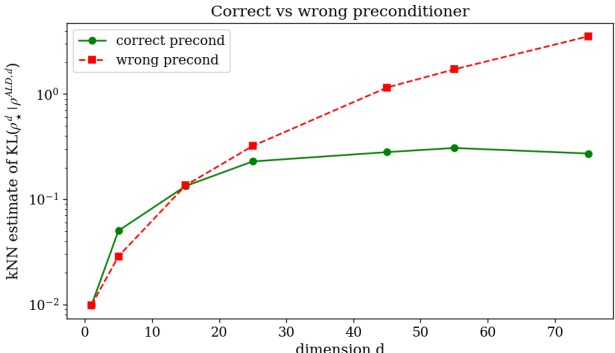

*Figure 4.* Preconditioning under score perturbations. Empirical $\mathrm{KL}(\rho_\star^d \| \rho_T^{\mathrm{ALD},d})$ as a function of the truncation dimension $d$, shown on a logarithmic $y$-axis. The ALD drift is computed using a covariance-perturbed score with $\Delta\sigma_{ij} = 0.05\, j^{-3}$. The green curve corresponds to ALD runs with preconditioner $\Gamma^d = \mathrm{Diag}(j^{-3.5})_{j=1}^d$, satisfying the sufficient perturbative conditions, while the red curve uses $\Gamma^d = \mathrm{Diag}(j^{-1})_{j=1}^d$, which violates the component-score summability condition. The KL is estimated via $k$NN with $k = 20$; robustness to $k$ is reported in Appendix D.

identical covariances $\Sigma_1 = \Sigma_2 = \mathrm{Diag}(j^{-2})_{j\geq 1}$, and separated means $m_2 - m_1 = 10e_1$. We use the smoothing covariance $C = \mathrm{Diag}(j^{-4})_{j\geq 1}$ and study its truncations at dimensions $d \in \{1, 5, \ldots, 75\}$. Following the perturbative setting of Section 4.2, and the sufficient conditions detailed in Appendix B.3, we introduce a covariance-only score perturbation, with $\Delta m_{ij} = 0$ and $\Delta\sigma_{ij} = 0.05\, j^{-3}$, so that the misspecified score uses component variances $\widetilde{\sigma}_{ij} = j^{-2} + 0.05\, j^{-3}$. We then compare two preconditioners through their finite-dimensional truncations: one satisfying the set of sufficient conditions,

$$\Gamma = \mathrm{Diag}(j^{-3.5})_{j\geq 1},$$

and one violating the component-score condition while satisfying the remaining conditions,

$$\Gamma = \mathrm{Diag}(j^{-1})_{j\geq 1}.$$

The contrast is again evident: with the admissible preconditioner, the error remains uniformly controlled as $d$ grows, whereas with the non-admissible one it increases with dimension. This supports the role predicted by the theory: under score mismatch, sufficient decay of the preconditioner is important for dimension-robust stability.

## 6. Discussion and Future Work

We studied the robustness of continuous-time annealed Langevin dynamics under successive finite-dimensional approximations of an infinite-dimensional multimodal target. In a Gaussian-mixture setting, we proved dimension-uniform control of the sampling error in Kullback–Leibler divergence under explicit conditions linking the component means and the spectra of the component covariances, the annealing smoothing covariance, and the preconditioner.

The analysis highlights the role of preconditioning in achieving dimension-uniformity. In the exact-score setting, the preconditioner enters the annealing-bias bound through its interaction with the smoothing covariance. In the perturbative regime, where initialization and score errors are present, sufficient spectral decay controls the accumulation of errors along the tail. A follow-up discretization analysis of the same ALD dynamics (Baldassari et al., 2026a) shows that this continuous-time picture must be complemented by stability considerations in discrete time: in particular, Euler–Maruyama can impose additional high-frequency constraints, absent from the continuous-time theory, that limit the applicability of ALD to more realistic settings.

As in related infinite-dimensional analyses (Baldassari et al., 2023; Pidstrigach et al., 2024; Baldassari et al., 2026b), we work under structural assumptions, most notably that the mixture covariances, smoothing covariance, and preconditioner are co-diagonalizable. This makes the dimension-uniform conditions explicit in terms of coordinatewise summability bounds. Nevertheless, the model should not be viewed as simplistic: at each fixed truncation level, mixtures with diagonal covariances can still represent a broad class of multimodal distributions, especially when the number of components is allowed to vary, or even be countable.

Several questions remain open, including posterior sampling in Bayesian nonlinear inverse problems, where classical Langevin is known to struggle (Bohr & Nickl, 2024; Nickl, 2023). Developing an ALD analogue of the theory in (Baldassari et al., 2026b) in this setting is a natural next step, and one we are actively pursuing.

## Acknowledgments

JG was supported by Agence de l'Innovation de Défense (AID) via the Centre Interdisciplinaire d'Études pour la Défense et la Sécurité (CIEDS) (project 2021–PRODIPO). KS was supported by the Air Force Office of Scientific Research under grant FA9550-22-1-0176 and the National Science Foundation under grant DMS-2308389. MVdH acknowledges support from the Department of Energy under grant DE-SC0020345, Oxy, the corporate members of the Geo-Mathematical Imaging Group at Rice University, and the Simons Foundation under the MATH+X program.

## Impact Statement

This paper presents work whose goal is to advance the field of machine learning. There are many potential societal consequences of our work, none of which we feel must be specifically highlighted here.

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

## A. Proofs of Section 3

Before proving Theorem 3.1, we recall a standard lemma. For completeness, we include the proof below.

**Lemma A.1.** *Let*

$$\begin{cases} dX_t^d = a_t(X_t^d)\, dt + \sqrt{2\Gamma^d}\, dW_t^d, & X_0^d \sim \mu^d, \\ dY_t^d = b_t(Y_t^d)\, dt + \sqrt{2\Gamma^d}\, dW_t^d, & Y_0^d \sim \nu^d, \end{cases}$$

*and denote by $P_{X^d}$ and $P_{Y^d}$ the path laws of $(X_t^d)_{t\in[0,T]}$ and $(Y_t^d)_{t\in[0,T]}$. Then*

$$\mathrm{KL}(P_{X^d} \,||\, P_{Y^d}) = \mathrm{KL}(\mu^d \,||\, \nu^d) + \frac{1}{4}\, \mathbb{E}_{X^d \sim P_{X^d}} \int_0^T \|a_t(X_t^d) - b_t(X_t^d)\|^2_{(\Gamma^d)^{-1}}\, dt,$$

*where $\|u\|^2_{(\Gamma^d)^{-1}} := \langle u, (\Gamma^d)^{-1} u \rangle$.*

*Proof.* In dimension $d$, Girsanov formula gives

$$\frac{dP_{X^d}}{dP_{Y^d}} = \frac{\mu^d(X_0^d)}{\nu^d(X_0^d)} \exp\left\{ \int_0^T \left\langle (2\Gamma^d)^{-1}\big(a_t(X_t^d) - b_t(X_t^d)\big),\, dX_t^d \right\rangle \right.$$
$$\left. - \frac{1}{2}\int_0^T \Big( \|(2\Gamma^d)^{-1/2} a_t(X_t^d)\|^2 - \|(2\Gamma^d)^{-1/2} b_t(X_t^d)\|^2 \Big)\, dt \right\}.$$

Taking the logarithm yields

$$\log \frac{dP_{X^d}}{dP_{Y^d}} = \log \frac{\mu^d(X_0^d)}{\nu^d(X_0^d)} + \int_0^T \left\langle (2\Gamma^d)^{-1}\big(a_t(X_t^d) - b_t(X_t^d)\big),\, dX_t^d \right\rangle$$
$$- \frac{1}{2}\int_0^T \Big( \|(2\Gamma^d)^{-1/2} a_t(X_t^d)\|^2 - \|(2\Gamma^d)^{-1/2} b_t(X_t^d)\|^2 \Big)\, dt.$$

Using the SDE for $dX_t^d$, we have

$$\left\langle (2\Gamma^d)^{-1}\big(a_t - b_t\big)(X_t^d),\, dX_t^d \right\rangle = \left\langle (2\Gamma^d)^{-1/2}\big(a_t - b_t\big)(X_t^d),\, (2\Gamma^d)^{-1/2} a_t(X_t^d) \right\rangle dt$$
$$+ \left\langle (2\Gamma^d)^{-1/2}\big(a_t - b_t\big)(X_t^d),\, dW_t^d \right\rangle.$$

The stochastic integral has mean zero under $P_{X^d}$. Since $\mathrm{KL}(P_{X^d}||P_{Y^d}) = \mathbb{E}_{P_{X^d}}\left[ \log \frac{dP_{X^d}}{dP_{Y^d}} \right]$, taking expectations gives

$$\mathrm{KL}(P_{X^d}||P_{Y^d}) = \mathrm{KL}(\mu^d||\nu^d) + \mathbb{E}_{P_{X^d}} \int_0^T \left\langle (2\Gamma^d)^{-1/2}\big(a_t - b_t\big)(X_t^d),\, (2\Gamma^d)^{-1/2} a_t(X_t^d) \right\rangle dt$$
$$- \frac{1}{2}\, \mathbb{E}_{P_{X^d}} \int_0^T \Big( \|(2\Gamma^d)^{-1/2} a_t(X_t^d)\|^2 - \|(2\Gamma^d)^{-1/2} b_t(X_t^d)\|^2 \Big)\, dt.$$

By the identity

$$\langle u - v,\, u \rangle - \tfrac{1}{2}\big( \|u\|^2 - \|v\|^2 \big) = \tfrac{1}{2}\, \|u - v\|^2,$$

applied to $u = (2\Gamma^d)^{-1/2} a_t(X_t^d)$ and $v = (2\Gamma^d)^{-1/2} b_t(X_t^d)$, we obtain

$$\mathrm{KL}(P_{X^d}||P_{Y^d}) = \mathrm{KL}(\mu^d||\nu^d) + \frac{1}{4}\, \mathbb{E}_{P_{X^d}} \int_0^T \|(\Gamma^d)^{-1/2}(a_t - b_t)(X_t^d)\|^2\, dt.$$

$\square$

We are now ready to prove Theorem 3.1. The proof builds on ideas from Appendix A of (Guo et al., 2025), adapted to our infinite-dimensional setting.

### A.1. Proof of Theorem 3.1

Let $\mathbb{Q}^d$ be the path measure on $C([0,T];\mathbb{R}^d)$ of the ALD diffusion (3) initialized at $X_0^d \sim \rho_0^d$, and let $\pi_T(\omega) = \omega(T)$ be the evaluation map. For any path measure $\mathbb{P}$ on $C([0,T];\mathbb{R}^d)$, the data-processing inequality yields

$$\mathrm{KL}\big((\pi_T)_\#\mathbb{P} \,\|\, (\pi_T)_\#\mathbb{Q}^d\big) \;\leq\; \mathrm{KL}(\mathbb{P} \,\|\, \mathbb{Q}^d).$$

We apply this with a reference path measure $\mathbb{P}^{v,d}$ constructed as follows. Let $\mathbb{P}^{v,d}$ denote the path law of

$$dY_t^d = \Gamma^d\big(\nabla \log \rho_t^d + v_t^d\big)(Y_t^d)\, dt + \sqrt{2\Gamma^d}\, dW_t^d, \qquad Y_0^d \sim \rho_0^d,$$

where the annealing path is

$$\rho_t^d = \rho_\star^d * \mathcal{N}\Big(0, \frac{T-t}{T} C^d\Big), \qquad t \in [0,T].$$

We choose $v^d = (v_t^d)_{t \in [0,T]}$ so that $\mathrm{Law}_{\mathbb{P}^{v,d}}(Y_t^d) = \rho_t^d$ for all $t \in [0,T]$. Since the diffusion matrix is constant ($2\Gamma^d$), the corresponding Fokker–Planck equation reduces to

$$\partial_t \rho_t^d = -\nabla \cdot (\rho_t^d \,\Gamma^d v_t^d). \tag{13}$$

On the other hand, the Gaussian-smoothing path satisfies the backward heat equation

$$\partial_t \rho_t^d = -\frac{1}{2T} \nabla \cdot \big(C^d \nabla \rho_t^d\big), \tag{14}$$

because differentiating with respect to the covariance parameter introduces a factor $1/2$, and $\frac{d}{dt}\big(\frac{T-t}{T}\big) = -\frac{1}{T}$. Comparing (13) and (14), an admissible choice is

$$\Gamma^d v_t^d = \frac{1}{2T} C^d \nabla \log \rho_t^d, \qquad \text{i.e.} \qquad v_t^d = \frac{1}{2T}(\Gamma^d)^{-1} C^d \nabla \log \rho_t^d.$$

With this choice, $(\pi_T)_\#\mathbb{P}^{v,d} = \rho_T^d = \rho_\star^d$. Since $(\pi_T)_\#\mathbb{Q}^d = \rho_T^{\mathrm{ALD},d}$ by definition, we have

$$\mathrm{KL}\big(\rho_\star^d \,\|\, \rho_T^{\mathrm{ALD},d}\big) \;\leq\; \mathrm{KL}(\mathbb{P}^{v,d} \,\|\, \mathbb{Q}^d). \tag{15}$$

It therefore remains to obtain an explicit upper bound on the path-space divergence $\mathrm{KL}(\mathbb{P}^{v,d} \,\|\, \mathbb{Q}^d)$.

By Lemma A.1, we have

$$\mathrm{KL}(\mathbb{P}^{v,d} \,\|\, \mathbb{Q}^d) = \frac{1}{4}\, \mathbb{E}_{\mathbb{P}^{v,d}} \int_0^T \|\Gamma^d v_t^d(Y_t^d)\|_{(\Gamma^d)^{-1}}^2 \, dt = \frac{1}{4} \int_0^T \int \langle v_t^d, \Gamma^d v_t^d \rangle \, d\rho_t^d \, dt$$

$$= \frac{1}{16T^2} \int_0^T \int \big\langle \nabla \log \rho_t^d,\, A^d \nabla \log \rho_t^d \big\rangle \, d\rho_t^d \, dt, \tag{16}$$

where $A^d := C^d (\Gamma^d)^{-1} C^d$.

Fix $t \in [0,T]$. Since $\rho_t^d$ is a Gaussian mixture,

$$\rho_t^d = \sum_{i \in I} w_i \, \mathcal{N}\Big(m_i^d, \Sigma_i^d + \frac{T-t}{T} C^d\Big), \qquad \rho_t^d(x) = \sum_{i \in I} w_i \, \varphi_{i,t}(x),$$

and with responsibilities $p_{i,t}^d(x) := \frac{w_i \varphi_{i,t}(x)}{\rho_t^d(x)}$ we obtain

$$\nabla \log \rho_t^d(x) = -\sum_{i \in I} p_{i,t}^d(x) \Big(\Sigma_i^d + \frac{T-t}{T} C^d\Big)^{-1} (x - m_i^d).$$

Since $z \mapsto z^\top A^d z$ is convex ($A^d \succeq 0$), Jensen's inequality yields

$$\big\langle \nabla \log \rho_t^d(x), A^d \nabla \log \rho_t^d(x) \big\rangle \leq \sum_{i \in I} p_{i,t}^d(x)\, (x - m_i^d)^\top \Big(\Sigma_i^d + \frac{T-t}{T} C^d\Big)^{-1} A^d \Big(\Sigma_i^d + \frac{T-t}{T} C^d\Big)^{-1} (x - m_i^d).$$

Integrating against $\rho_t^d(x)\, dx$ and using $\rho_t^d p_{i,t} = w_i \varphi_{i,t}$ gives

$$\int \langle \nabla \log \rho_t^d, A^d \nabla \log \rho_t^d \rangle \, d\rho_t^d \leq \sum_{i \in I} w_i \operatorname{Tr}\Big( A^d \Big( \Sigma_i^d + \frac{T-t}{T} C^d \Big)^{-1} \Big),$$

where we used that, in the diagonal setting, all matrices commute and $\operatorname{Cov}(\varphi_{i,t}) = \Sigma_i^d + \frac{T-t}{T} C^d$. Plugging this into (16) yields

$$\operatorname{KL}(\mathbb{P}^{v,d} \,\|\, \mathbb{Q}^d) \leq \frac{1}{16T^2} \sum_{i \in I} w_i \int_0^T \operatorname{Tr}\Big( A^d \Big( \Sigma_i^d + \frac{T-t}{T} C^d \Big)^{-1} \Big) \, dt.$$

Changing variables $s := \frac{T-t}{T} \in [0,1]$ (so $dt = -T\, ds$) gives

$$\int_0^T \operatorname{Tr}\Big( A^d \Big( \Sigma_i^d + \frac{T-t}{T} C^d \Big)^{-1} \Big) \, dt = T \int_0^1 \operatorname{Tr}\Big( A^d (\Sigma_i^d + s C^d)^{-1} \Big) \, ds.$$

Since $A^d$, $C^d$, and $\Sigma_i^d$ are diagonal with eigenvalues $\lambda_j^2 / \gamma_j$, $\lambda_j$, and $\sigma_{ij}$, respectively, we have

$$\operatorname{Tr}\big( A^d (\Sigma_i^d + s C^d)^{-1} \big) = \sum_{j=1}^d \frac{\lambda_j^2}{\gamma_j (\sigma_{ij} + s \lambda_j)}.$$

Therefore,

$$\operatorname{KL}(\mathbb{P}^{v,d} \,\|\, \mathbb{Q}^d) \leq \frac{1}{16T} \sum_{i \in I} w_i \sum_{j=1}^d \frac{\lambda_j^2}{\gamma_j} \int_0^1 \frac{ds}{\sigma_{ij} + s \lambda_j} = \frac{1}{16T} \sum_{i \in I} w_i \sum_{j=1}^d \frac{\lambda_j}{\gamma_j} \log\Big( 1 + \frac{\lambda_j}{\sigma_{ij}} \Big).$$

Combining this estimate with (15) gives

$$\operatorname{KL}\big( \rho_\star^d \,\|\, \rho_T^{\mathrm{ALD},d} \big) \leq \frac{1}{16T} \sum_{i \in I} w_i \sum_{j=1}^d \frac{\lambda_j}{\gamma_j} \log\Big( 1 + \frac{\lambda_j}{\sigma_{ij}} \Big).$$

Now fix $\epsilon > 0$. Choosing

$$T = \frac{1}{16\epsilon} \sum_{i \in I} w_i \sum_{j=1}^d \frac{\lambda_j}{\gamma_j} \log\Big( 1 + \frac{\lambda_j}{\sigma_{ij}} \Big)$$

yields

$$\operatorname{KL}\big( \rho_\star^d \,\|\, \rho_T^{\mathrm{ALD},d} \big) \leq \epsilon.$$

## B. Proofs of Section 4

### B.1. Proof of Proposition 4.1

As in Subsection A.1, we consider the reference SDE

$$dY_t^d = \Gamma^d \big( \nabla \log \rho_t^d + v_t^d \big) (Y_t^d) dt + \sqrt{2\Gamma^d} dW_t^d, \qquad Y_0^d \sim \rho_0^d,$$

with path measure $\mathbb{P}^{v,d}$ and $v^d = (v_t^d)_{t \in [0,T]}$ so that $\operatorname{Law}_{\mathbb{P}^{v,d}}(Y_t^d) = \rho_t^d$, and compare it to the approximate SDE

$$d\widetilde{X}_t^d = \Gamma^d \widetilde{s}_t^d (\widetilde{X}_t^d) dt + \sqrt{2\Gamma^d} d\widetilde{W}_t^d, \qquad \widetilde{X}_0^d \sim \widetilde{\rho}_0^d,$$

with path measure $\widetilde{\mathbb{Q}}^d$. Recall that $\epsilon_t^d(x) := \nabla \log \rho_t^d(x) - \widetilde{s}_t^d(x)$.

By Lemma A.1,

$$\operatorname{KL}(\mathbb{P}^{v,d} \,\|\, \widetilde{\mathbb{Q}}^d) = \operatorname{KL}(\rho_0^d \,\|\, \widetilde{\rho}_0^d) + \frac{1}{4} \int_0^T \int \big\| (\Gamma^d)^{1/2} \big( v_t^d + \epsilon_t^d \big)(x) \big\|^2 \, d\rho_t^d(x) \, dt.$$

Using $\|a + b\|^2 \leq 2\|a\|^2 + 2\|b\|^2$, we obtain

$$\mathrm{KL}(\mathbb{P}^{v,d} \,\|\, \widetilde{\mathbb{Q}}^d) \leq \mathrm{KL}(\rho_0^d \,\|\, \widetilde{\rho}_0^d) + \frac{1}{2} \int_0^T \int \left\|(\Gamma^d)^{1/2} v_t^d(x)\right\|^2 d\rho_t^d(x)\, dt + \frac{1}{2} \int_0^T \int \left\|(\Gamma^d)^{1/2} \epsilon_t^d(x)\right\|^2 d\rho_t^d(x)\, dt.$$

Since $v^d$ is chosen so that $\mathrm{Law}_{\mathbb{P}^{v,d}}(Y_t^d) = \rho_t^d$, the data-processing inequality yields $\mathrm{KL}(\rho_\star^d \,\|\, \widetilde{\rho}_T^{\mathrm{ALD},d}) \leq \mathrm{KL}(\mathbb{P}^{v,d} \,\|\, \widetilde{\mathbb{Q}}^d)$.
Hence

$$\mathrm{KL}(\rho_\star^d \,\|\, \widetilde{\rho}_T^{\mathrm{ALD},d}) \leq \mathrm{KL}(\rho_0^d \,\|\, \widetilde{\rho}_0^d) + \frac{1}{2} \int_0^T \int \left\|(\Gamma^d)^{1/2} v_t^d(x)\right\|^2 d\rho_t^d(x)\, dt + \frac{1}{2} \int_0^T \int \left\|(\Gamma^d)^{1/2} \epsilon_t^d(x)\right\|^2 d\rho_t^d(x)\, dt.$$

Taking the infimum over all such $v^d$, which satisfy $\partial_t \rho_t^d = -\nabla \cdot (\rho_t^d \Gamma^d v_t^d)$, yields

$$\mathrm{KL}(\rho_\star^d \,\|\, \widetilde{\rho}_T^{\mathrm{ALD},d}) \leq \mathcal{E}_{\mathrm{init}}^d + \mathcal{E}_{\mathrm{bias},T}^d + \mathcal{E}_{\mathrm{score},T}^d,$$

where

$$\mathcal{E}_{\mathrm{init}}^d := \mathrm{KL}(\rho_0^d \,\|\, \widetilde{\rho}_0^d), \qquad \mathcal{E}_{\mathrm{score},T}^d := \frac{1}{2} \int_0^T \int \left\|(\Gamma^d)^{1/2} \epsilon_t^d(x)\right\|^2 d\rho_t^d(x)\, dt,$$

and

$$\mathcal{E}_{\mathrm{bias},T}^d := \frac{1}{2} \inf_{v^d:\, \partial_t \rho_t^d = -\nabla \cdot (\rho_t^d \Gamma^d v_t^d)} \int_0^T \int \left\|(\Gamma^d)^{1/2} v_t^d(x)\right\|^2 d\rho_t^d(x)\, dt.$$

### B.2. Proof of Proposition 4.3

At $t = 0$,

$$\rho_0^d(x) = \sum_{i \in I} w_i \varphi_{i,0}^d(x), \qquad \widetilde{\rho}_0^d(x) = \sum_{i \in I} \widetilde{w}_i \widetilde{\varphi}_{i,0}^d(x),$$

where

$$\varphi_{i,0}^d = \mathcal{N}(m_i^d, \Sigma_i^d + C^d), \qquad \widetilde{\varphi}_{i,0}^d = \mathcal{N}(\widetilde{m}_i^d, \widetilde{\Sigma}_i^d + C^d).$$

We have

$$\mathrm{KL}(\rho_0^d \| \widetilde{\rho}_0^d) = \sum_{i \in I} w_i \mathbb{E}_{X \sim \varphi_{i,0}^d} \left[\log \frac{\sum_{\ell \in I} w_\ell \varphi_{\ell,0}^d(X)}{\sum_{\ell \in I} \widetilde{w}_\ell \widetilde{\varphi}_{\ell,0}^d(X)}\right].$$

Consider the joint densities

$$\phi^d(i, x) = w_i \varphi_{i,0}^d(x), \qquad \widetilde{\phi}^d(i, x) = \widetilde{w}_i \widetilde{\varphi}_{i,0}^d(x).$$

Then by data-processing inequality,

$$\mathrm{KL}(\rho_0^d \| \widetilde{\rho}_0^d) \leq \mathrm{KL}(\phi^d \| \widetilde{\phi}^d) = \mathrm{KL}(w \| \widetilde{w}) + \sum_{i \in I} w_i \mathrm{KL}(\varphi_{i,0}^d \| \widetilde{\varphi}_{i,0}^d),$$

where

$$\mathrm{KL}(w \| \widetilde{w}) = \sum_{i \in I} w_i \log \frac{w_i}{\widetilde{w}_i},$$

and each Gaussian-Gaussian KL is explicit:

$$
\begin{aligned}
&\mathrm{KL}\big(\varphi_{i,0}^d \,\big\|\, \widetilde{\varphi}_{i,0}^d\big) \\
&= \frac{1}{2} \Big[ \big\|(\widetilde{\Sigma}_i^d + C^d)^{-1/2}(\widetilde{m}_i^d - m_i^d)\big\|_2^2 \\
&+ \mathrm{Tr}\Big((\widetilde{\Sigma}_i^d + C^d)^{-1/2}(\Sigma_i^d + C^d)(\widetilde{\Sigma}_i^d + C^d)^{-1/2} - I - \log\big((\widetilde{\Sigma}_i^d + C^d)^{-1/2}(\Sigma_i^d + C^d)(\widetilde{\Sigma}_i^d + C^d)^{-1/2}\big)\Big)\Big].
\end{aligned}
$$

Under the diagonal assumption,

$$\mathrm{KL}\big(\varphi_{i,0}^d \,\big\|\, \widetilde{\varphi}_{i,0}^d\big) = \frac{1}{2} \sum_{j=1}^d \left[\frac{\sigma_{ij} + \lambda_j}{\sigma_{ij} + \Delta\sigma_{ij} + \lambda_j} - 1 + \frac{(\Delta m_{ij})^2}{\sigma_{ij} + \Delta\sigma_{ij} + \lambda_j} + \log\left(\frac{\sigma_{ij} + \Delta\sigma_{ij} + \lambda_j}{\sigma_{ij} + \lambda_j}\right)\right].$$

## B.3. Proof of Proposition 4.4

We work with the two annealed mixtures $\rho_t^d$ and $\widetilde{\rho}_t^d$ introduced in the main text. Set

$$\Sigma_{i,t}^d := \Sigma_i^d + \kappa_t C^d, \qquad \widetilde{\Sigma}_{i,t}^d := \widetilde{\Sigma}_i^d + \kappa_t C^d,$$

where $\kappa_t := (T - t)/T \in [0, 1]$. By the diagonal assumptions from Sections 2–4,

$$C^d = \operatorname{Diag}(\lambda_j)_{j=1}^d, \quad \Gamma^d = \operatorname{Diag}(\gamma_j)_{j=1}^d, \quad \Sigma_i^d = \operatorname{Diag}(\sigma_{ij})_{j=1}^d, \quad \widetilde{\Sigma}_i^d = \operatorname{Diag}(\sigma_{ij} + \Delta\sigma_{ij})_{j=1}^d,$$

and therefore

$$\Sigma_{i,t}^d = \operatorname{Diag}(\sigma_{ij} + \kappa_t \lambda_j)_{j=1}^d, \qquad \widetilde{\Sigma}_{i,t}^d = \operatorname{Diag}(\sigma_{ij} + \Delta\sigma_{ij} + \kappa_t \lambda_j)_{j=1}^d.$$

We denote the (true and perturbed) component densities by

$$\varphi_{i,t}^d := \mathcal{N}(m_i^d, \Sigma_{i,t}^d), \qquad \widetilde{\varphi}_{i,t}^d := \mathcal{N}(\widetilde{m}_i^d, \widetilde{\Sigma}_{i,t}^d),$$

with $\widetilde{m}_i^d = m_i^d + \Delta m_i^d$, so that

$$\rho_t^d(x) = \sum_{i \in I} w_i\, \varphi_{i,t}^d(x), \qquad \widetilde{\rho}_t^d(x) = \sum_{i \in I} \widetilde{w}_i\, \widetilde{\varphi}_{i,t}^d(x), \qquad \sum_{i \in I} w_i = \sum_{i \in I} \widetilde{w}_i = 1.$$

The responsibilities and component scores are

$$p_{i,t}^d(x) := \frac{w_i \varphi_{i,t}^d(x)}{\rho_t^d(x)}, \qquad \widetilde{p}_{i,t}^d(x) := \frac{\widetilde{w}_i \widetilde{\varphi}_{i,t}^d(x)}{\widetilde{\rho}_t^d(x)},$$

and

$$S_{i,t}^d(x) := -(\Sigma_{i,t}^d)^{-1}(x - m_i^d), \qquad \widetilde{S}_{i,t}^d(x) := -(\widetilde{\Sigma}_{i,t}^d)^{-1}(x - \widetilde{m}_i^d).$$

With this notation,

$$\nabla \log \rho_t^d(x) = \sum_{i \in I} p_{i,t}^d(x)\, S_{i,t}^d(x), \qquad \nabla \log \widetilde{\rho}_t^d(x) = \sum_{i \in I} \widetilde{p}_{i,t}^d(x)\, \widetilde{S}_{i,t}^d(x),$$

and the score error $\epsilon_t^d := \nabla \log \rho_t^d - \nabla \log \widetilde{\rho}_t^d$ decomposes as

$$\epsilon_t^d(x) = \underbrace{\sum_{i \in I} p_{i,t}^d(x)\big(S_{i,t}^d(x) - \widetilde{S}_{i,t}^d(x)\big)}_{=:I_1(x,t)} + \underbrace{\sum_{i \in I} \big(p_{i,t}^d(x) - \widetilde{p}_{i,t}^d(x)\big)\, \widetilde{S}_{i,t}^d(x)}_{=:I_2(x,t)}.$$

Hence proving Proposition 4.4 boils down to bounding the following two terms:

(1) A component-score mismatch $\mathbb{E}_{\rho_t^d}\big[\|(\Gamma^d)^{1/2} I_1(X, t)\|^2\big]$.

(2) A responsibility mismatch $\mathbb{E}_{\rho_t^d}\big[\|(\Gamma^d)^{1/2} I_2(X, t)\|^2\big]$.

In what follows, we use the same notation for finite and countable mixtures. For countable $I$, each estimate can be justified by first replacing the mixtures with their normalized restrictions to finite sets $I_N \uparrow I$ and then letting $N \to \infty$. In the component-score mismatch, this normalization only changes the finite mixture weights and disappears in the limit; in the responsibility mismatch, the same normalization cancels in the ratios defining the responsibilities. Thus the finite-mixture estimates pass to the countable case whenever the quantities appearing in the resulting upper bounds are finite.

### (1) COMPONENT-SCORE MISMATCH: PROOF OF (11)

Define $\Delta S_{i,t}^d := \widetilde{S}_{i,t}^d - S_{i,t}^d$. A direct expansion gives

$$\Delta S_{i,t}^d(x) = \big((\Sigma_{i,t}^d)^{-1} - (\widetilde{\Sigma}_{i,t}^d)^{-1}\big)(x - m_i^d) + (\widetilde{\Sigma}_{i,t}^d)^{-1}(\widetilde{m}_i^d - m_i^d).$$

Let $\Delta m_i^d := \widetilde{m}_i^d - m_i^d$ and write $\Delta m_{ij}$ for its coordinates.

Using Jensen's inequality for the convex quadratic form $z \mapsto \|(\Gamma^d)^{1/2} z\|^2$ and $\sum_i p_{i,t}^d(x) = 1$,

$$\left\|(\Gamma^d)^{1/2} I_1(x,t)\right\|^2 = \left\|(\Gamma^d)^{1/2} \sum_{i \in I} p_{i,t}^d(x) \Delta S_{i,t}^d(x)\right\|^2 \leq \sum_{i \in I} p_{i,t}^d(x) \left\|(\Gamma^d)^{1/2} \Delta S_{i,t}^d(x)\right\|^2.$$

Integrating and using $\rho_t^d(x) p_{i,t}^d(x) = w_i \varphi_{i,t}^d(x)$ yields

$$\mathbb{E}_{\rho_t^d}\left[\|(\Gamma^d)^{1/2} I_1(X,t)\|^2\right] \leq \sum_{i \in I} w_i \int_{\mathbb{R}^d} \left\|(\Gamma^d)^{1/2} \Delta S_{i,t}^d(x)\right\|^2 \varphi_{i,t}^d(x) \, dx. \tag{17}$$

Using $(a+b)^2 \leq 2a^2 + 2b^2$,

$$\left\|(\Gamma^d)^{1/2} \Delta S_{i,t}^d(x)\right\|^2 \leq 2 \left\|(\Gamma^d)^{1/2}\left((\Sigma_{i,t}^d)^{-1} - (\widetilde{\Sigma}_{i,t}^d)^{-1}\right)(x - m_i^d)\right\|^2 + 2\left\|(\Gamma^d)^{1/2}(\widetilde{\Sigma}_{i,t}^d)^{-1} \Delta m_i^d\right\|^2.$$

Since everything is diagonal, the second term is explicit:

$$\left\|(\Gamma^d)^{1/2}(\widetilde{\Sigma}_{i,t}^d)^{-1} \Delta m_i^d\right\|^2 = \sum_{j=1}^d \gamma_j \frac{(\Delta m_{ij})^2}{(\sigma_{ij} + \Delta\sigma_{ij} + \kappa_t \lambda_j)^2}.$$

For the first term, set

$$b_{ij}(t) := \frac{1}{\sigma_{ij} + \kappa_t \lambda_j} - \frac{1}{\sigma_{ij} + \Delta\sigma_{ij} + \kappa_t \lambda_j} = \frac{\Delta\sigma_{ij}}{(\sigma_{ij} + \kappa_t \lambda_j)(\sigma_{ij} + \Delta\sigma_{ij} + \kappa_t \lambda_j)},$$

so that $(\Sigma_{i,t}^d)^{-1} - (\widetilde{\Sigma}_{i,t}^d)^{-1}$ has $j$-th diagonal entry $b_{ij}(t)$. Then

$$\left\|(\Gamma^d)^{1/2}\left((\Sigma_{i,t}^d)^{-1} - (\widetilde{\Sigma}_{i,t}^d)^{-1}\right)(x - m_i^d)\right\|^2 = \sum_{j=1}^d \gamma_j \, b_{ij}(t)^2 \, (x_j - m_{ij})^2.$$

Since $\mathbb{E}[(X_j - m_{ij})^2] = \sigma_{ij} + \kappa_t \lambda_j$ with $X \sim \mathcal{N}(m_i^d, \Sigma_{i,t}^d)$, we have

$$\int \left\|(\Gamma^d)^{1/2}\left((\Sigma_{i,t}^d)^{-1} - (\widetilde{\Sigma}_{i,t}^d)^{-1}\right)(x - m_i^d)\right\|^2 \varphi_{i,t}^d(x) \, dx = \sum_{j=1}^d \gamma_j \, b_{ij}(t)^2 \, (\sigma_{ij} + \kappa_t \lambda_j)$$

$$= \sum_{j=1}^d \gamma_j \frac{(\Delta\sigma_{ij})^2}{(\sigma_{ij} + \kappa_t \lambda_j)(\sigma_{ij} + \Delta\sigma_{ij} + \kappa_t \lambda_j)^2}.$$

Plugging everything into (17) gives

$$\mathbb{E}_{\rho_t^d}\left[\|(\Gamma^d)^{1/2} I_1(X,t)\|^2\right] \leq 2 \sum_{i \in I} w_i \sum_{j=1}^d \gamma_j \left[\frac{(\Delta m_{ij})^2}{(\sigma_{ij} + \Delta\sigma_{ij} + \kappa_t \lambda_j)^2} + \frac{(\Delta\sigma_{ij})^2}{(\sigma_{ij} + \kappa_t \lambda_j)(\sigma_{ij} + \Delta\sigma_{ij} + \kappa_t \lambda_j)^2}\right],$$

which is exactly (11) with

$$\mathsf{B}_{\mathrm{comp}}^d(t) := 2 \sum_{i \in I} w_i \sum_{j=1}^d \gamma_j \frac{(\Delta m_{ij})^2 + \frac{(\Delta\sigma_{ij})^2}{\sigma_{ij} + \kappa_t \lambda_j}}{\left(\sigma_{ij} + \Delta\sigma_{ij} + \kappa_t \lambda_j\right)^2}.$$

This proves the first part of Proposition 4.4.

**(2) RESPONSIBILITY MISMATCH: PROOF OF** (12)

In this part it is convenient to abbreviate

$$\rho := \rho_t^d, \qquad \widetilde{\rho} := \widetilde{\rho}_t^d, \qquad \varphi_i := \varphi_{i,t}^d, \qquad \widetilde{\varphi}_i := \widetilde{\varphi}_{i,t}^d,$$

$$p_i := \frac{w_i \varphi_i}{\rho}, \qquad \widetilde{p}_i := \frac{\widetilde{w}_i \widetilde{\varphi}_i}{\widetilde{\rho}}, \qquad \widetilde{S}_i := \widetilde{S}_{i,t}^d(\cdot).$$

To control the responsibility mismatch

$$\mathbb{E}_{\rho_t^d}\left[\left\|(\Gamma^d)^{1/2} I_2(X,t)\right\|^2\right], \qquad \text{with} \quad I_2(x,t) := \sum_{i \in I} \left(p_{i,t}^d(x) - \widetilde{p}_{i,t}^d(x)\right) \widetilde{S}_{i,t}^d(x),$$

requires three steps:

1. First, we rewrite the perturbed responsibilities in terms of the exact ones and express $I_2$ in centered form around the barycenter

$$\overline{S}(x) := \sum_{i \in I} p_i(x) \widetilde{S}_i(x).$$

   This leads to the pointwise decomposition

$$\left\|(\Gamma^d)^{1/2} I_2(x,t)\right\|^2 \leq A_{\mathrm{resp}}(x,t)\, V_{\mathrm{score}}(x,t),$$

   given in (18). The factor $A_{\mathrm{resp}}$ measures the change in the responsibilities, while $V_{\mathrm{score}}$ measures the corresponding variation of the perturbed component scores.

2. Second, we bound these two factors separately. The responsibility factor is controlled by the quantities $\mathcal{D}(t,d)$ and $\mathcal{R}(t,d)$, defined in (24) and (25); this gives (27). The score-variation factor is controlled through pairwise differences $\widetilde{S}_i - \widetilde{S}_\ell$, leading to the quantity $\mathcal{S}_{\mathrm{pair}}(t,d)$ defined in (36); this gives (37). Combining the two estimates yields Proposition B.1.

3. Third, we derive sufficient conditions ensuring that $\mathcal{D}(t,d)$, $\mathcal{R}(t,d)$, and $\mathcal{S}_{\mathrm{pair}}(t,d)$ are bounded uniformly in $d \geq 1$ and $t \in [0,T]$. The quantities $\mathcal{D}$ and $\mathcal{R}$ are controlled by expanding the Gaussian density-ratio moments; the resulting conditions are given in Proposition B.5. The pairwise variation of the perturbed component scores is controlled in Proposition B.6. The final sufficient conditions, written directly in terms of

$$(\sigma_{ij}, m_{ij}, \Delta\sigma_{ij}, \Delta m_{ij}, \lambda_j, \gamma_j),$$

   are summarized in Proposition B.7.

**Decomposition of the responsibility mismatch.** We first bound $\left\|(\Gamma^d)^{1/2} I_2(x,t)\right\|^2$ by a product of two quantities: one measuring the change in the responsibilities and one measuring the corresponding variation of the perturbed component scores.

Define

$$\Delta w_i := \widetilde{w}_i - w_i, \qquad \beta_i := \frac{\widetilde{w}_i}{w_i}, \qquad r_i(x) := \frac{\widetilde{\varphi}_i(x)}{\varphi_i(x)}, \qquad \beta_* := \inf_{i \in I} \beta_i.$$

The perturbed responsibilities can be written in terms of the exact responsibilities. Indeed, since

$$\widetilde{w}_i \widetilde{\varphi}_i(x) = w_i \varphi_i(x)\, \beta_i r_i(x) = \rho(x)\, p_i(x)\, \beta_i r_i(x),$$

we have

$$\widetilde{\rho}(x) = \sum_{k \in I} \widetilde{w}_k \widetilde{\varphi}_k(x) = \rho(x) \sum_{k \in I} p_k(x) \beta_k r_k(x).$$

Therefore,

$$\widetilde{p}_i(x) = \frac{\widetilde{w}_i \widetilde{\varphi}_i(x)}{\widetilde{\rho}(x)} = \frac{\rho(x) p_i(x) \beta_i r_i(x)}{\rho(x) \sum_{k \in I} p_k(x) \beta_k r_k(x)} = p_i(x) \frac{\beta_i r_i(x)}{\sum_{k \in I} p_k(x) \beta_k r_k(x)}.$$

Hence

$$\widetilde{p}_i - p_i = p_i \left( \frac{\beta_i r_i}{\sum_{k \in I} p_k \beta_k r_k} - 1 \right).$$

Consequently,

$$I_2(x,t) = \sum_{i \in I} p_i(x)\left(1 - \frac{\beta_i r_i(x)}{\sum_{k \in I} p_k(x)\beta_k r_k(x)}\right)\widetilde{S}_i(x).$$

Introduce

$$z_i(x) := \frac{\beta_i r_i(x)}{\sum_{k \in I} p_k(x)\beta_k r_k(x)}.$$

Then

$$I_2(x,t) = -\sum_{i \in I} p_i(x)(z_i(x) - 1)\widetilde{S}_i(x).$$

Now define the $p_i$-weighted barycenter

$$\overline{S}(x) := \sum_{i \in I} p_i(x)\widetilde{S}_i(x).$$

Since

$$\sum_{i \in I} p_i(x)(z_i(x) - 1) = 0,$$

we may subtract this barycenter and obtain the centered representation

$$I_2(x,t) = \sum_{i \in I} p_i(x)(z_i(x) - 1)\big(\widetilde{S}_i(x) - \overline{S}(x)\big).$$

Weighted Cauchy–Schwarz gives

$$\left\|(\Gamma^d)^{1/2} I_2(x,t)\right\|^2 \le A_{\mathrm{resp}}(x,t)\, V_{\mathrm{score}}(x,t), \tag{18}$$

where

$$A_{\mathrm{resp}}(x,t) := \sum_{i \in I} p_i(x)\left(\frac{\beta_i r_i(x)}{\sum_{k \in I} p_k(x)\beta_k r_k(x)} - 1\right)^2,$$

and

$$V_{\mathrm{score}}(x,t) := \frac{1}{2}\sum_{i,\ell \in I} p_i(x)p_\ell(x)\left\|(\Gamma^d)^{1/2}\big(\widetilde{S}_i(x) - \widetilde{S}_\ell(x)\big)\right\|^2. \tag{19}$$

Here (19) comes from applying the weighted variance identity to

$$\sum_{i \in I} p_i(x)\left\|(\Gamma^d)^{1/2}\big(\widetilde{S}_i(x) - \overline{S}(x)\big)\right\|^2.$$

**Controlling the change in responsibilities.**  We control the factor $A_{\mathrm{resp}}$ in (18). This factor depends only on how the responsibilities of the perturbed annealed mixture $\widetilde{\rho}_t^d$ differ from those of the exact annealed mixture $\rho_t^d$. The estimate keeps the quantity $\beta_i r_i - 1$ explicit, so that this part of the bound vanishes when the weights and component densities are not perturbed. The goal of the calculation below is to bound

$$\mathbb{E}_\rho[A_{\mathrm{resp}}(X,t)^2],$$

in terms of the two quantities $\mathcal{D}(t,d)$ and $\mathcal{R}(t,d)$ defined in (24) and (25).

Set

$$\overline{z}(x) := \sum_{k \in I} p_k(x)\beta_k r_k(x).$$

Then

$$A_{\mathrm{resp}}(x,t) = \sum_{i \in I} p_i(x)\left(\frac{\beta_i r_i(x)}{\overline{z}(x)} - 1\right)^2 = \overline{z}(x)^{-2}\sum_{i \in I} p_i(x)\big(\beta_i r_i(x) - \overline{z}(x)\big)^2.$$

Using the weighted variance inequality

$$\sum_i p_i(u_i - \overline{u})^2 \leq \sum_i p_i(u_i - c)^2 \qquad \text{for every } c \in \mathbb{R},$$

with $u_i = \beta_i r_i$ and $c = 1$, we obtain

$$A_{\text{resp}}(x,t) \leq \frac{\sum_{i \in I} p_i(x)(\beta_i r_i(x) - 1)^2}{\overline{z}(x)^2}. \tag{20}$$

Assume $\beta_* > 0$. Since $\beta_i \geq \beta_*$ and $u \mapsto u^{-2}$ is convex on $(0, \infty)$,

$$\overline{z}(x)^{-2} \leq \sum_{k \in I} p_k(x)\beta_k^{-2} r_k(x)^{-2} \leq \beta_*^{-2} \sum_{k \in I} p_k(x)r_k(x)^{-2}. \tag{21}$$

Combining (20) and (21) gives

$$A_{\text{resp}}(x,t) \leq \beta_*^{-2} \left( \sum_{i \in I} p_i(x)(\beta_i r_i(x) - 1)^2 \right) \left( \sum_{k \in I} p_k(x)r_k(x)^{-2} \right). \tag{22}$$

Therefore, by Jensen's inequality applied separately to the two sums,

$$A_{\text{resp}}(x,t)^2 \leq \beta_*^{-4} \left( \sum_{i \in I} p_i(x)(\beta_i r_i(x) - 1)^4 \right) \left( \sum_{k \in I} p_k(x)r_k(x)^{-4} \right). \tag{23}$$

Introduce

$$\mathcal{D}(t,d) := \sum_{i \in I} w_i D_i(t,d), \qquad D_i(t,d) := \mathbb{E}_{\varphi_i}\left[ (\beta_i r_i(X) - 1)^8 \right], \tag{24}$$

and

$$\mathcal{R}(t,d) := \sum_{i \in I} w_i R_i(t,d), \qquad R_i(t,d) := \mathbb{E}_{\varphi_i}\left[ r_i(X)^{-8} \right]. \tag{25}$$

Set

$$X_{\text{resp}}(x) := \sum_{i \in I} p_i(x)(\beta_i r_i(x) - 1)^4, \qquad Y_{\text{resp}}(x) := \sum_{k \in I} p_k(x)r_k(x)^{-4}.$$

Then (23) reads

$$A_{\text{resp}}(x,t)^2 \leq \beta_*^{-4} X_{\text{resp}}(x)Y_{\text{resp}}(x).$$

Hence, by Cauchy–Schwarz in $L^2(\rho)$,

$$\mathbb{E}_\rho[A_{\text{resp}}(X,t)^2] \leq \beta_*^{-4}\mathbb{E}_\rho[X_{\text{resp}}(X)^2]^{1/2}\mathbb{E}_\rho[Y_{\text{resp}}(X)^2]^{1/2}. \tag{26}$$

Since the $p_i(x)$ are probability weights,

$$X_{\text{resp}}(x)^2 \leq \sum_{i \in I} p_i(x)(\beta_i r_i(x) - 1)^8, \qquad Y_{\text{resp}}(x)^2 \leq \sum_{k \in I} p_k(x)r_k(x)^{-8}.$$

Integrating and using $p_i \rho = w_i \varphi_i$, we obtain

$$\mathbb{E}_\rho[X_{\text{resp}}(X)^2] \leq \mathcal{D}(t,d), \qquad \mathbb{E}_\rho[Y_{\text{resp}}(X)^2] \leq \mathcal{R}(t,d).$$

Therefore (26) yields

$$\mathbb{E}_\rho[A_{\text{resp}}(X,t)^2] \leq \beta_*^{-4}\mathcal{D}(t,d)^{1/2}\mathcal{R}(t,d)^{1/2}. \tag{27}$$

**Controlling the pairwise variation of the perturbed component scores.** We now control the factor $V_{\text{score}}$ in (18). This factor contains the variation of the perturbed component scores that is multiplied by the responsibility error. The purpose of this part is to bound its $L^2(\rho)$ contribution by the pairwise quantity $\mathcal{S}_{\text{pair}}(t, d)$ defined in (36).

For each $\ell \in I$, define

$$G_{i\ell}(x,t) := \left\| (\Gamma^d)^{1/2}\big(\widetilde{S}_i(x) - \widetilde{S}_\ell(x)\big) \right\|^2, \qquad U_\ell(x,t) := \sum_{i \in I} p_i(x) G_{i\ell}(x,t).$$

The barycenter $\overline{S}(x)$ minimizes the weighted quadratic energy. Indeed, for every $c \in \mathbb{R}^d$,

$$\sum_{i \in I} p_i(x) \left\| (\Gamma^d)^{1/2}(\widetilde{S}_i(x) - c) \right\|^2 = V_{\text{score}}(x,t) + \left\| (\Gamma^d)^{1/2}(\overline{S}(x) - c) \right\|^2.$$

Taking $c = \widetilde{S}_\ell(x)$ gives

$$V_{\text{score}}(x,t) \leq U_\ell(x,t) \qquad \text{for every } \ell \in I.$$

Averaging this inequality with respect to the original mixture weights gives

$$V_{\text{score}}(x,t) \leq \overline{U}(x,t), \qquad \overline{U}(x,t) := \sum_{\ell \in I} w_\ell U_\ell(x,t). \tag{28}$$

Combining (18) and (28), we obtain

$$\left\| (\Gamma^d)^{1/2} I_2(x,t) \right\|^2 \leq A_{\text{resp}}(x,t)\, \overline{U}(x,t). \tag{29}$$

It remains to bound the $L^2(\rho)$ norm of $\overline{U}$. Since $(w_\ell)_{\ell \in I}$ is a probability vector, Jensen's inequality gives

$$\overline{U}(x,t)^2 \leq \sum_{\ell \in I} w_\ell U_\ell(x,t)^2.$$

Therefore

$$\mathbb{E}_\rho[\overline{U}(X,t)^2] \leq \sum_{\ell \in I} w_\ell \mathbb{E}_\rho[U_\ell(X,t)^2]. \tag{30}$$

For fixed $\ell$, using again $\sum_i p_i(x) = 1$, Jensen's inequality gives

$$U_\ell(x,t)^2 \leq \sum_{i \in I} p_i(x) G_{i\ell}(x,t)^2.$$

Integrating against $\rho$ and using $p_i \rho = w_i \varphi_i$, we obtain

$$\mathbb{E}_\rho[U_\ell(X,t)^2] \leq \sum_{i \in I} w_i \mathbb{E}_{\varphi_i}[G_{i\ell}(X,t)^2]. \tag{31}$$

Combining (30) and (31),

$$\mathbb{E}_\rho[\overline{U}(X,t)^2] \leq \sum_{i,\ell \in I} w_i w_\ell \mathbb{E}_{\varphi_i}[G_{i\ell}(X,t)^2]. \tag{32}$$

We now compute the last expectation explicitly. Define

$$v_{ij}(t) := \sigma_{ij} + \kappa_t \lambda_j, \qquad \widetilde{v}_{ij}(t) := \sigma_{ij} + \Delta\sigma_{ij} + \kappa_t \lambda_j, \qquad \widetilde{m}_{ij} := m_{ij} + \Delta m_{ij}.$$

Coordinatewise,

$$\widetilde{S}_{i,t,j}^d(x) - \widetilde{S}_{\ell,t,j}^d(x) = -a_{i\ell,j}(t) x_j + b_{i\ell,j}(t),$$

where

$$a_{i\ell,j}(t) := \frac{1}{\widetilde{v}_{ij}(t)} - \frac{1}{\widetilde{v}_{\ell j}(t)} = \frac{(\sigma_{\ell j} - \sigma_{ij}) + (\Delta\sigma_{\ell j} - \Delta\sigma_{ij})}{\widetilde{v}_{ij}(t)\widetilde{v}_{\ell j}(t)}, \tag{33}$$

and

$$b_{i\ell,j}(t) := \frac{\widetilde{m}_{ij}}{\widetilde{v}_{ij}(t)} - \frac{\widetilde{m}_{\ell j}}{\widetilde{v}_{\ell j}(t)}.$$

If $X \sim \varphi_i$, then $X_j \sim \mathcal{N}(m_{ij}, v_{ij}(t))$. Define

$$\mu_{i\ell,j}(t) := b_{i\ell,j}(t) - a_{i\ell,j}(t)m_{ij} = \frac{\Delta m_{ij}}{\widetilde{v}_{ij}(t)} - \frac{m_{\ell j} - m_{ij} + \Delta m_{\ell j}}{\widetilde{v}_{\ell j}(t)}. \tag{34}$$

Then

$$-a_{i\ell,j}(t)X_j + b_{i\ell,j}(t) = -a_{i\ell,j}(t)(X_j - m_{ij}) + \mu_{i\ell,j}(t),$$

and this Gaussian random variable has mean $\mu_{i\ell,j}(t)$ and variance $a_{i\ell,j}(t)^2 v_{ij}(t)$. Hence its fourth moment is

$$3a_{i\ell,j}(t)^4 v_{ij}(t)^2 + 6a_{i\ell,j}(t)^2 v_{ij}(t)\mu_{i\ell,j}(t)^2 + \mu_{i\ell,j}(t)^4.$$

Define

$$H_{i\ell,j}(t) := \left(3a_{i\ell,j}(t)^4 v_{ij}(t)^2 + 6a_{i\ell,j}(t)^2 v_{ij}(t)\mu_{i\ell,j}(t)^2 + \mu_{i\ell,j}(t)^4\right)^{1/2}. \tag{35}$$

Since

$$G_{i\ell}(X,t) = \sum_{j=1}^{d} \gamma_j \left(-a_{i\ell,j}(t)X_j + b_{i\ell,j}(t)\right)^2,$$

Minkowski's inequality in $L^2(\varphi_i)$ gives

$$\mathbb{E}_{\varphi_i}[G_{i\ell}(X,t)^2]^{1/2} \le \sum_{j=1}^{d} \gamma_j H_{i\ell,j}(t).$$

Introduce the pairwise score-variation factor

$$\mathcal{S}_{\mathrm{pair}}(t,d) := \sum_{i,\ell \in I} w_i w_\ell \left(\sum_{j=1}^{d} \gamma_j H_{i\ell,j}(t)\right)^2. \tag{36}$$

By (32),

$$\mathbb{E}_\rho[\overline{U}(X,t)^2]^{1/2} \le \mathcal{S}_{\mathrm{pair}}(t,d)^{1/2}. \tag{37}$$

**Responsibility-mismatch bound.** Combining the estimate for the change in responsibilities with the estimate for the pairwise variation of the perturbed component scores yields the following bound.

**Proposition B.1.** *Assume $\beta_* := \inf_{i \in I} \beta_i > 0$. Then, for every $d \ge 1$ and $t \in [0, T]$,*

$$\mathbb{E}_{\rho_t^d}\left[\left\|(\Gamma^d)^{1/2}I_2(X,t)\right\|^2\right] \le \beta_*^{-2}\mathcal{D}(t,d)^{1/4}\mathcal{R}(t,d)^{1/4}\mathcal{S}_{\mathrm{pair}}(t,d)^{1/2}. \tag{38}$$

*Moreover, if all perturbations vanish, that is,*

$$\Delta w_i = 0, \qquad \Delta\sigma_{ij} = 0, \qquad \Delta m_{ij} = 0 \qquad \text{for all } i, j,$$

*then $\beta_i = 1$ and $r_i \equiv 1$. Hence*

$$\mathcal{D}(t,d) = 0,$$

*and the right-hand side of (38) is equal to zero.*

*Proof.* By (29) and Cauchy–Schwarz in $L^2(\rho)$,

$$\mathbb{E}_{\rho_t^d}\left[\left\|(\Gamma^d)^{1/2}I_2(X,t)\right\|^2\right] \le \mathbb{E}_\rho[A_{\mathrm{resp}}(X,t)^2]^{1/2}\mathbb{E}_\rho[\overline{U}(X,t)^2]^{1/2}.$$

Applying (27) and (37) gives (38).

If all perturbations vanish, then $\widetilde{w}_i = w_i$ and $\widetilde{\varphi}_i = \varphi_i$, hence $\beta_i = 1$ and $r_i \equiv 1$. Therefore

$$D_i(t,d) = \mathbb{E}_{\varphi_i}\left[(\beta_i r_i(X) - 1)^8\right] = 0 \qquad \text{for every } i,$$

and so $\mathcal{D}(t,d) = 0$. $\qquad\square$

The bound separates the two mechanisms that enter the responsibility mismatch. The quantities $\mathcal{D}(t,d)$ and $\mathcal{R}(t,d)$ control the change in the responsibilities through moments of

$$\frac{\widetilde{w}_i}{w_i}\frac{\widetilde{\varphi}_{i,t}^d}{\varphi_{i,t}^d} - 1 \quad \text{and} \quad \left(\frac{\widetilde{\varphi}_{i,t}^d}{\varphi_{i,t}^d}\right)^{-1}.$$

The quantity $\mathcal{S}_{\mathrm{pair}}(t,d)$ controls the pairwise variation of the perturbed component scores,

$$\widetilde{S}_{i,t}^d - \widetilde{S}_{\ell,t}^d,$$

weighted coordinate by coordinate by the preconditioner spectrum $(\gamma_j)$.

We will use the following consequence of Proposition B.1.

**Corollary B.2.** *Assume* $\beta_* := \inf_{i \in I} \beta_i > 0$ *and*

$$\sup_{t \in [0,T]} \sup_{d \geq 1} \mathcal{D}(t,d) < \infty, \qquad \sup_{t \in [0,T]} \sup_{d \geq 1} \mathcal{R}(t,d) < \infty, \tag{39}$$

*together with*

$$\sup_{t \in [0,T]} \sup_{d \geq 1} \mathcal{S}_{\mathrm{pair}}(t,d) < \infty.$$

*Then*

$$\sup_{t \in [0,T]} \sup_{d \geq 1} \mathbb{E}_{\rho_t^d}\left[\left\|(\Gamma^d)^{1/2} I_2(X,t)\right\|^2\right] < \infty.$$

**Controlling $\mathcal{D}(t,d)$ and $\mathcal{R}(t,d)$.** We now turn to the two quantities that control the change in the responsibilities in Proposition B.1. Recall that

$$\mathcal{D}(t,d) = \sum_{i \in I} w_i D_i(t,d), \qquad D_i(t,d) = \mathbb{E}_{\varphi_i}\left[(\beta_i r_i(X) - 1)^8\right],$$

and

$$\mathcal{R}(t,d) = \sum_{i \in I} w_i R_i(t,d), \qquad R_i(t,d) = \mathbb{E}_{\varphi_i}\left[r_i(X)^{-8}\right].$$

Thus the problem is to control moments of ratios between perturbed and exact Gaussian component densities, uniformly in the truncation dimension $d \geq 1$ and in the annealing time $t \in [0,T]$. Since the component densities are diagonal Gaussians, these moments can be reduced to products of one-coordinate Gaussian integrals. The purpose of the next definitions is to make this reduction explicit.

For $a, b \in \mathbb{R}$, define

$$\Psi_{ik}^{(a,b)}(t,d) := \mathbb{E}_{\varphi_i}\left[r_i(X)^a r_k(X)^b\right]. \tag{40}$$

We write $\phi(x; m, v)$ for the one-dimensional Gaussian density with mean $m$ and variance $v > 0$:

$$\phi(x; m, v) = \frac{1}{\sqrt{2\pi v}} \exp\left(-\frac{(x-m)^2}{2v}\right).$$

Since $\varphi_i$ and $\widetilde{\varphi}_i$ are product Gaussian densities in the diagonal setting, the ratio moments in (40) factorize over the coordinates. To make this explicit, define the one-coordinate ratios

$$r_{i,j}(x) := \frac{\phi(x; \widetilde{m}_{ij}, \widetilde{v}_{ij}(t))}{\phi(x; m_{ij}, v_{ij}(t))}, \qquad r_{k,j}(x) := \frac{\phi(x; \widetilde{m}_{kj}, \widetilde{v}_{kj}(t))}{\phi(x; m_{kj}, v_{kj}(t))},$$

where

$$v_{ij}(t) = \sigma_{ij} + \kappa_t \lambda_j, \qquad \widetilde{v}_{ij}(t) = \sigma_{ij} + \Delta\sigma_{ij} + \kappa_t \lambda_j.$$

Then, for $x = (x_1, \ldots, x_d)$,

$$r_i(x) = \prod_{j=1}^d r_{i,j}(x_j), \qquad r_k(x) = \prod_{j=1}^d r_{k,j}(x_j).$$

Moreover, if $X \sim \varphi_i$, then the coordinates are independent and $X_j \sim \mathcal{N}(m_{ij}, v_{ij}(t))$. Hence

$$\Psi_{ik}^{(a,b)}(t,d) = \prod_{j=1}^{d} \psi_{ik,j}^{(a,b)}(t), \tag{41}$$

where

$$\psi_{ik,j}^{(a,b)}(t) := \int_{\mathbb{R}} r_{i,j}(x)^a r_{k,j}(x)^b \phi(x; m_{ij}, v_{ij}(t)) \, dx.$$

We now identify which of these moments are needed for $\mathcal{D}(t,d)$ and $\mathcal{R}(t,d)$. In both cases only diagonal moments are involved. By the binomial formula,

$$(\beta_i r_i - 1)^8 = \sum_{q=0}^{8} \binom{8}{q} (-1)^{8-q} \beta_i^q r_i^q,$$

and hence

$$D_i(t,d) = \sum_{q=0}^{8} \binom{8}{q} (-1)^{8-q} \beta_i^q \Psi_{ii}^{(q,0)}(t,d). \tag{42}$$

Similarly,

$$R_i(t,d) = \Psi_{ii}^{(0,-8)}(t,d). \tag{43}$$

Therefore, to bound $\mathcal{D}(t,d)$ and $\mathcal{R}(t,d)$, it is enough to control the one-coordinate factors $\psi_{ii,j}^{(q,0)}(t)$ for $q = 0, \ldots, 8$ and $\psi_{ii,j}^{(0,-8)}(t)$. The next lemma computes the more general quantity $\psi_{ik,j}^{(a,b)}(t)$, which will cover all these cases at once.

**Lemma B.3.** *Fix $a, b \in \mathbb{R}$, $i, k \in I$, $j \geq 1$, and $t \in [0, T]$. Let*

$$v_i := v_{ij}(t), \qquad v_k := v_{kj}(t), \qquad \widetilde{v}_i := \widetilde{v}_{ij}(t), \qquad \widetilde{v}_k := \widetilde{v}_{kj}(t),$$

*and*

$$m_i := m_{ij}, \qquad m_k := m_{kj}, \qquad \widetilde{m}_i := \widetilde{m}_{ij}, \qquad \widetilde{m}_k := \widetilde{m}_{kj}.$$

*Assume that all four variances are positive. Define*

$$A_{ik,j}^{(a,b)}(t) := \frac{1-a}{v_i} + \frac{a}{\widetilde{v}_i} + \frac{b}{\widetilde{v}_k} - \frac{b}{v_k},$$

$$B_{ik,j}^{(a,b)}(t) := \frac{(1-a)m_i}{v_i} + \frac{a\widetilde{m}_i}{\widetilde{v}_i} + \frac{b\widetilde{m}_k}{\widetilde{v}_k} - \frac{bm_k}{v_k},$$

*and*

$$C_{ik,j}^{(a,b)}(t) := \frac{(1-a)m_i^2}{v_i} + \frac{a\widetilde{m}_i^2}{\widetilde{v}_i} + \frac{b\widetilde{m}_k^2}{\widetilde{v}_k} - \frac{bm_k^2}{v_k}.$$

*If $A_{ik,j}^{(a,b)}(t) > 0$, then*

$$\psi_{ik,j}^{(a,b)}(t) = \left(\frac{v_i}{\widetilde{v}_i}\right)^{a/2} \left(\frac{v_k}{\widetilde{v}_k}\right)^{b/2} \frac{1}{\sqrt{v_i A_{ik,j}^{(a,b)}(t)}} \exp\left(\frac{\left(B_{ik,j}^{(a,b)}(t)\right)^2}{2A_{ik,j}^{(a,b)}(t)} - \frac{1}{2} C_{ik,j}^{(a,b)}(t)\right). \tag{44}$$

*Proof.* By definition,

$$\psi_{ik,j}^{(a,b)}(t) = \int_{\mathbb{R}} \left(\frac{\phi(x; \widetilde{m}_i, \widetilde{v}_i)}{\phi(x; m_i, v_i)}\right)^a \left(\frac{\phi(x; \widetilde{m}_k, \widetilde{v}_k)}{\phi(x; m_k, v_k)}\right)^b \phi(x; m_i, v_i) \, dx.$$

Using the explicit Gaussian density,

$$\frac{\phi(x; \widetilde{m}_i, \widetilde{v}_i)^a \phi(x; \widetilde{m}_k, \widetilde{v}_k)^b \phi(x; m_i, v_i)}{\phi(x; m_i, v_i)^a \phi(x; m_k, v_k)^b} = (2\pi)^{-1/2} \left(\frac{v_i}{\widetilde{v}_i}\right)^{a/2} \left(\frac{v_k}{\widetilde{v}_k}\right)^{b/2} v_i^{-1/2} \exp\left(-\frac{1}{2} Q(x)\right),$$

where

$$Q(x) = A_{ik,j}^{(a,b)}(t)x^2 - 2B_{ik,j}^{(a,b)}(t)x + C_{ik,j}^{(a,b)}(t).$$

Thus the integrability of this one-dimensional Gaussian-type integral is ensured by the positivity condition $A_{ik,j}^{(a,b)}(t) > 0$. Under this condition, completing the square gives

$$Q(x) = A_{ik,j}^{(a,b)}(t)\left(x - \frac{B_{ik,j}^{(a,b)}(t)}{A_{ik,j}^{(a,b)}(t)}\right)^2 - \frac{\left(B_{ik,j}^{(a,b)}(t)\right)^2}{A_{ik,j}^{(a,b)}(t)} + C_{ik,j}^{(a,b)}(t).$$

Therefore

$$\psi_{ik,j}^{(a,b)}(t) = (2\pi)^{-1/2}\left(\frac{v_i}{\widetilde{v}_i}\right)^{a/2}\left(\frac{v_k}{\widetilde{v}_k}\right)^{b/2} v_i^{-1/2}$$

$$\times \exp\left(\frac{\left(B_{ik,j}^{(a,b)}(t)\right)^2}{2A_{ik,j}^{(a,b)}(t)} - \frac{1}{2}C_{ik,j}^{(a,b)}(t)\right)$$

$$\times \int_{\mathbb{R}} \exp\left(-\frac{1}{2}A_{ik,j}^{(a,b)}(t)\left(x - \frac{B_{ik,j}^{(a,b)}(t)}{A_{ik,j}^{(a,b)}(t)}\right)^2\right)\,dx.$$

The Gaussian integral equals $\sqrt{2\pi/A_{ik,j}^{(a,b)}(t)}$, which gives (44). $\qquad\square$

We now specialize the lemma to the moments appearing in (42) and (43). The required exponents are

$$(a, b) = (q, 0), \qquad q = 0, 1, \ldots, 8,$$

and

$$(a, b) = (0, -8).$$

Thus the relevant positivity conditions are

$$A_{ii,j}^{(q,0)}(t) > 0 \qquad \text{for all } i, j, t, \quad q = 0, 1, \ldots, 8, \tag{45}$$

and

$$A_{ii,j}^{(0,-8)}(t) > 0 \qquad \text{for all } i, j, t. \tag{46}$$

In the diagonal setting, these positivity conditions can be written directly in terms of the covariance perturbations. Indeed, for $q = 0, \ldots, 8$,

$$A_{ii,j}^{(q,0)}(t) = \frac{v_{ij}(t) + (1 - q)\Delta\sigma_{ij}}{v_{ij}(t)\widetilde{v}_{ij}(t)}.$$

Thus, since $v_{ij}(t) > 0$ and $\widetilde{v}_{ij}(t) > 0$, the condition $A_{ii,j}^{(q,0)}(t) > 0$ is equivalent to

$$v_{ij}(t) + (1 - q)\Delta\sigma_{ij} > 0.$$

Similarly,

$$A_{ii,j}^{(0,-8)}(t) = \frac{v_{ij}(t) + 9\Delta\sigma_{ij}}{v_{ij}(t)\widetilde{v}_{ij}(t)},$$

so $A_{ii,j}^{(0,-8)}(t) > 0$ is equivalent to

$$v_{ij}(t) + 9\Delta\sigma_{ij} > 0.$$

Equivalently, with

$$\alpha_{ij}(t) := \frac{\Delta\sigma_{ij}}{v_{ij}(t)},$$

the positivity conditions are

$$1 + \alpha_{ij}(t) > 0, \qquad 1 + (1-q)\alpha_{ij}(t) > 0 \quad \text{for } q = 0, \ldots, 8,$$

and

$$1 + 9\alpha_{ij}(t) > 0.$$

Thus the required positivity can be enforced as follows: for the finitely many low modes, one assumes the inequalities directly, while on the tail they follow from

$$|\alpha_{ij}(t)| \leq \rho_{\text{tail}}, \qquad \rho_{\text{tail}} \in (0, 1/9).$$

The product formula (41) shows why it is natural to work with logarithms of the one-coordinate factors: uniform control of the products follows from uniform control of the corresponding sums of logarithms. For $i \in I$, $j \geq 1$, $t \in [0, T]$, and $q = 0, 1, \ldots, 8$, define

$$\Lambda_{ij}^{(q)}(t) := \log \psi_{ii,j}^{(q,0)}(t),$$

and also

$$\Lambda_{ij}^{(-8)}(t) := \log \psi_{ii,j}^{(0,-8)}(t).$$

By Lemma B.3, for $q = 0, 1, \ldots, 8$,

$$\Lambda_{ij}^{(q)}(t) = \frac{q}{2} \log \left( \frac{v_{ij}(t)}{\widetilde{v}_{ij}(t)} \right) - \frac{1}{2} \log \left( v_{ij}(t) A_{ii,j}^{(q,0)}(t) \right) + \frac{\left( B_{ii,j}^{(q,0)}(t) \right)^2}{2 A_{ii,j}^{(q,0)}(t)} - \frac{1}{2} C_{ii,j}^{(q,0)}(t), \tag{47}$$

whereas

$$\Lambda_{ij}^{(-8)}(t) = -4 \log \left( \frac{v_{ij}(t)}{\widetilde{v}_{ij}(t)} \right) - \frac{1}{2} \log \left( v_{ij}(t) A_{ii,j}^{(0,-8)}(t) \right) + \frac{\left( B_{ii,j}^{(0,-8)}(t) \right)^2}{2 A_{ii,j}^{(0,-8)}(t)} - \frac{1}{2} C_{ii,j}^{(0,-8)}(t). \tag{48}$$

The next corollary is the passage from one-coordinate control to uniform control of $\mathcal{D}(t, d)$ and $\mathcal{R}(t, d)$.

**Corollary B.4.** *Assume that*

$$\beta^* := \sup_{i \in I} \beta_i < \infty,$$

*that the positivity conditions (45) and (46) hold, and that, for every $q = 0, 1, \ldots, 8$,*

$$\sup_{i \in I} \sup_{t \in [0,T]} \sup_{d \geq 1} \sum_{j=1}^{d} \left| \Lambda_{ij}^{(q)}(t) \right| < \infty, \tag{49}$$

*and*

$$\sup_{i \in I} \sup_{t \in [0,T]} \sup_{d \geq 1} \sum_{j=1}^{d} \left| \Lambda_{ij}^{(-8)}(t) \right| < \infty. \tag{50}$$

*Then*

$$\sup_{t \in [0,T]} \sup_{d \geq 1} \mathcal{D}(t, d) < \infty, \qquad \sup_{t \in [0,T]} \sup_{d \geq 1} \mathcal{R}(t, d) < \infty.$$

*Proof.* For each $i \in I$, $t \in [0, T]$, and $d \geq 1$, define

$$L_i^{(q)}(t, d) := \sum_{j=1}^{d} \Lambda_{ij}^{(q)}(t), \qquad q = 0, 1, \ldots, 8,$$

and

$$L_i^{(-8)}(t, d) := \sum_{j=1}^{d} \Lambda_{ij}^{(-8)}(t).$$

By the factorization formula (41),

$$\Psi_{ii}^{(q,0)}(t,d) = \exp\left(L_i^{(q)}(t,d)\right), \qquad q = 0, 1, \ldots, 8,$$

and

$$R_i(t,d) = \Psi_{ii}^{(0,-8)}(t,d) = \exp\left(L_i^{(-8)}(t,d)\right).$$

The assumptions imply that, for each $q = 0, 1, \ldots, 8$, there exists $C_q < \infty$ such that

$$|L_i^{(q)}(t,d)| \leq C_q \qquad \text{for all } i, t, d.$$

Hence

$$\Psi_{ii}^{(q,0)}(t,d) \leq e^{C_q}.$$

Similarly, there exists $C_{-8} < \infty$ such that

$$R_i(t,d) \leq e^{C_{-8}} \qquad \text{for all } i, t, d.$$

Using (42),

$$|D_i(t,d)| \leq \sum_{q=0}^{8} \binom{8}{q} \beta_i^q \Psi_{ii}^{(q,0)}(t,d) \leq \sum_{q=0}^{8} \binom{8}{q} (\beta^*)^q e^{C_q}.$$

Therefore

$$\mathcal{D}(t,d) = \sum_{i \in I} w_i D_i(t,d) \leq \sum_{q=0}^{8} \binom{8}{q} (\beta^*)^q e^{C_q} < \infty,$$

uniformly in $t$ and $d$. Likewise,

$$\mathcal{R}(t,d) = \sum_{i \in I} w_i R_i(t,d) \leq e^{C_{-8}} \sum_{i \in I} w_i = e^{C_{-8}}.$$

$\square$

It remains to give conditions under which the sums in (49) and (50) are finite uniformly in $i$, $t$, and $d$. The previous formulas show that the relevant quantities depend on the covariance and mean perturbations through normalized ratios. Set

$$S := \{-8, 0, 1, \ldots, 8\}.$$

For $i \in I$, $j \geq 1$, and $t \in [0, T]$, define

$$\alpha_{ij}(t) := \frac{\Delta \sigma_{ij}}{v_{ij}(t)}, \qquad \zeta_{ij}(t) := \frac{\Delta m_{ij}}{\sqrt{v_{ij}(t)}}.$$

For $s \in S$, define

$$F_s(\alpha, \zeta) := -\frac{s-1}{2} \log(1+\alpha) - \frac{1}{2} \log\big(1 + (1-s)\alpha\big) + \frac{s(s-1)}{2} \frac{\zeta^2}{1 + (1-s)\alpha},$$

on the domain

$$\mathcal{U}_s := \left\{ (\alpha, \zeta) \in \mathbb{R}^2 : 1 + \alpha > 0, \quad 1 + (1-s)\alpha > 0 \right\}.$$

**Proposition B.5.** *For every* $q = 0, 1, \ldots, 8$,

$$\Lambda_{ij}^{(q)}(t) = F_q\big(\alpha_{ij}(t), \zeta_{ij}(t)\big), \tag{51}$$

*and*

$$\Lambda_{ij}^{(-8)}(t) = F_{-8}\big(\alpha_{ij}(t), \zeta_{ij}(t)\big). \tag{52}$$

*Assume that there exist* $J \in \mathbb{N}$, $\rho_{\text{tail}} \in (0, 1/9)$, *and* $C_{\text{tail}} < \infty$ *such that the following conditions hold.*

*For the low modes, the logarithmic factors are well defined and uniformly bounded:*

$$C_{\text{low}} := \sup_{i \in I} \sup_{t \in [0,T]} \sum_{j=1}^{J-1} \max_{s \in S} \left| F_s\big(\alpha_{ij}(t), \zeta_{ij}(t)\big) \right| < \infty.$$

*Here well defined means that*

$$\big(\alpha_{ij}(t), \zeta_{ij}(t)\big) \in \mathcal{U}_s \qquad \text{for all } i \in I, \quad j = 1, \ldots, J-1, \quad t \in [0,T], \quad s \in S.$$

*On the tail, the normalized perturbations are uniformly small:*

$$|\alpha_{ij}(t)| \le \rho_{\text{tail}}, \qquad |\zeta_{ij}(t)| \le \rho_{\text{tail}}, \qquad \text{for all } i \in I, \ j \ge J, \ t \in [0,T]. \tag{53}$$

*Finally, the normalized tail perturbations are uniformly summable:*

$$\sup_{i \in I} \sup_{t \in [0,T]} \sup_{d \ge 1} \sum_{j=J}^{d} \big(\alpha_{ij}(t)^2 + \zeta_{ij}(t)^2\big) \le C_{\text{tail}}.$$

*Then the positivity conditions* (45) *and* (46) *hold, and, for every* $q = 0, 1, \ldots, 8$,

$$\sup_{i \in I} \sup_{t \in [0,T]} \sup_{d \ge 1} \sum_{j=1}^{d} \big| \Lambda_{ij}^{(q)}(t) \big| < \infty,$$

*and also*

$$\sup_{i \in I} \sup_{t \in [0,T]} \sup_{d \ge 1} \sum_{j=1}^{d} \big| \Lambda_{ij}^{(-8)}(t) \big| < \infty.$$

*Consequently, if* $\sup_i \beta_i < \infty$, *then the conclusion of Corollary B.4 holds.*

*Proof.* Fix $i \in I$, $j \ge 1$, and $t \in [0,T]$. Write

$$v := v_{ij}(t), \qquad \eta := \Delta\sigma_{ij}, \qquad \delta := \Delta m_{ij}, \qquad \widetilde{v} := v + \eta.$$

We first treat $q = 0, 1, \ldots, 8$. In the diagonal case,

$$A_{ii,j}^{(q,0)}(t) = \frac{1-q}{v} + \frac{q}{\widetilde{v}} = \frac{v + (1-q)\eta}{v\widetilde{v}}.$$

Therefore

$$v A_{ii,j}^{(q,0)}(t) = \frac{v + (1-q)\eta}{\widetilde{v}} = \frac{1 + (1-q)\alpha_{ij}(t)}{1 + \alpha_{ij}(t)}.$$

Moreover,

$$B_{ii,j}^{(q,0)}(t) = \frac{(1-q)m_{ij}}{v} + \frac{q(m_{ij} + \delta)}{\widetilde{v}},$$

and

$$C_{ii,j}^{(q,0)}(t) = \frac{(1-q)m_{ij}^2}{v} + \frac{q(m_{ij} + \delta)^2}{\widetilde{v}}.$$

A direct simplification gives

$$\frac{\big(B_{ii,j}^{(q,0)}(t)\big)^2}{2 A_{ii,j}^{(q,0)}(t)} - \frac{1}{2} C_{ii,j}^{(q,0)}(t) = \frac{q(q-1)\delta^2}{2\big(v + (1-q)\eta\big)}.$$

Since $\delta^2 = v\zeta_{ij}(t)^2$ and

$$v + (1-q)\eta = v\big(1 + (1-q)\alpha_{ij}(t)\big),$$

we obtain

$$\frac{\left(B_{ii,j}^{(q,0)}(t)\right)^2}{2A_{ii,j}^{(q,0)}(t)} - \frac{1}{2}C_{ii,j}^{(q,0)}(t) = \frac{q(q-1)}{2}\frac{\zeta_{ij}(t)^2}{1+(1-q)\alpha_{ij}(t)}.$$

Substituting into (47) gives (51).

The case $s = -8$ is identical, using (48). In that case

$$A_{ii,j}^{(0,-8)}(t) = \frac{1}{v} - \frac{8}{\widetilde{v}} + \frac{8}{v} = \frac{v+9\eta}{v\widetilde{v}},$$

so

$$vA_{ii,j}^{(0,-8)}(t) = \frac{1+9\alpha_{ij}(t)}{1+\alpha_{ij}(t)}.$$

The same simplification gives

$$\frac{\left(B_{ii,j}^{(0,-8)}(t)\right)^2}{2A_{ii,j}^{(0,-8)}(t)} - \frac{1}{2}C_{ii,j}^{(0,-8)}(t) = \frac{(-8)(-9)}{2}\frac{\zeta_{ij}(t)^2}{1+9\alpha_{ij}(t)}.$$

Substituting into (48) gives (52).

We now prove that the stated assumptions imply both the positivity conditions and the logarithmic summability bounds. On the low modes, the condition $(\alpha_{ij}(t), \zeta_{ij}(t)) \in \mathcal{U}_s$ for every $s \in S$ gives

$$1 + \alpha_{ij}(t) > 0, \qquad 1 + (1-s)\alpha_{ij}(t) > 0, \qquad s \in S.$$

This is exactly the positivity needed for (45) and (46). On the tail, since $\rho_{\text{tail}} < 1/9$, (53) implies, for every $s \in S$,

$$1 + \alpha_{ij}(t) \geq 1 - \rho_{\text{tail}} > 0, \qquad 1 + (1-s)\alpha_{ij}(t) \geq 1 - 9\rho_{\text{tail}} > 0.$$

Thus the positivity conditions hold for all modes.

It remains to prove the uniform bounds on the sums of $|\Lambda_{ij}^{(q)}(t)|$ and $|\Lambda_{ij}^{(-8)}(t)|$. On the tail, the same inequalities show that the functions $F_s$ are smooth on a neighborhood of the compact rectangle

$$[-\rho_{\text{tail}}, \rho_{\text{tail}}] \times [-\rho_{\text{tail}}, \rho_{\text{tail}}].$$

Moreover,

$$F_s(0,0) = 0, \qquad \nabla F_s(0,0) = 0.$$

Indeed, the derivative in $\zeta$ is proportional to $\zeta$, while the derivative in $\alpha$ at $(0,0)$ is

$$-\frac{s-1}{2} - \frac{1-s}{2} = 0.$$

Taylor's theorem therefore gives a constant $C_s(\rho_{\text{tail}}) < \infty$ such that

$$|F_s(\alpha, \zeta)| \leq C_s(\rho_{\text{tail}})(\alpha^2 + \zeta^2)$$

on this rectangle. Since $S$ is finite, define

$$C_{\rho_{\text{tail}}} := \max_{s \in S} C_s(\rho_{\text{tail}}) < \infty.$$

Then, for $j \geq J$,

$$|\Lambda_{ij}^{(q)}(t)| \leq C_{\rho_{\text{tail}}}\left(\alpha_{ij}(t)^2 + \zeta_{ij}(t)^2\right), \qquad q = 0, 1, \ldots, 8,$$

and similarly

$$|\Lambda_{ij}^{(-8)}(t)| \leq C_{\rho_{\text{tail}}}\left(\alpha_{ij}(t)^2 + \zeta_{ij}(t)^2\right).$$

Fix $q = 0, 1, \ldots, 8$. For every $i, t, d$,

$$\sum_{j=1}^{d} |\Lambda_{ij}^{(q)}(t)| \leq \sum_{j=1}^{\min\{d, J-1\}} |\Lambda_{ij}^{(q)}(t)| + \sum_{j=J}^{d} |\Lambda_{ij}^{(q)}(t)|.$$

The first term is bounded by $C_{\text{low}}$, and the second is bounded by

$$C_{\rho_{\text{tail}}} \sum_{j=J}^{d} \left( \alpha_{ij}(t)^2 + \zeta_{ij}(t)^2 \right) \leq C_{\rho_{\text{tail}}} C_{\text{tail}}.$$

This proves the required bound for $\Lambda_{ij}^{(q)}$. The proof for $\Lambda_{ij}^{(-8)}$ is identical. $\qquad \square$

The preceding argument keeps the cancellation at zero perturbation explicit. Indeed, if

$$L_i^{(q)}(t, d) := \sum_{j=1}^{d} \Lambda_{ij}^{(q)}(t),$$

then

$$D_i(t, d) = \sum_{q=0}^{8} \binom{8}{q} (-1)^{8-q} \beta_i^q e^{L_i^{(q)}(t,d)}.$$

Since

$$\sum_{q=0}^{8} \binom{8}{q} (-1)^{8-q} = 0,$$

we may also write

$$D_i(t, d) = \sum_{q=0}^{8} \binom{8}{q} (-1)^{8-q} \left( \beta_i^q e^{L_i^{(q)}(t,d)} - 1 \right).$$

Thus $D_i(t, d)$ tends to zero uniformly along any perturbative regime in which $\beta_i \to 1$ uniformly and $L_i^{(q)}(t, d) \to 0$ uniformly for $q = 0, \ldots, 8$.

**Controlling the pairwise variation of the perturbed component scores.** It remains to give a condition ensuring that $\mathcal{S}_{\text{pair}}(t, d)$ is bounded uniformly in $d \geq 1$ and $t \in [0, T]$. Recall from (36) that this quantity is built from the pairwise differences

$$\widetilde{S}_{i,t}^d - \widetilde{S}_{\ell,t}^d,$$

with each coordinate weighted by the preconditioner spectrum $(\gamma_j)$. The next proposition gives a simpler sufficient condition by bounding $H_{i\ell,j}(t)$, the square root of the fourth moment, under $X \sim \varphi_i$, of the $j$-th coordinate of $\widetilde{S}_{i,t}^d - \widetilde{S}_{\ell,t}^d$.

**Proposition B.6.** *Assume that*

$$\sup_{t \in [0,T]} \sup_{d \geq 1} \sum_{i, \ell \in I} w_i w_\ell \left( \sum_{j=1}^{d} \gamma_j \left( a_{i\ell,j}(t)^2 v_{ij}(t) + \mu_{i\ell,j}(t)^2 \right) \right)^2 < \infty. \tag{54}$$

*Then*

$$\sup_{t \in [0,T]} \sup_{d \geq 1} \mathcal{S}_{\text{pair}}(t, d) < \infty. \tag{55}$$

*Proof.* Fix $i, \ell, j, t$ and set

$$A := a_{i\ell,j}(t)^2 v_{ij}(t), \qquad B := \mu_{i\ell,j}(t)^2.$$

By the definition of $H_{i\ell,j}(t)$ in (35),

$$H_{i\ell,j}(t)^2 = 3A^2 + 6AB + B^2 \leq 3(A + B)^2.$$

Therefore

$$H_{i\ell,j}(t) \leq \sqrt{3}\big(a_{i\ell,j}(t)^2 v_{ij}(t) + \mu_{i\ell,j}(t)^2\big).$$

Substituting this bound into the definition of $\mathcal{S}_{\text{pair}}(t,d)$ gives

$$\mathcal{S}_{\text{pair}}(t,d) \leq 3 \sum_{i,\ell \in I} w_i w_\ell \left( \sum_{j=1}^{d} \gamma_j \big(a_{i\ell,j}(t)^2 v_{ij}(t) + \mu_{i\ell,j}(t)^2\big) \right)^2.$$

Thus (54) implies (55). □

Let us now rewrite the condition in the original variables. By (33),

$$a_{i\ell,j}(t)^2 v_{ij}(t) = \frac{\big((\sigma_{\ell j} - \sigma_{ij}) + (\Delta\sigma_{\ell j} - \Delta\sigma_{ij})\big)^2 (\sigma_{ij} + \kappa_t \lambda_j)}{(\sigma_{ij} + \Delta\sigma_{ij} + \kappa_t \lambda_j)^2 (\sigma_{\ell j} + \Delta\sigma_{\ell j} + \kappa_t \lambda_j)^2},$$

while (34) gives

$$\mu_{i\ell,j}(t) = \frac{\Delta m_{ij}}{\sigma_{ij} + \Delta\sigma_{ij} + \kappa_t \lambda_j} - \frac{m_{\ell j} - m_{ij} + \Delta m_{\ell j}}{\sigma_{\ell j} + \Delta\sigma_{\ell j} + \kappa_t \lambda_j}.$$

Hence (54) controls the pairwise differences between the perturbed component scores through the preconditioner-weighted quantities

$$\gamma_j \frac{\big((\sigma_{\ell j} - \sigma_{ij}) + (\Delta\sigma_{\ell j} - \Delta\sigma_{ij})\big)^2 (\sigma_{ij} + \kappa_t \lambda_j)}{(\sigma_{ij} + \Delta\sigma_{ij} + \kappa_t \lambda_j)^2 (\sigma_{\ell j} + \Delta\sigma_{\ell j} + \kappa_t \lambda_j)^2}$$

and

$$\gamma_j \left( \frac{\Delta m_{ij}}{\sigma_{ij} + \Delta\sigma_{ij} + \kappa_t \lambda_j} - \frac{m_{\ell j} - m_{ij} + \Delta m_{\ell j}}{\sigma_{\ell j} + \Delta\sigma_{\ell j} + \kappa_t \lambda_j} \right)^2.$$

This is the point at which the preconditioner enters the responsibility mismatch: sufficient decay of $(\gamma_j)$ prevents the pairwise component-score differences from accumulating over the tail coordinates.

**Concrete sufficient conditions for the responsibility term.** We now collect the previous estimates into a single set of sufficient conditions written directly in terms of the coefficient arrays. The purpose is to make explicit how the two parts of the responsibility bound are controlled. The quantities $\mathcal{D}(t,d)$ and $\mathcal{R}(t,d)$ control how much the responsibilities change; they are governed by relative covariance perturbations and normalized mean perturbations. The quantity $\mathcal{S}_{\text{pair}}(t,d)$ controls the pairwise variation of the perturbed component scores; this is where the preconditioner spectrum $(\gamma_j)$ enters.

**Proposition B.7.** *Write*

$$v_{ij}(t) := \sigma_{ij} + \kappa_t \lambda_j, \qquad \widetilde{v}_{ij}(t) := \sigma_{ij} + \Delta\sigma_{ij} + \kappa_t \lambda_j, \qquad \beta_i := \frac{\widetilde{w}_i}{w_i} = \frac{w_i + \Delta w_i}{w_i}.$$

*Assume first that the weight perturbation is uniformly controlled:*

$$0 < \inf_{i \in I} \beta_i \leq \sup_{i \in I} \beta_i < \infty. \tag{56}$$

*Assume next that there exist $J \in \mathbb{N}$ and $\rho_{\text{tail}} \in (0, 1/9)$ such that the following conditions hold.*

*For each low mode $j = 1, \ldots, J - 1$, there exist constants*

$$c_j > 0, \qquad C_j < \infty, \qquad M_j < \infty$$

*such that, for all $i \in I$ and all $t \in [0, T]$,*

$$c_j \leq \sigma_{ij} + \kappa_t \lambda_j \leq C_j, \tag{57}$$

$$c_j \leq \sigma_{ij} + \Delta\sigma_{ij} + \kappa_t \lambda_j \leq C_j, \tag{58}$$

$$|\Delta m_{ij}| \leq M_j, \tag{59}$$

*and, for every $s \in \{-8, 0, 1, \ldots, 8\}$,*

$$\sigma_{ij} + \kappa_t \lambda_j + (1 - s)\Delta\sigma_{ij} \geq c_j. \tag{60}$$

*For every $i \in I$, every $j \geq J$, and every $t \in [0, T]$,*

$$|\Delta\sigma_{ij}| \leq \rho_{\text{tail}}(\sigma_{ij} + \kappa_t\lambda_j), \qquad |\Delta m_{ij}| \leq \rho_{\text{tail}}\sqrt{\sigma_{ij} + \kappa_t\lambda_j}. \tag{61}$$

*Finally, the normalized tail perturbations are uniformly summable:*

$$\sup_{i \in I} \sup_{t \in [0, T]} \sup_{d \geq 1} \sum_{j=J}^{d} \left( \frac{(\Delta\sigma_{ij})^2}{(\sigma_{ij} + \kappa_t\lambda_j)^2} + \frac{(\Delta m_{ij})^2}{\sigma_{ij} + \kappa_t\lambda_j} \right) < \infty. \tag{62}$$

*Assume, in addition, that the pairwise variation of the perturbed component scores satisfies*

$$\sup_{t \in [0, T]} \sup_{d \geq 1} \sum_{i, \ell \in I} w_i w_\ell \left( \sum_{j=1}^{d} \gamma_j \left[ \frac{\big((\sigma_{\ell j} - \sigma_{ij}) + (\Delta\sigma_{\ell j} - \Delta\sigma_{ij})\big)^2 (\sigma_{ij} + \kappa_t\lambda_j)}{(\sigma_{ij} + \Delta\sigma_{ij} + \kappa_t\lambda_j)^2 (\sigma_{\ell j} + \Delta\sigma_{\ell j} + \kappa_t\lambda_j)^2} \right. \right. \tag{63}$$

$$\left. \left. + \left( \frac{\Delta m_{ij}}{\sigma_{ij} + \Delta\sigma_{ij} + \kappa_t\lambda_j} - \frac{m_{\ell j} - m_{ij} + \Delta m_{\ell j}}{\sigma_{\ell j} + \Delta\sigma_{\ell j} + \kappa_t\lambda_j} \right)^2 \right] \right)^2 < \infty.$$

*Then*

$$\sup_{t \in [0, T]} \sup_{d \geq 1} \mathbb{E}_{\rho_t^d} \left[ \left\| (\Gamma^d)^{1/2} I_2(X, t) \right\|^2 \right] < \infty.$$

*Proof.* We verify that the assumptions imply the three uniform bounds entering Corollary B.2.

First, (56) gives both

$$\beta_* := \inf_{i \in I} \beta_i > 0 \qquad \text{and} \qquad \beta^* := \sup_{i \in I} \beta_i < \infty.$$

We next control $\mathcal{D}(t, d)$ and $\mathcal{R}(t, d)$. Recall the normalized perturbations

$$\alpha_{ij}(t) = \frac{\Delta\sigma_{ij}}{\sigma_{ij} + \kappa_t\lambda_j}, \qquad \zeta_{ij}(t) = \frac{\Delta m_{ij}}{\sqrt{\sigma_{ij} + \kappa_t\lambda_j}}.$$

For the low modes $j = 1, \ldots, J - 1$, the bounds (57)–(60) ensure that the functions

$$F_s\big(\alpha_{ij}(t), \zeta_{ij}(t)\big), \qquad s \in \{-8, 0, 1, \ldots, 8\},$$

are well defined and uniformly bounded in $i \in I$ and $t \in [0, T]$. Since there are only finitely many such modes, this gives the low-mode control required in Proposition B.5.

For the tail modes, (61) gives

$$|\alpha_{ij}(t)| \leq \rho_{\text{tail}}, \qquad |\zeta_{ij}(t)| \leq \rho_{\text{tail}}, \qquad \rho_{\text{tail}} \in (0, 1/9),$$

while (62) is exactly the required uniform summability of

$$\alpha_{ij}(t)^2 + \zeta_{ij}(t)^2.$$

Therefore Proposition B.5 applies. Together with $\beta^* < \infty$, Corollary B.4 yields

$$\sup_{t \in [0, T]} \sup_{d \geq 1} \mathcal{D}(t, d) < \infty, \qquad \sup_{t \in [0, T]} \sup_{d \geq 1} \mathcal{R}(t, d) < \infty.$$

It remains to control $\mathcal{S}_{\text{pair}}(t, d)$. By (33),

$$a_{i\ell, j}(t)^2 v_{ij}(t) = \frac{\big((\sigma_{\ell j} - \sigma_{ij}) + (\Delta\sigma_{\ell j} - \Delta\sigma_{ij})\big)^2 (\sigma_{ij} + \kappa_t\lambda_j)}{(\sigma_{ij} + \Delta\sigma_{ij} + \kappa_t\lambda_j)^2 (\sigma_{\ell j} + \Delta\sigma_{\ell j} + \kappa_t\lambda_j)^2},$$

and by (34),

$$\mu_{i\ell,j}(t) = \frac{\Delta m_{ij}}{\sigma_{ij} + \Delta\sigma_{ij} + \kappa_t\lambda_j} - \frac{m_{\ell j} - m_{ij} + \Delta m_{\ell j}}{\sigma_{\ell j} + \Delta\sigma_{\ell j} + \kappa_t\lambda_j}.$$

Thus (63) is precisely the sufficient condition (54). Proposition B.6 therefore gives

$$\sup_{t\in[0,T]} \sup_{d\geq 1} \mathcal{S}_{\mathrm{pair}}(t,d) < \infty.$$

We have shown that $\beta_* > 0$, that $\mathcal{D}(t,d)$ and $\mathcal{R}(t,d)$ are uniformly bounded, and that $\mathcal{S}_{\mathrm{pair}}(t,d)$ is uniformly bounded. The conclusion follows from Corollary B.2. $\qquad\square$

The proposition shows explicitly how the responsibility mismatch is controlled. The change in the responsibilities is governed by the relative covariance perturbations

$$\frac{\Delta\sigma_{ij}}{\sigma_{ij} + \kappa_t\lambda_j}$$

and the normalized mean perturbations

$$\frac{\Delta m_{ij}}{\sqrt{\sigma_{ij} + \kappa_t\lambda_j}}.$$

The remaining factor is the pairwise variation of the perturbed component scores,

$$\widetilde{S}_{i,t}^d - \widetilde{S}_{\ell,t}^d,$$

weighted coordinate by coordinate by the preconditioner spectrum $(\gamma_j)$. Thus sufficient decay of $(\gamma_j)$ prevents these pairwise component-score differences from accumulating over the tail coordinates.

Finally, defining

$$\mathsf{B}_{\mathrm{resp}}^d(t) := \beta_*^{-2}\mathcal{D}(t,d)^{1/4}\mathcal{R}(t,d)^{1/4}\mathcal{S}_{\mathrm{pair}}(t,d)^{1/2}, \tag{64}$$

Proposition B.1 gives

$$\mathbb{E}_{\rho_t^d}\left[\left\|(\Gamma^d)^{1/2}I_2(X,t)\right\|^2\right] \leq \mathsf{B}_{\mathrm{resp}}^d(t),$$

which proves (12).

## C. Annealed Classifier Free Guidance

We have discussed annealing in the context of sampling from a target distribution via a forward diffusion by an appropriate choice of a time dependent drift. We remark here that annealing also may be useful in the context of Classifier Free Guidance when we also condition on a class label.

In order to relate to the above discussion we write the annealed Langevin diffusion in (3) as

$$dX_t^d = v(t, X_t^d)\, dt + \sqrt{2}\, dW_t^d, \qquad t \in [0, T],$$

where we have chosen $\Gamma^d$ to be the identity and we have

$$v(t, X) = \nabla \log \rho_t^d(X) = \nabla \log \left(\rho_\star^d * \mathcal{N}\left(0, \frac{T-t}{T}C^d\right)\right),$$

with $\rho_\star^d$ the multimodal Gaussian mixture in (1). By appropriate choice of the time dependent drift via the selection of the time horizon $T$ we can show that the distribution of $X_T^d$ is close to $\rho_\star^d$ in Kullback-Leibler (KL) divergence.

To contrast this with diffusion generative modeling consider $p_t^d$ the distribution of $Y^d$ for $Y^d$ an Ornstein-Uhlenbeck process solving

$$dY_t^d = -Y_t^d\, dt + \sqrt{2}\, dW_t^d, \qquad t \in [0, T],$$

with $p_0^d = \rho_\star^d$. Then for a Gaussian noise initial condition, $Y_0 \sim \mathcal{N}(0, \mathcal{I}^d)$, and $Y^d$ solving in reverse time $\tau = T - t$ the SDE

$$dY_\tau^d = w(\tau, Y_\tau^d)\, d\tau + \sqrt{2}\, dW_t^d, \qquad \tau \in [0, T],$$

one can show that for $T$ large the distribution $Y_T^d$ is close to $\rho_\star^d$ for the choice

$$w(\tau, Y) = Y + 2S(\tau, Y)$$

with the score function defined by $S(\tau, Y) = \nabla \log p_{T-\tau}^d(Y)$ (Baldassari et al., 2023).

In Classifier Free Guidance (CFG) (Ho & Salimans) one considers conditional generation via conditioning on observation of a class label $c$ and replace $w$ by

$$w^{\mathrm{CFG}}(\tau, Y) = Y + 2S^{\mathrm{CFG}}(\tau, Y),$$

for

$$S^{\mathrm{CFG}}(\tau, Y) = \nabla \log p_{T-\tau}^d(Y|c) + \omega \left( \nabla \log p_{T-\tau}^d(Y|c) - \nabla \log p_{T-\tau}^d(Y) \right).$$

Here $\omega$ is a 'guidance scale' and by increasing this one can generate samples more aligned with the data. In fact a relatively large guidance scale relaxation $\omega > 0$ is often considered. However, a large guidance scale steers the samples toward a high likelihood mode at the cost of reducing sample diversity and fidelity (Astolfi et al., 2024). It is clear that the distribution induced by CFG in general modifies the (conditional) target distribution and recent work shows in particular that in case of Gaussian mixtures in one and finite dimensions, it results in general in a sharper distribution than the target one (Bradley & Nakkiran; Pavasovic et al., 2025; Wu et al., 2024).

In order to improve the performance of CFG with a constant guidance scale recent works have considered annealing CFG where the guidance scale is chosen to be time dependent (Pavasovic et al., 2025; Yehezkel et al., 2025; Wang et al., 2024). One may then start with a large guidance scale to promote rapid alignment with data and then reduce the guidance scale close to terminal time $T$ to promote alignment with the target distribution. In this manner one can achieve both prompt alignment as well as high fidelity with the target distribution.

We remark also that in (Pavasovic et al., 2025) the authors discuss what they refer to as 'blessing-of-dimensionality' that in fact CFG generates samples with the right target distribution in the limit of high dimension $d$. They do this in the setting of a (centered) two mode gaussian mixture of equal strength, a setting analogous to the one we considered in the motivating example in the introduction, but with equal weights for the modes. In a first phase, up until what the authors refer to as a specification time the CFG correction serves to push the trajectory in the direction of the mean vector. After the specification time the effect of the CFG correction becomes negligible, in the limit of high dimension, thus explaining that CFG may generate high fidelity samples. The authors also introduce a power law form of the CFG score correction motivated by this observation and find that this improves fidelity and which effectively gives a time dependent guidance scale. In (Yehezkel et al., 2025) a learning algorithm is introduced to identify a time dependent score which also introduce an effective score that in general is nonlinear in the difference between the conditional and unconditional scores. Finally, in (Wang et al., 2024) various guidance scales that are only time dependent were considered and the authors found that the choice

$$\omega(\tau) = \omega_0 \left( \frac{T - \tau}{T} \right),$$

gave essentially the best performance among those tried. The annealing schedule for the classifier Free Guidance then starts with a simulation phase where the diffusion is driven toward the data distribution and with a monotonous predetermined tapering. This is analogous to the situation considered in this paper.

## D. Further Details on the Experiments in Section 5

This appendix collects implementation details and additional diagnostics for the experiments of Section 5. All figures were generated in Google Colab (13GB RAM).

### D.1. Targets

In the experiments of Section 5, the target at truncation level $d$ is a two-component diagonal Gaussian mixture on $\mathbb{R}^d$,

$$\rho_\star^d = w_1 \mathcal{N}(m_1^d, \Sigma_1^d) + w_2 \mathcal{N}(m_2^d, \Sigma_2^d), \qquad w = (0.75, 0.25),$$

with separated means $m_1^d = (0, \ldots, 0)$ and $m_2^d = (10, 0, \ldots, 0)$.

For Figure 3, the component covariances are of the form

$$\Sigma_1^d = \tau_1 \Sigma^d, \qquad \Sigma_2^d = \tau_2 \Sigma^d, \qquad \Sigma^d = \mathrm{Diag}(j^{-1.25})_{j=1}^d,$$

with $\tau_1 = 1.2$ and $\tau_2 = 2$.

For Figure 4, the component covariances are identical:

$$\Sigma_1^d = \Sigma_2^d = \Sigma^d, \qquad \Sigma^d = \mathrm{Diag}(j^{-2})_{j=1}^d.$$

## D.2. Implementation of Preconditioned ALD

We implement preconditioned ALD for the experiments as an Euler–Maruyama discretization of the time-inhomogeneous diffusion

$$dX_t^d = \Gamma^d \nabla \log \rho_t^d(X_t^d)\, dt + \sqrt{2\Gamma^d}\, dW_t^d,$$

where $\Gamma^d = \mathrm{Diag}(\gamma_j)_{j=1}^d$ is the diagonal preconditioner and $\rho_t^d$ denotes the Gaussian-smoothed mixture along the annealing path. In discrete time, with stepsize $\Delta t$ and $N$ steps, the iteration is

$$X_{k+1}^d = X_k^d + \Delta t\, \Gamma^d\, s_{\theta_k}^d(X_k^d) + \sqrt{2\Delta t\, \Gamma^d}\, \xi_k^d, \qquad \xi_k^d \sim \mathcal{N}(0, I_d).$$

In the exact-score experiment,

$$s_\theta^d(x) = \nabla \log(\rho_\star^d * \mathcal{N}(0, \theta C^d))(x),$$

where $C^d = \mathrm{Diag}(\lambda_j)_{j=1}^d$ is the diagonal smoothing operator. In the score-error experiment of Figure 4, this analytic score is replaced by the analytic score of the misspecified mixture described below.

We use the linear schedule

$$\theta_k = 2S\left(1 - \frac{k}{N-1}\right), \qquad k = 0, \ldots, N-1,$$

so $\theta_0 = 2S$ and $\theta_{N-1} = 0$. In the experiments, $S = 20, \Delta t = 9 \times 10^{-3}, N = 20000, T_{\mathrm{cont}} := (N-1)\Delta t \approx 1.80 \times 10^2$.

## D.3. ALD Diffusion Design Choices: $\Gamma^d$ and $C^d$

Both the preconditioner and smoothing are chosen as diagonal power laws, possibly up to constant factors,

$$\Gamma^d = \mathrm{Diag}(\gamma_j)_{j=1}^d, \qquad \gamma_j \propto j^{-\alpha_{\mathrm{pre}}}, \qquad C^d = \mathrm{Diag}(\lambda_j)_{j=1}^d, \qquad \lambda_j \propto j^{-\alpha_{\mathrm{smooth}}}.$$

In Figure 3, we contrast two configurations:

$$(\Gamma^d, C^d) = \left(\mathrm{Diag}(j^{-1.5})_{j=1}^d,\ \mathrm{Diag}(j^{-2.7})_{j=1}^d\right), \qquad (\Gamma^d, C^d) = (I_d,\ I_d),$$

with initial smoothing scale $2S = 40$ in both cases.

In Figure 4, we fix

$$C^d = \mathrm{Diag}(j^{-4})_{j=1}^d$$

and vary only the preconditioner:

$$\Gamma^d = \mathrm{Diag}(j^{-3.5})_{j=1}^d, \qquad \Gamma^d = \mathrm{Diag}(j^{-1})_{j=1}^d.$$

The first choice satisfies the sufficient perturbative conditions used in the analysis, whereas the second violates the component-score summability condition.

## D.4. Initialization Choices

In Figure 3, we initialize from the smoothed target law

$$X_0^d \sim 0.75\, \mathcal{N}(m_1^d, \tau_1 \Sigma^d + 2SC^d) + 0.25\, \mathcal{N}(m_2^d, \tau_2 \Sigma^d + 2SC^d),$$

with

$$C^d = \mathrm{Diag}(j^{-2.7})_{j=1}^d, \qquad 2S = 40, \qquad \tau_1 = 1.2, \qquad \tau_2 = 2, \qquad \Sigma^d = \mathrm{Diag}(j^{-1.25})_{j=1}^d.$$

In Figure 4, to isolate the role of preconditioning under score error, we use the same initialization for both preconditioners. Specifically, we initialize the ALD diffusion from a mixture whose component means and covariances coincide with those of the target, but whose mixture weights are incorrect:

$$X_0^d \sim 0.1\,\mathcal{N}(m_1^d, \Sigma^d) + 0.9\,\mathcal{N}(m_2^d, \Sigma^d), \qquad \Sigma^d = \mathrm{Diag}(j^{-2})_{j=1}^d.$$

Thus, relative to the target mixture, the initialization mismatch is confined to the mixture weights. Relative to the smoothed initial law of the annealing path, there is also a covariance mismatch, but it is uniformly controlled in $d$ because $(2S\lambda_j)/\sigma_j = 40j^{-2}$ is square-summable.

### D.5. Score-Error Model in Figure 4: Covariance-Only Perturbations

To simulate the score error while keeping the setting analytically transparent, we compute the drift using the exact score of a misspecified mixture in which only the diagonal covariances are perturbed, while the weights and means remain unchanged. Concretely, if

$$\Sigma_i^d = \mathrm{Diag}(\sigma_{ij})_{j=1}^d,$$

we set

$$\widetilde{\Sigma}_i^d = \mathrm{Diag}(\widetilde{\sigma}_{ij})_{j=1}^d, \qquad \widetilde{\sigma}_{ij} = \sigma_{ij} + \Delta\sigma_{ij},$$

with the covariance-only perturbation

$$\Delta\sigma_{ij} = 0.05\,j^{-3}.$$

The ALD drift is then

$$\Gamma^d \nabla \log\big(\widetilde{\rho}_\star^d * \mathcal{N}(0, \theta C^d)\big).$$

This construction aligns with the covariance-perturbation error model of Section 4.

### D.6. KL Estimation via $k$NN

To estimate $\mathrm{KL}(\rho_\star^d \,\|\, \rho_T^{\mathrm{ALD},d})$, we generate $n = 2500$ samples $X_1^{(p)}, \ldots, X_n^{(p)} \sim \rho_\star^d$ and $m = 2500$ samples $X_1^{(q)}, \ldots, X_m^{(q)} \sim \rho^{\mathrm{ALD},d}$, and apply a fixed-$k$ nearest-neighbor estimator of $\mathrm{KL}(P\|Q)$ (Pérez-Cruz, 2008; Wang et al., 2009). Below we report plots for $k \in \{20, 50, 80\}$ for both Figures 3 and 4 of the paper to confirm that the qualitative trends are robust with respect to the neighborhood size (the main text reports $k = 20$).

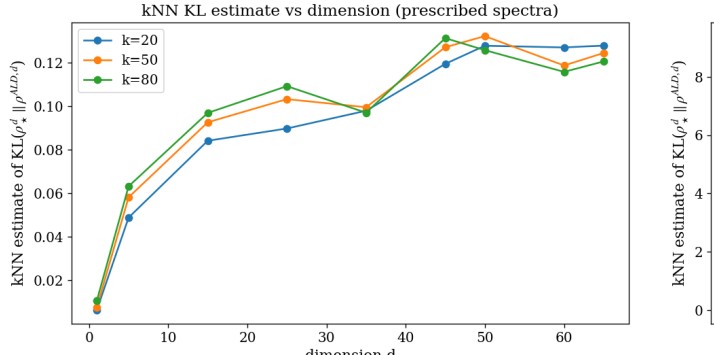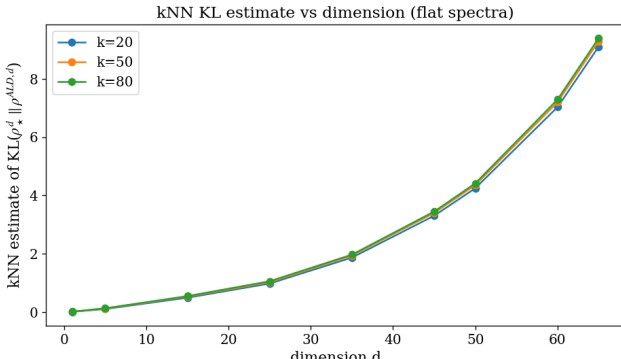

*Figure 5.* Supplementary plots for Figure 3. We report the $k$NN estimate for $k \in \{20, 50, 80\}$ (the main text reports $k = 20$), illustrating that the observed trend is stable across these choices of $k$.

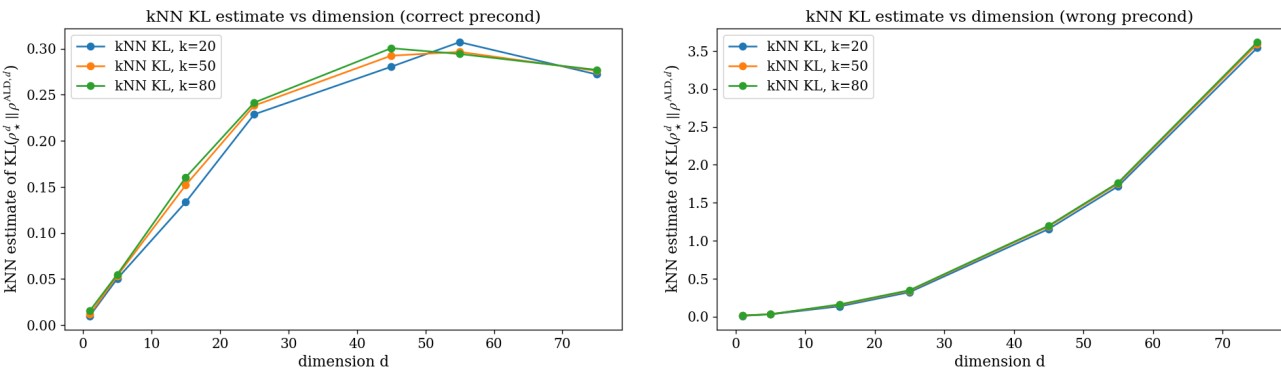

*Figure 6.* Supplementary plots for Figure 4. We report the $k$NN estimate for $k \in \{20, 50, 80\}$ (the main text reports $k = 20$), illustrating that the observed trend is stable across these choices of $k$.

