# OpenReview forum: "Dimension-Free Multimodal Sampling via Preconditioned Annealed Langevin Dynamics"
_ICML.cc/2026/Conference — ICML 2026 regular_

### Official Review · Reviewer_8sr4 · 2026-02-24

**Soundness:** 3
**Presentation:** 3
**Significance:** 2
**Originality:** 3
**Overall Recommendation:** 5
**Confidence:** 2

**Summary:**

This paper studies dimension-robust sampling for multimodal targets using preconditioned annealed Langevin dynamics (ALD). The authors consider an infinite-dimensional Gaussian mixture model and analyze continuous-time ALD under Gaussian smoothing schedules. The main contribution is a dimension-uniform analysis, which shows that under explicit spectral conditions (e.g., the smoothing covariance, mixture geometry, and preconditioner), the tracking error of ALD can be controlled uniformly as the dimension increases. The analysis also covers robustness to imperfect initialization and score mismatch.

One insight from the analysis is that the preconditioner must have a sufficiently decaying spectrum to prevent coordinate-wise error accumulation in high dimensions. The authors further support their analysis with numerical experiments, which demonstrate that appropriate spectral design can remove the empirical dependence of required steps on dimension.

**Compliance With Llm Reviewing Policy:**

Affirmed.

**Final Justification:**

After reading the author's response, my major concern about the preconditioner and its role in the dimension-free result has been resolved. I am therefore adjusting my score accordingly.

**Key Questions For Authors:**

1. In realistic settings where the mixture structure or covariance spectrum is unknown, how should one select the preconditioner $\gamma_j$? Is there a data-driven / adaptive way / instructions to approximate the required spectral decay?

2. Do the authors expect the dimension-free guarantees to extend to discrete-time ALD with a fixed step size independent of dimension? Could discretization error destroy dimension-uniform stability?

3. see Weaknesses.

**Limitations:**

1. The experimental validation is limited to toy examples like Gaussian mixtures. No validation in a real-world dataset. Since this work mainly focuses on theoretical analysis / convergence guarantee for continuous Langevin SDE, I think this is acceptable.

2. Also please check the first point of Weaknesses.

**Strengths And Weaknesses:**

# Strengths

1. The paper provides a technically careful and structured infinite-dimensional analysis of annealed Langevin dynamics, which appears to meaningfully extend prior function-space MCMC results to multimodal Gaussian mixtures. The theoretical development is organized and appears internally consistent within the stated assumptions.

2. The treatment of imperfect initialization and score mismatch strengthens the theoretical contribution.

3. The paper clarifies why naive ALD may fail in high dimensions and shows how spectral decay can prevent coordinate-wise error accumulation.

# Weaknesses

1. A central mechanism in the paper is the introduction of a preconditioner that scales both the drift and diffusion terms in the Langevin SDE: $dX_t = \Gamma \nabla \log \rho_t(X_t), dt + \sqrt{2\Gamma}, dW_t.$

   If we ignore the diffusion term, the dynamics reduce to gradient descent, which is typically dimension-free under reasonable conditions. In contrast, the dimension dependence in Langevin dynamics primarily *stems from the Brownian motion term*, since the total noise energy grows with dimension.

   Therefore, when the paper imposes a decaying spectrum on the preconditioner (i.e., shrinking ($\gamma_j$) in high-frequency directions), one possible interpretation from my understanding is that the authors are **effectively and artificially controlling the magnitude of the noise magnitude as the dimension increases.**

   This raises my conceptual concern: if the preconditioner strongly attenuates high-frequency noise, does the sampler still sufficiently explore those directions? While mathematically the invariant distribution may remain correct when drift and diffusion are scaled consistently, it is unclear to me whether the resulting dynamics preserve meaningful exploration in practice. There seems to be a potential tradeoff between stability (dimension-uniform bounds) and exploration (adequate stochasticity in all coordinates), but the paper does not explicitly discuss this tradeoff.

   I would appreciate the authors could include discussion to clarify:

      - Whether the dimension-free result is fundamentally due to controlling Brownian motion growth?
      - How to interpret the preconditioner beyond "decreasing noise magnitude"?
      - Whether strong spectral decay could limit effective exploration in high-frequency directions?

2. The theoretical results are established for continuous-time SDEs, while experiments necessarily rely on discretized implementations. The paper does not analyze discretization error or clarify whether dimension-free guarantees extend to discrete-time ALD. This I believe induces a gap between analysis and experiments that should be addressed or discussed more explicitly.

---

> ### Author Rebuttal · Authors · 2026-03-30
>
> We would like to thank the Reviewer for their constructive feedback.
>
> We are happy to clarify our manuscript in response to the Reviewer's questions.  We hope this will lead to an increase in score.
>
> **Conceptual question about preconditioning with decaying spectrum.** The role of preconditioning in achieving dimension-robust behavior is not new; we referenced several related works in the Introduction, e.g., [1,2,3,4]. In general, it is not accurate to interpret the preconditioner simply as a device for "shrinking the Brownian motion" in high-frequency directions: the preconditioner scales \emph{both} drift and diffusion, and what matters is their joint interaction with the geometry of the target. As an example, consider a Gaussian target
> $$\pi = \mathcal N(0,\Sigma), \qquad \Sigma=\operatorname{Diag}(\sigma_j),$$
> and the preconditioned Langevin SDE
> $$
> dX_t = -\Gamma \Sigma^{-1}X_t dt + \sqrt{2\Gamma} dW_t, \qquad \Gamma=\operatorname{Diag}(\gamma_j).$$
> Mode by mode, this reads
> $$dX_t^{(j)} = -\frac{\gamma_j}{\sigma_j}X_t^{(j)} dt + \sqrt{2\gamma_j} dW_t^{(j)}.$$
> Thus the mean-reversion rate in coordinate $j$ is
> $$
> \frac{\gamma_j}{\sigma_j},
> $$ while the invariant variance in that coordinate remains exactly $\sigma_j$. In particular, for
> $$
> Y_t^{(j)}:=\frac{X_t^{(j)}}{\sqrt{\sigma_j}},$$ we have
> $$dY_t^{(j)} = -\frac{\gamma_j}{\sigma_j}Y_t^{(j)} dt + \sqrt{2\frac{\gamma_j}{\sigma_j}} dW_t^{(j)}.$$
> This shows that the relevant quantity is not the absolute size of $\gamma_j$, but the ratio $\gamma_j/\sigma_j$. For instance, when $\gamma_j=\sigma_j$, all normalized modes satisfy the same dynamics
> $$ dY_t^{(j)}=-Y_t^{(j)} dt+\sqrt{2} dW_t^{(j)}.$$
> This analysis interacts with one key feature of the infinite-dimensional setting, i.e. that the target has trace-class covariance, so that $(\sigma_j)$ decays sufficiently fast as $j\to \infty$. Here choosing a decaying preconditioner should not be interpreted as "artificially underexploring small-variance directions''; rather, its purpose is to make the sampler explore each direction on the _correct scale_ relative to the target geometry. That said, there is a possible tradeoff here. In the Gaussian example above, if $\gamma_j$ is chosen much smaller than $\sigma_j$, then the normalized dynamics in the $j$th coordinate become slower, and one may therefore lose effective exploration in that standardized direction. In our setting, the same interpretation is less straightforward. In the exact-score setting of Section 3, the key issue is the spectral balance between the preconditioner and the smoothing covariance.  In that regime, a decaying preconditioner is not being used to suppress target directions, but to balance the geometry of the dynamics along the annealing path; under the admissible spectral choices identified in Section 3, the annealing-induced KL error can be made small without dimension-dependent deterioration. In the score-perturbed setting of Section 4, however, stronger decay of $(\gamma_j)$ may be needed to prevent the accumulation of score errors across infinitely many coordinates. This effect could therefore be interpreted as damping the _contribution of high-frequency score error to the overall error_, rather than removing or ignoring high-frequency target modes.
>
> **Continuous-time SDE vs. discretization (Weakness 2 + Question 2).** Our continuous-time analysis isolates the dimension-dependent mechanism through which the smoothing covariance and the preconditioner affect the dynamics. Crucially, this phenomenon is not tied to a particular discretization scheme, but to the infinite-dimensional spectral structure of the target and the sampler. For this reason, we expect the same mechanism to remain relevant for discretized ALD as well. This is supported by our experiments, and also by the comparison with [1], where no dimension-uniform stability is observed in the absence of the appropriate spectral structure. We will make this point explicit in the Discussion and Future Work section of the paper.
>
> **How to choose $\Gamma$.** We thank the reviewer for this question. In practice, our analysis does not require recovering the full mixture structure or exact spectrum. The key is to avoid strong spectral mismatch between the preconditioner and the smoothing operator, especially in the tail, where only the decay rate matters. This suggests a practical strategy: work in a basis where the smoothing covariance is approximately diagonal, match the tail decay of $\Gamma$ (or slightly damp it), and adapt only a few leading modes using problem-specific information or exploratory samples. While not analyzed rigorously, our theory indicates that coarse tail matching combined with low-rank adaptation is sufficient.
>
> **References**
>
> [1] S.L. Cotter et al., arXiv:1202.0709
>
> [2] A. Beskos et al., arXiv:1606.06351
>
> [3] J. Pidstrigach et al., arXiv:2302.10130
>
> [4] L. Baldassari et al., arXiv:2505.18276

---

> > ### Author Rebuttal · Reviewer_8sr4 · 2026-04-03
> >
> > I appreciate the authors' detailed response. My major conceptual concern about the role of the preconditioner in the dimension-free result has been addressed. I have adjusted my score to reflect this.

---

### Official Review · Reviewer_kBG3 · 2026-03-09

**Soundness:** 3
**Presentation:** 2
**Significance:** 2
**Originality:** 3
**Overall Recommendation:** 4
**Confidence:** 3

**Summary:**

The article studies the dependency of the Annealed Langevin Dynamics (ALD) algorithm with respect to the dimension.

In a first idealized model, where we can sample a random variable $X + \varepsilon$, with $X$ distributed according to the target measure and $\varepsilon$ a centered Gaussian vector, the article shows that by taking $\varepsilon$ small enough with respect to the dimension, a specific annealing schedule of the ALD algorithm reaches a precision $\delta$ after a time $T$ that is independent of the dimension.

In a second, more realistic setting, the case where the initial particle is sampled from a known distribution is studied. The authors show that if this distribution is sufficiently close to the target distribution as the dimension increases, then there exists an annealing schedule such that the ALD algorithm reaches a precision $\delta$ after a time $T$ that is independent of the dimension.

The authors support their claims with numerical experiments.

The contribution of the paper is that it proposes a method to ensure the robustness of the ALD algorithm with respect to the dimension.

**Compliance With Llm Reviewing Policy:**

Affirmed.

**Final Justification:**

The paper tackles an important and interesting problem, namely the choice of parameters that achieve, for the annealed Langevin algorithm, an error independent of the dimension. Not surprisingly, this can be achieved when the initialization is close to the target. This raises the question of the practical relevance of the paper when such an initialization close to the target is unavailable.

The paper also provides some insights into the choice of preconditioning. This might be useful for practitioners; however, their first theorem suggests taking $\gamma_i = i^p$ for $p$ large enough. This recommendation stems from the continuous-time analysis, whereas in the discrete-time setting, $\gamma_i$ should be chosen more carefully.

Discussions with the authors have confirmed my concerns: without a discrete-time analysis, the paper has limited significance for practitioners. The authors suggest addressing this issue in the appendix, but if the paper is accepted, I believe they should only include a discussion of the discrete-time algorithm in the main body of the paper.

Overall, the paper is theoretically solid and interesting, but I believe it lacks precise guidelines (or examples) demonstrating its practical usefulness. Furthermore, the fact that the results are presented in continuous time diminishes their impact for practitioners.

**Key Questions For Authors:**

- 1: In L. 156, why is it important to study the continuous-time particle system in high dimension before discretizing it? It seems to me that discretization could provide additional insights.

- 2: What is the impact of changing the schedule independently of the initial measure? In the theoretical results, modifying the schedule also modifies the initial measure, which makes it difficult to determine an optimal schedule for a fixed initial measure.

- 3: How can your result be used in a practical application where the initial measure can only be taken at a distance that grows polynomially with the dimension from the target?

I do not see how the contribution of this paper could be useful in practical applications, and I would be happy to raise my score if the answers to my questions demonstrate its practical relevance.

**Limitations:**

yes

**Strengths And Weaknesses:**

**Strengths**:
- The authors provide an explicit characterization of the dependence of the Annealed Langevin Algorithm on the dimension and offer insights regarding the choice of the annealing schedule.

**Weaknesses**:
- The main contribution is not entirely clear. In L. 95, the authors motivate the design of an ALD algorithm whose performance is independent of the dimension. However, this is not entirely accurate, as unrealistic assumptions on the initial measure are required. In particular, the fact that the distance between the initial distribution and the target distribution is assumed to be independent of the dimension should be clearly stated before presenting Figure 1.

- Since all theoretical results and numerical simulations are presented in an unrealistic setting, where the distance between the initial measure and the target is bounded uniformly in $d$, we do not know how these results could be useful in a "real-world" application.

- As mentioned below Theorem 3.1, the impact of the choice of the preconditioning matrix $\Gamma$ is unclear in continuous time, since it can be compensated for by adjusting the step size. A discrete-time result would therefore provide additional insight into the role of preconditioning.

- Some advice:
  - The assumptions should be stated explicitly in the theoretical results.
  - Perhaps replace $T^d$ with $T^{(d)}$, since the notation $^d$ may be interpreted as an exponent.

---

> ### Author Rebuttal · Authors · 2026-03-30
>
> We thank the Reviewer for the constructive feedback. In the following, we clarify the significance of the paper by addressing the three main concerns raised in the review:
>
> 1. Does the paper assume that the distance between the initialization and the target is independent of the dimension?
> 2. What is the impact of preconditioning in continuous-time?
> 3. Does the analysis account for changing the schedule independently of the initial measure?
>
> As these seem to be the main issues limiting the paper's usefulness, we hope the answers below will support an increase in score.
>
> **On the assumption that the initialization-to-target distance is independent of the dimension (Weaknesses 1-2 + Question 3).** Our analysis does _not_ assume that the KL distance between the initial distribution and the target distribution is bounded uniformly in the dimension. To make this point explicit, consider the single-Gaussian case $I=${$1$} where the score and the initialization are exact, so that the upper bound of Proposition 4.1 for the total error $\mathrm{KL}(\rho_\star^d || \rho_T^{\text{ALD},d})$  reduces to the annealing bias term alone. According to Theorem 3.1, this bias is controlled whenever
>           $$
>             \sup_{d\geq 1} \mathcal K_d <\infty, \qquad
>            \mathcal K_d = \frac{1}{16}
> \sum_{j=1}^d
> \frac{\lambda_j}{\gamma_j}
> \log \left(1+\frac{\lambda_j}{\sigma_{j}}\right).
>             $$
> At the same time, the KL divergence between the target and the initialization can be much larger. Indeed, let
> $$
> \rho_\star^d=\mathcal N(m^d,\Sigma^d),
> \qquad
> \rho_0^d=\mathcal N(m^d,\Sigma^d+C^d),
> $$
> with
> $$
> \Sigma^d=\mathrm{Diag}(\sigma_1,\dots,\sigma_d),
> \qquad
> C^d=\mathrm{Diag}(\lambda_1,\dots,\lambda_d).
> $$
> Then
> $$
> \mathrm{KL}(\rho_\star^d || \rho_0^d)
> =
> \frac12\sum_{j=1}^d \log \left(1+\frac{\lambda_j}{\sigma_j}\right) - \frac{\lambda_j}{\sigma_j+\lambda_j}.
> $$
> Now let
> $$
> \gamma_j=j^{-2},
> \qquad
> \sigma_j=j^{-3},
> \qquad
> \lambda_j=j^{-13/4}.
> $$
> Then
> $$
> \mathcal  K_d  =
> \frac{1}{16}\sum_{j=1}^d
> j^{-5/4}\log \left(1+j^{-1/4}\right)
> \asymp
> \sum_{j=1}^d j^{-3/2},
> $$
> which is uniformly bounded in $d$.
> On the other hand, since $\lambda_j/\sigma_j=j^{-1/4}$,
> $$
> \mathrm{KL}(\rho_\star^d || \rho_0^d)
> \asymp
> \sum_{j=1}^d
> \left(\frac{\lambda_j}{\sigma_j}\right)^2
> =
> \sum_{j=1}^d j^{-1/2}
> \asymp d^{1/2}.
> $$
> So in this perfectly admissible regime, with trace-class preconditioner $\Gamma^d=\mathrm{Diag}(j^{-2})_{j=1}^d$, _the KL distance between the initial law and the target grows polynomially with the dimension, while the annealing bias remains dimension-uniformly controlled_.
> In other words, our theory does not rely on the requirement that the initial distribution remain at uniformly bounded KL distance from the target as $d$ grows. We hope this clarification addresses the reviewer's concern.
>
> **Unclear impact of preconditioning in continuous time (Weakness 3).** We agree that a discrete-time result would provide additional insight. However, Theorem 3.1 does not rely on scalar rescaling of $\Gamma^d$, which only changes the time scale in continuous time. What matters is the _spectral shape_ of $\Gamma^d$ relative to the smoothing covariance $C^d$, which determines dimension-uniform control. This mechanism cannot be removed by time rescaling and becomes even more critical in Section 4 when score and initialization errors are included. The continuous-time analysis isolates this effect without discretization artifacts. A discrete-time analysis would be a natural extension, and we expect similar behavior, as supported by our numerical results.
>
> **Line 156.** This is motivated by a discretization-invariant perspective, natural when the target is infinite-dimensional (e.g., a function in a Hilbert space). The goal is not just to perform well at a fixed resolution, but to remain stable under refinement---a classical issue in areas like Bayesian inverse problems, where performance may degrade with discretization [1-3]. Studying continuous-time dynamics isolates the dimension-dependent effect of preconditioning without discretization artifacts. A discretization analysis would be valuable, but lies beyond the scope of this work.
>
> **Initialization vs. annealing schedule.** In Section 3, the schedule and warm start are coupled by construction, since the annealing path defines the reference initialization. This setting isolates how the schedule and preconditioner control the annealing bias under exact warm start, without accounting for initialization mismatch.
> Proposition 4.1 decouples these effects: starting from a fixed initialization, the error splits into initialization, score, and annealing terms. The initialization term measures the mismatch with the schedule-induced reference, while the others capture the effects of the annealing path and score perturbation. Thus, for a fixed initialization, choosing the schedule amounts to balancing these contributions, and Proposition 4.1 provides the natural framework for this analysis.

---

> > ### Author Rebuttal · Reviewer_kBG3 · 2026-04-04
> >
> > I thank the authors for their responses. They are correct in that they do not assume that the KL divergence between the target and the initialization is uniform in the dimension in Theorem 3.1. However, in this theorem, one can choose any sequence $\Gamma$.
> >
> > Therefore, by choosing $\gamma_i = i^{p}$, where $p$ can be arbitrarily large, the assumption of the theorem can be made to hold for any $C$ and $\Sigma$.
> >
> > I understand that this issue arises from the continuous-time analysis. In the discrete-time setting, not every choice of $\Gamma$ is admissible, as the discretization introduces a bias that accumulates over time. Therefore, one must impose a condition of the form (as in L. 383):
> > $$
> > \sum \frac{\gamma_i}{\sigma_i} < \infty.
> > $$
> >
> > However, under this condition, the example provided in the rebuttal by the authors no longer holds, and the KL divergence between the initialization and the target becomes uniformly bounded with respect to the dimension.
> >
> > Moreover, I still wonder what takeaway a practitioner should draw from this work, given that the assumptions on the initialization of the algorithm appear, to the best of my knowledge, unrealistic in practice.

---

> > > ### Author Response · Authors · 2026-04-05
> > >
> > > We thank the reviewer for this thoughtful follow-up. We believe the comment is very helpful, because it highlights the constraints that may arise after discretization. First, we would like to emphasize that any discrete-time analysis necessarily depends on the numerical scheme under consideration. If one chooses Euler-Maruyama (EM), then stronger spectral assumptions may be needed to keep the discretization bias uniform in the dimension; conditions of the form $$\sum_j \frac{\gamma_j}{\sigma_j}<\infty$$ arise in that setting. At the same time, it is important to stress that this is a statement about a specific numerical scheme, rather than about ALD itself. This is precisely why we view our continuous-time analysis as practically useful, since it identifies the spectral mechanism underlying dimension-uniform control at the level of the dynamics, before one imposes scheme-dependent restrictions required by a particular discretization. For example, one may consider discretizations that are more closely adapted to the structure of the dynamics. For simplicity consider a diagonal shared-covariance setting. Here the exact-score preconditioned ALD takes the form $$dX_t^d = - A_t^d X_t^d dt + A_t^d\bar m_t^d(X_t^d) dt + \sqrt{2\Gamma^d} dW_t^d, \qquad A_t^d:=\Gamma^d(D_t^d)^{-1}, $$ where $D_t^d=\Sigma^d+\kappa_t C^d$, with $\kappa_t=(T-t)/T$, and $\bar m_t^d$ denotes the posterior-mean map associated with the annealed law $\rho_t^d$.  Guo et al. [1] noticed that, while EM is a straightforward choice, it is suboptimal for ALD. Their alternative is to integrate the linear Gaussian part exactly on each step, while freezing only the nonlinear posterior-mean term. More precisely, on $$ 0=t_0<t_1<\dots<t_N=T, \qquad h_n:=t_{n+1}-t_n, \qquad h_{\max}:=\max_{0\le n\le N-1} h_n, $$ one considers on each interval $[t_n,t_{n+1})$ the frozen SDE $$ dY_t^d = - A_t^dY_t^d dt+A_t^d\bar m_{t_n}^d(Y_{t_n}^d) dt+\sqrt{2\Gamma^d} dW_t^d,\qquad t\in[t_n,t_{n+1}), $$ started from $Y_0^d\sim \rho_0^d$. Since the linear part is diagonal, this step can be solved coordinate-wise. Writing $$ v_{t,j}:=\sigma_j+\kappa_t\lambda_j,\qquad a_j(t):=\frac{\gamma_j}{v_{t,j}}, $$ define $$ \phi_{n,j}:=\exp \left(-\int_{t_n}^{t_{n+1}} a_j(s) ds\right), $$ $$ q_{n,j}:= 2\gamma_j\int_{t_n}^{t_{n+1}} \exp \left(-2\int_s^{t_{n+1}} a_j(u) du\right) ds. $$ The resulting exact-linear-part scheme is $$ Y_{n+1,j} = \phi_{n,j}Y_{n,j} + (1-\phi_{n,j})\bar m_{t_n,j}(Y_n) + \xi_{n,j}, $$ where the $\xi_{n,j}$ are independent centered Gaussian random variables with variances $q_{n,j}$. We believe it is useful to spell this out, because it makes clear that the contraction coming from the linear Gaussian part is treated exactly at each step, and only the nonlinear posterior-mean term is frozen. Thus, the discretization error arises from the one-step freezing defect, rather than from an Euler approximation of the full drift.  Our preliminary analysis in this setting indicates that, for this type of discretization, one obtains a bound of the form $$ \mathrm{KL} (\rho_\star^d|\mathrm{Law}(Y_T^d) ) \le 2J_{ann}^d(T)+C h_{\max}, $$ so that the discretization error is controlled by the annealing bias and the one-step freezing error, rather than by a dimension-uniform bound on $$ \mathrm{KL}(\rho_\star^d|\rho_0^d). $$ In this setting, we are able to construct explicit diagonal examples for which the relevant coefficient-level summability conditions remain finite, the above bound is uniform in $d$, and yet $$ \mathrm{KL}(\rho_\star^d |\rho_0^d)\to\infty. $$ For instance, taking $$ \sigma_j=j^{-4},\qquad \lambda_j=j^{-9/2},\qquad \gamma_j=j^{-3}, $$ and $$ m_{1,j}=j^{-5},\qquad m_{2,j}=j^{-5}+\mathbf 1_{\{j=1\}}, $$ one gets uniform control of the coefficient sums entering the exact-linear-part discretization analysis, while $$ \mathrm{KL}(\rho_\star^d\|\rho_0^d) \gtrsim \sum_{j=2}^d \left(\frac{\lambda_j}{\sigma_j}\right)^2 = \sum_{j=2}^d j^{-1}\to\infty. $$ This is exactly why we believe the continuous-time analysis is significant. Its purpose is to isolate the spectral mechanism governing dimension-uniform control before one imposes the constraints required by a particular discretization. Our preliminary analysis indicates that such assumptions required by EM are tied to that particular discretization, and that different schemes may impose different, potentially more favorable, conditions on the discretized ALD dynamics.  That said, we very much appreciate that the reviewer raised this point. If the reviewer feels that such an analysis is essential for the paper to be complete, we will include it in the Appendix. However,  our preference would be to develop this analysis following the reviewer's insightful remarks more fully in follow-up work, since we believe that the continuous-time analysis is already a relevant contribution in its own right, as it helps clarify the spectral mechanism underlying dimension-uniform control in ALD.
> > >
> > >
> > > [1] W. Guo et al., arXiv:2407.16936

---

### Official Review · Reviewer_zs8T · 2026-03-13

**Soundness:** 3
**Presentation:** 3
**Significance:** 3
**Originality:** 3
**Overall Recommendation:** 5
**Confidence:** 4

**Summary:**

This paper analyzes annealed Langevin dynamics (ALD) in the infinite dimensional setting, where $d$ grows. The paper investigates design choices under which the ALD stays uniformly stable as $d$ grows, which is a significant development to show why ALD can be effective in high-dimensional, real-world settings.

**Compliance With Llm Reviewing Policy:**

Affirmed.

**Final Justification:**

I think this paper considers an interesting setting and gives informative insights, authors replied my questions satisfactorily.

**Key Questions For Authors:**

1) In the paper, both annealing and a decaying preconditioner appear necessary for dimension-uniform guarantees. However, empirically annealing alone already enables effective multimodal exploration in high dimensions, as evidenced by score-based diffusions that do not use explicit preconditioning. Could the authors clarify which phenomenon specifically fails without preconditioning? In particular, is preconditioning fundamentally required for the dimension-uniform theory, or mainly for robustness to score and initialization errors?

2) The analysis is carried out for Gaussian mixtures with diagonal covariances, where the preconditioner is chosen via coordinatewise decay conditions. Is there a broader insight for how one should design preconditioners for more general multimodal targets? In particular, can the authors clarify whether their results could give insights on practical ways to choose the preconditioner (e.g., from the target covariance or score structure) beyond the Gaussian-mixture setting -- perhaps using similar conditions on inverse Hessians and using them as preconditioners?

3) The analysis focuses on continuous-time ALD, while the experiments use an Euler–Maruyama implementation. Could the authors comment on whether the dimension-uniform guarantees are expected to remain stable under discretization? In particular, are there step-size conditions under which the discretized algorithm inherits the same dimension-robust behavior?

4) The analysis considers a specific Gaussian-smoothing annealing path with a fixed time horizon. How sensitive are the dimension-uniform guarantees to the particular annealing schedule? For instance, would similar results hold under alternative schedules or noise families commonly used in diffusion models?

**Limitations:**

yes

**Strengths And Weaknesses:**

Strengths:

- The paper is analyzing an interesting and significant stochastic process, diffusion annealed Langevin dynamics, in an infinite dimensional setting which is both significant and original. This sheds light onto understanding precise improvement/impact of annealing + preconditioning in the paper's setting.
- The paper is very nicely written, the bounds and the results are intuitive and informative, the presentation is very well structured; and the results are sound.

Weaknesses:

- The setup is a bit limited, which is mixture of Gaussians. While a good example of multimodality, I think the title is a bit too general for this setting.

---

> ### Author Rebuttal · Authors · 2026-03-30
>
> We thank the Reviewer for the positive feedback and are happy to clarify the manuscript in response to their questions. We hope this will lead to an increase in score.
>
> **Title a bit too general.** Thanks for the suggestion. We can add "Gaussian mixtures'' to the title.
>
> **Is preconditioning fundamentally required for the dimension-uniform theory?** This is a great question. To illustrate the point, let us consider the simple case of a unimodal diagonal Gaussian target. In coordinates, ALD reads
> $$
> d X_{t}^{(j)} = -\frac{\gamma_j}{\lambda_j + \kappa_t } X_{t}^{(j)} dt + \sqrt{2\gamma_j}  dW_t^{(j)}.
> $$
> Assume first that $\gamma_j=1$ (no preconditioning). As $t \to T$, the annealing parameter satisfies $\kappa_t \to 0$, so the drift coefficient approaches $1/\lambda_j$.
> Since $\lambda_j \to 0$ as $j\to\infty$, the terminal part of the dynamics becomes arbitrarily stiff in the high-order modes.
> The practical consequence is that, near the target, any time discretization of the ALD dynamics must use a step size $h$ that resolves the largest drift scale.
> In the terminal regime $\kappa_t \approx 0$, this reduces to $
> h \lesssim \lambda_d$, which tends to $0$ as $d\to\infty$. Thus, without preconditioning, the algorithm becomes increasingly stiff as the discretization is refined, and one loses any dimension-robust control of the final part of the annealing path.
> By contrast, if one chooses $\gamma_j \asymp \lambda_j$, then $\gamma_j/(\lambda_j+\kappa_t)$ remains uniformly bounded as $\kappa_t \to 0$, avoiding spectral imbalance across modes. This shows that preconditioning is essentially required in infinite dimensions: annealing regularizes high modes at positive noise levels, but stiffness reappears near the target. We will add this clarification and thank the reviewer for the suggestion.
>
> **Practical ways to select the preconditioner.** Our analysis does not prescribe a unique preconditioner, but identifies a spectral compatibility condition with the smoothing covariance as the key to dimension-uniform control. In the misspecified setting, however, the preconditioner also acts on score errors, so choices that are too large in poorly estimated directions may amplify them and break stability. The main design principle is therefore to balance spectral compatibility with robustness to score error.
> In practice, this suggests using regularized preconditioners whose spectrum matches the smoothing operator, especially in the tail where score accuracy is weakest, while allowing limited adaptation in well-informed directions. Inverse-Hessian or score-based constructions may be useful, but only with sufficient damping or truncation. We will clarify this trade-off, which our continuous-time analysis makes explicit.
>
> **Continuous-time SDE vs. discretization.** We expect dimension-uniform behavior to persist under discretization, although we do not prove it. Our continuous-time analysis isolates the key spectral mechanism linking the smoothing covariance and preconditioner, which we expect to also govern the discretized algorithm. This is supported by our experiments and by prior work showing lack of stability without preconditioning. A rigorous result would require a separate discretization analysis, combining our approach with existing techniques [1], which we will mention as a natural next step.
>
> **Dimension-uniform guarantees vs. annealing schedule.** The linear schedule used in the proofs is not essential. What matters is that the annealing path is given by an explicit Gaussian smoothing whose covariance decreases in time. More precisely, let
> $$
> \rho_t^d=\rho_\star^d * \mathcal N(0,\kappa(t/T) C^d),
> $$
> where $\kappa(1)=0$ and  $\kappa(0)=1$.
> The same argument as in the proof of Theorem 3.1 give
> $$
> \mathrm{KL} (\rho_\star^d || \rho_T^{\mathrm{ALD},d})
> \le
> \frac{\mathcal K_d}{T}
> $$
> with
> $$
> \mathcal K_d
> :=
> \frac{1}{16}
> \sum_{i\in I} w_i
> \sum_{j=1}^d
> \frac{\lambda_j^2}{\gamma_j} q_{ij},\qquad
> q_{ij} = \int_0^1
> \frac{(\kappa'(u))^2}{\sigma_{ij}+\kappa(u)\lambda_j} du
> $$
> If we take $
> T^d= \varepsilon^{-1} \mathcal K_d$,
> then
> $$
> \mathrm{KL} (\rho_\star^d||\rho_{T_d}^{\mathrm{ALD},d} )\le \varepsilon.
> $$
> This shows that the dimension-uniform guarantees are not tied to the linear choice $\kappa(t/T)=(T-t)/T$; the logarithmic factor in Theorem 3.1 is replaced by the schedule functional $q_{ij}$.
> Moreover, if $\kappa$ is nonincreasing,
> $$\mathcal K_d \leq Q \frac{1}{16} \sum_{i\in I} w_i \sum_{j=1}^d \frac{\lambda_j}{\gamma_j} \log \left(1+\frac{\lambda_j}{\sigma_{ij}}\right),$$
> where $Q=||\kappa'||_{L^\infty(0,1)}$. Whenever $Q$ is bounded independently of $d$, the same summability condition of Theorem 3.1 gives a dimension-uniform annealing horizon.
> A similar remark applies to Proposition 4.4. Thus, the same message of the paper remains valid under a broad class of annealing schedules. We will add a remark in the revised version to make this explicit.
>
> **References**
>
> [1] L. Baldassari et al., arXiv:2505.18276

---

> > ### Author Rebuttal · Reviewer_zs8T · 2026-04-02
> >
> > Thanks for your reply -- I am happy with the response, I will adjust my score.

---

### Official Review · Reviewer_AZAG · 2026-03-17

**Soundness:** 3
**Presentation:** 3
**Significance:** 1
**Originality:** 3
**Overall Recommendation:** 4
**Confidence:** 3

**Summary:**

This paper studies annealed Langevin dynamics for sampling from multimodal distributions, focusing on a refinement setting where higher dimension means a finer truncation of the same infinite-dimensional Gaussian mixture. The main contribution is a theoretical analysis showing when the sampling error can remain uniformly controlled as dimension grows, based on the spectral choices of the annealing covariance and the preconditioner. The paper also extends the analysis to imperfect initialization and approximate scores, showing that preconditioning plays an important role in preventing these errors from accumulating across coordinates.

**Compliance With Llm Reviewing Policy:**

Affirmed.

**Final Justification:**

The authors have addressed all of my questions. I find the paper well written, and I believe it begins to offer meaningful theoretical insight into a challenging problem. My only remaining hesitation is that the setting still feels somewhat idealized, which limits the paper’s ability to provide concrete guidance to practitioners. However, the paper's primary contribution is theoretical, and on that basis I believe it should be accepted.

**Key Questions For Authors:**

1) The paper gives a clean continuous time theory, but I am still unclear on the practical implications. Are there any concrete takeaways a practitioner should draw from these results when implementing ALD in a realistic finite dimensional and discretized setting?

2) How relevant do the main assumptions seem for practical ALD problems? In particular, I would appreciate more discussion of whether the shared diagonalization, continuous time analysis, and misspecified Gaussian mixture score model are mainly technical devices for tractability, or whether the authors see them as good approximations to important real use cases.

3) The paper emphasizes the role of the smoothing covariance and preconditioner spectra in obtaining dimension uniform control. Do the authors believe this message is specific to the setting studied here, or do they expect it to hold more broadly for multimodal ALD beyond the Gaussian mixture setting?

**Limitations:**

Yes

**Strengths And Weaknesses:**

Soundness: The paper seems technically sound. The main claims are supported by explicit theoretical bounds, and the experiments are aligned with the theory and illustrate the claimed role of the smoothing covariance and the preconditioner. I also appreciate that the paper is reasonably careful about the scope of its results. That said, the analysis is carried out in a fairly stylized setting, including continuous time ALD, Gaussian mixture targets, co-diagonalizable covariances, and a misspecified mixture model for score error. There is also no discretization analysis, so the results feel more like a continuous time theoretical result than a practically complete sampling analysis. I did not check every proof detail in the appendix.

Presentation: The paper is generally well written and easy to follow. One suggestion is that some of the proofs seem to rely on fairly simple core ideas, and it would help to state those ideas more explicitly in the main body before giving the formal argument. That would make the paper easier to read and would improve intuition for the main results.

Significance: I am not yet convinced that the paper answers a major theoretical question or provides clear practical benefits, so I currently view the significance as somewhat limited and specialized.

Originality: I am not deeply familiar with all of the closest related theory papers, but the dimension robust multimodal ALD perspective and the explicit emphasis on the spectral role of preconditioning seem novel to me.

---

> ### Author Rebuttal · Authors · 2026-03-30
>
> We thank the reviewer for recognizing the theoretical nature of our contribution and address the points below.
>
> **Discretization.** Our goal is to isolate the dimension-dependent effect of preconditioning in multimodal annealed Langevin dynamics in infinite dimensions, which remains largely open beyond unimodal settings [6,8]. The continuous-time framework makes this mechanism explicit without conflating discretization effects. A discretization analysis would be a natural extension, building on existing techniques [1], and we expect similar behavior, as supported by our numerics and prior evidence showing the lack of dimension-uniformity without preconditioning.
>
> **Soundness.** The co-diagonalizability assumption  is not needed uniformly throughout the paper.
> In particular, it can  be removed in Theorem 3.1, leading to the sufficient condition
> $$
> \sup_{d\ge 1}
> \sum_{i\in I} w_i
> \operatorname{Tr} \Big(
> (\Gamma^d)^{-1}C^d(\Sigma_i^d)^{-1}C^d
> \Big)
> <\infty,
> $$
> and
> similarly in Proposition 4.1 for the analysis of $\mathcal E_{\text{init}}^d$ and $\mathcal E_{\text{bias}, T}^d$. The diagonal assumptions are only needed in Proposition 4.4, in the refined score-error robustness analysis. Such structure is standard in infinite-dimensional analysis [6-8] and allows us to make the spectral role of preconditioning explicit in the multimodal setting. Moreover, under the successive finite-dimensional truncations on which the analysis is based, diagonal Gaussian mixtures remain highly expressive and can approximate general mixtures in KL divergence. The diagonal structure should be understood as a
> tractable and flexible representation that preserves infinite-dimensional analytical control while still allowing rich
> multimodal behavior at every finite resolution. We will clarify this point.
>
> **Significance.** Our paper addresses a difficult and open problem concerning the role of dimension in annealed Langevin dynamics, and in particular how preconditioning interacts with multimodality beyond the unimodal setting as one moves toward the infinite-dimensional regime. This is also practically important for discretization-robust sampling, where performance should remain stable under refinement. Our main result shows that suitable preconditioning is essential to prevent coordinate-wise error accumulation. We make this mechanism explicit in a setting that goes beyond existing analyses, which are finite-dimensional, rely on stronger structural assumptions and cannot handle countably infinite mixtures [2-5]. Combined with the high expressivity of our model class, this supports the relevance of our framework. We therefore view our work as a meaningful theoretical contribution, with the practical message that preconditioning is key to discretization-invariant sampling - a point that has only recently started to receive renewed attention, though so far mainly in simpler, unimodal settings [6,8].
>
> **Presentation.** We will revise the paper to highlight the main ideas more clearly in the proofs.
>
> **Question 1.** The main practical takeaway is that, for multimodal targets from infinite-dimensional problems, preconditioning is essential for ALD to remain stable under truncation when the score is perturbed. Our analysis shows this is a genuine dimension-dependent spectral effect, not a discretization artifact. Thus, even in finite-dimensional implementations, ALD should be paired with an appropriate preconditioner as the discretization is refined, otherwise performance will degrade with dimension. We will emphasize this point more clearly in the experiments section.
>
> **Question 2.** We address this point in detail in our **Discretization analysis**, **Stylized setting**, and **Significance** answers. We will also incorporate part of that discussion into the revised version of the paper, to help readers better assess the significance of the work.
>
> **Question 3.** The role of preconditioning in obtaining dimension-uniform control is not new, and is already well understood for unimodal targets [8].  Since GMMs form a flexible  approximation class for general multimodal distributions [9], we expect this message to extend  more broadly to multimodal ALD beyond the specific setting studied here. We will make this point more explicit in the revision,  and will also clarify that the structured Gaussian-mixture family considered in the paper is less restrictive than it may appear at first sight. In this way, we hope to better convey both the significance of the analysis and its potential practical implications.
>
>
> **References**
>
> [1] W. Guo et al., arXiv:2407.16936
>
> [2] P. Cordero-Encinar et al., arXiv:2502.09306
>
> [3] R. Ge et al.,  arXiv:1710.02736
>
> [4] A. Vacher et al., arXiv:2501.00565
>
> [5] R. Han, arXiv:2508.02763
>
> [6] J. Pidstrigach et al.,  arXiv:2302.10130
>
> [7] L. Baldassari et al., arXiv:2305.19147
>
> [8] L. Baldassari et al., arXiv:2505.18276
>
> [9] G. McLachlan et al., Finite mixture models, 2000

---

> > ### Author Rebuttal · Reviewer_AZAG · 2026-04-02
> >
> > Q1: Would you mind commenting more on why Assumption 4.2 is a reasonable assumption to make?
> >
> > Q2: Would you mind giving some concrete examples of potentials that practitioners are interested in sampling from using ALD and explaining why GMMs approximate them well, or why you expect the theory to apply in those settings too?

---

> > > ### Author Response · Authors · 2026-04-02
> > >
> > > We are happy to answer your questions. We will include part of this discussion in the revised version of the paper.
> > >
> > > **Q1:** Our paper takes a sampling, rather than a generative-modelling, perspective. Accordingly, Section 4 studies the robustness of ALD under drift perturbations. This is natural in sampling theory, where one often treats the drift as an imperfect oracle and asks how such perturbations affect the resulting sampler [10]. Thus, the purpose of Section 4 is not to model the full statistical process by which a score is learned, but to analyze the robustness of ALD under explicit multimodal score misspecification in infinite dimensions.
> > > For this purpose, a black-box score-error bound is not the right model: it does not track how the multimodal geometry is changed. Assumption 4.2 captures those geometric changes. It perturbs the mixture weights, means, and covariances, and therefore allows for mode reweighting, mean displacement, covariance distortion, including regimes in which some modes become nearly negligible. Moreover, it is not restricted to finitely many Gaussian-mixture components, but also covers countably infinite mixtures. In this sense, it provides an explicit and general misspecification model for the multimodal setting we study, while remaining tractable enough for rigorous analysis.
> > > This is also practically motivated. In applications, approximate scores may distort the target in a structured way, e.g. by misweighting modes, shifting their locations, or misestimating their local covariance structure. These are the kinds of errors that can affect sampling behavior. Assumption 4.2 isolates this regime and makes it possible to study whether such errors accumulate or remain controlled as the dimension grows.
> > >
> > >
> > > **Q2:** A natural class of targets for annealing is given by Gibbs measures associated with nonconvex, multi-well energies. If $\pi(x)\propto e^{-V(x)},$ and $V$ has many wells, then overdamped Langevin dynamics may mix very slowly because it can remain trapped for long times inside metastable basins before crossing the barriers separating them ([11] and references therein). This is exactly the regime in which ALD is useful: one first explores smoother distributions and only gradually returns to the original target (see Figure 2 of our submission).
> > > This is especially relevant in infinite-dimensional settings arising from continuum field models. One example is [12], whose invariant measure is associated with an energy functional with double-well potential. After discretization, this yields a high-dimensional multi-basin target. Another example is molecular dynamics [17]. We also expect our analysis to be relevant for posterior sampling in nonlinear inverse problems, where classical Langevin is known to struggle [13]. Recent works also use ALD for posterior sampling [15]; as discussed in our paper, we expect our analysis to help motivate similar questions in infinite dimensions, which is the natural setting for many most inverse problems.
> > > GMMs should be viewed as explicit and tractable surrogates for multimodality. This is common in theoretical analyses and is justified by the fact that a GMM is a universal approximator of densities [16, Section 3.9.6]; see also [14]. Near a nondegenerate local minimum of a smooth potential, a 2nd-order Taylor expansion gives a quadratic local energy, hence a Gaussian approximation of the associated local Gibbs weight. After smoothing/tempering, this local approximation becomes more accurate. Thus, when a target has several dominant basins, a Gaussian mixture is a natural global surrogate: the weights encode the relative importance of the basins, the means their locations, and the covariances their local anisotropy.  Finally, since our setting is built from successive finite-dimensional refinements, and since diagonal GMMs already approximates well general GMMs, allowing countably infinite mixtures gives a particularly rich model class.
> > > For this reason, we expect the key points of our analysis to remain relevant beyond exact GMM targets.In that case, one would expect the total error to consist of the terms analyzed in our paper plus an additional approximation term measuring the discrepancy between the true target and its GMM surrogate.
> > > Finally, the infinite-dimensional viewpoint is important because many practically relevant multimodal targets arise as discretizations of functions. In this sense, the question of whether ALD remains stable as the discretization dimension increases is not merely technical; it is connected to whether one can sample reliably from increasingly fine approximations of a multimodal target.
> > >
> > >
> > > [10] J.H. Huggins et al., arXiv:1605.06420
> > >
> > > [11] A. Christie et al., arXiv:2303.18168
> > >
> > > [12] F. Otto et al., arXiv:1301.0408
> > >
> > > [13] R. Nickl, Bayesian non-linear statistical inverse problems
> > >
> > > [14] Y. Huang, arXiv:2509.25232
> > >
> > > [15] Z. Xun et al., arXiv:2510.26324
> > >
> > > [16] I. Goodfellow et al., Deep Learning
> > >
> > > [17] Z. Ding, arXiv:2411.13443

---

### Official Review · Reviewer_7Ajd · 2026-03-23

**Soundness:** 2
**Presentation:** 3
**Significance:** 2
**Originality:** 3
**Overall Recommendation:** 3
**Confidence:** 4

**Summary:**

This work investigates the theoretical properties of the annealed Langevin dynamics for multimodal distributions. It focuses on the infinite-dimensional setting and assumes the target is a GMM. The authors proposes a precondition modification to improve the dimensional dependence.

**Compliance With Llm Reviewing Policy:**

Affirmed.

**Final Justification:**

My main concern remains after the rebuttal. The complexity of sampling has a clear definition as the number of calls to the potential or its gradient. The score approximation error shouldn't be treated as a separate issue. I will keep my original score.

**Key Questions For Authors:**

Question:
1. Given an approximation of score, is ALD the most effective algorithm?
2. How to choose preconditioner effectively?

**Limitations:**

yes

**Strengths And Weaknesses:**

Strength:
1. The general problem considered in this paper: sampling from infinite-dimensional distributions is an extremely challenging problem.
2. The paper is reasonably well written and easy to follow.

Weakness:
1. The analysis focuses on a very specific setting with a structured GMM. It is unclear whether the insights learned in this paper has any indication on general cases.
2. From the presentation, it is unclear whether the authors try to solve a sampling problem or a generative modeling problem. I assume it is the sampling problem. In ALD for sampling problems, the annealed distributions are assumed to have closed form and be computable directly using the expression of the target distribution. This is clearly not the case in this paper.
3. In this paper, an approximation of the noisy score function is given. It is unclear how such a score is obtained in the sampling setting and cost to obtain the score is ignored.
4. Assumption 4.2 is artificial.
5. This is no comparison to diffusion model, which can also sample from the target given approximated score.

---

> ### Author Rebuttal · Authors · 2026-03-30
>
> We thank the Reviewer for the feedback.
> We are happy to clarify the paper, especially regarding its sampling perspective. We hope this supports a higher score.
>
> **W1.** This is a key aspect of our contribution. First, our framework is broader than many existing analyses of ALD sampling, which are mostly finite-dimensional and impose stronger structural assumptions (e.g., finitely many Gaussian modes with shared covariances [1,2]). Even more recent multimodal works remain finite and do not cover countably infinite mixtures [3,4], while spectral decompositions are standard in infinite dimensions [5-6].
> Second, several parts of our analysis extend beyond the diagonal case: Theorem 3.1 admits a natural extension leading to the sufficient condition
> $$
> \sup_{d\ge 1}\sum_{i\in I}w_i\operatorname{Tr} ((\Gamma^d)^{-1}C^d(\Sigma_i^d)^{-1}C^d)
> <\infty.
> $$
> Likewise, the analysis of the initialization and bias terms in Proposition 4.1 is not inherently tied to the diagonal case. The main point where diagonality is required is the refined score-error analysis in Proposition 4.4.
>
> Third, the diagonal assumption is less restrictive at the level of finite-dimensional truncations on which the analysis is based: for each fixed dimension, diagonal Gaussian mixtures remain highly expressive and can approximate general mixtures in KL distance. It should therefore be seen as a tractable representation enabling infinite-dimensional control while allowing rich multimodal behavior at each finite resolution.
> We agree that this point was not clear enough and will revise the paper to better explain the expressivity of the model and the scope of the theory.
>
> **W2.** We thank the reviewer for this remark. We agree that the term "learned score" in Assumption 4.2 may be misleading and will revise it to avoid suggesting a generative modelling perspective. Section 4 instead provides a robustness analysis for sampling: it quantifies how dimension-uniform control degrades under drift perturbations and how preconditioning mitigates error accumulation. It should therefore be viewed as a perturbation analysis of the sampling dynamics, rather than a score-learning procedure. We will clarify this point in the revision.
>
> **W3.** Since this paper takes a sampling perspective, not a generative model one, Section 4 is not meant to explain how an approximate score is obtained or what its cost is, but to study the robustness of ALD once such a score is given. This is natural from the viewpoint of sampling theory, where one often works with an imperfect drift and asks how it affects the sampler [7]. We will clarify this point in the revision.
>
> **W4.** We thank the reviewer for this comment but disagree that Assumption 4.2 is artificial. Section 4 does not aim to model the full training of a score network, but to analyze the robustness of ALD under explicit multimodal score misspecification in infinite dimensions. A generic norm-based error bound is insufficient: it does not capture how multimodal geometry is altered. In contrast, Assumption 4.2 explicitly perturbs mixture weights, means, and covariances, allowing for mode reweighting, shifts, and distortions. This provides a flexible yet tractable misspecification model. To our knowledge, prior infinite-dimensional works do not address multimodal score misspecification in this way. Our contribution is to make this setting explicit and show that preconditioning prevents coordinate-wise error accumulation under dimension-uniform conditions.
>
> **W5+Q1.** Our goal is not to compare ALD with other approximate-score or diffusion-based samplers, but to address a specific theoretical question: for preconditioned ALD in a multimodal infinite-dimensional setting, under which explicit spectral conditions can one obtain dimension-uniform control, and how do score misspecification and initialization errors scale with dimension?A direct comparison with diffusion models is not straightforward, as this term encompasses a wide range of methods with different dynamics and approximations. A meaningful comparison would require fixing a specific model and analysis framework, which lies beyond the scope of this work.
>
> **Q2.** We thank the reviewer for this question. Our analysis suggests that the preconditioner should be chosen by balancing two aspects. Theorem 3.1, which controls the annealing-induced bias, excludes choices that are too small relative to the smoothing scale, since this produce an unfavorable dimension-dependent spectral mismatch. On the other hand, the score-error analysis of Section 4 excludes choices that are too large in directions where the score is inaccurate, since then the preconditioner may amplify the misspecification in the drift.
>
>
> **Refs**
>
> [1] Cordero-Encinar et al., arXiv:2502.09306
>
> [2] Ge et al., arXiv:1710.02736
>
> [3] Vacher et al., rXiv:2501.00565
>
> [4] Han, arXiv:2508.02763
>
> [5] Pidstrigach et al., arXiv:2302.10130
>
> [6] Baldassari et al., arXiv:2505.18276
>
> [7] Huggins et al., arXiv:1605.06420

---

> > ### Author Rebuttal · Reviewer_7Ajd · 2026-04-04
> >
> > I would like to thank the authors for the rebuttal. Unfortunately, my main concern remains. If this paper takes a sampling standpoint, then the a complete complexity of drawing a sample should be analyzed. In the adopted diffusion-based ALD algorithm, this means the complexity of estimating the score up to the accuracy the paper assumes should be accounted. Otherwise, it is possible that the complexity of this score estimation is main computational bottleneck but overlooked.

---

> > > ### Author Response · Authors · 2026-04-05
> > >
> > > We would like to argue that our paper is formulated at the standard level of abstraction of sampling theory, where one analyzes the sampler under an inexact oracle rather than the cost of constructing that oracle. This is standard in the sampling literature [1,2,3]. From this perspective, the question studied in the paper is how score misspecification propagates through ALD, and under which spectral conditions dimension-uniform control remains possible. The Reviewer's comment conflates two distinct issues: the complexity of constructing a score approximation, and the perturbation analysis needed to understand how such misspecification affects the sampler once such an approximation is given. Our analysis concerns the latter. The former is certainly interesting and challenging, but addressing it here would change the scope of the paper rather than fill a gap in the present contribution.
> > >
> > > [1] Huggins, arXiv:1605.06420
> > >
> > > [2]  Dalalyan, arXiv:1710.00095
> > >
> > > [3] Dalalyan, arXiv:1412.7392

---

### Decision · Program_Chairs · 2026-04-30

**Decision:**

Accept (regular)

**Comment:**

The paper shows a setting where annealed Langevin dynamics has dimension-free dependence in continuous time, for distributions arising as truncations of an infinite-dimensional distribution approximable by Gaussian mixture models, for the interpolation path given by Gaussian convolution. Reviewers found the paper well presented, theoretical contribution valuable, and the result of the necessity of preconditioning to be insightful. The main concern is that the model is quite stylized, and relies on score functions of the intermediate distributions being available (so cannot just use info on the target distribution as a black-box). Given this, I find the results valuable as long as the paper is properly scoped.